# IMPROVED PROBABILISTIC REGRESSION USING DIFFUSION MODELS

## ABSTRACT

Probabilistic regression models the entire predictive distribution of a response variable, offering richer insights than classical point estimates and directly allowing for uncertainty quantification. While diffusion-based generative models have shown remarkable success in generating complex, high-dimensional data, their usage in general regression tasks often lacks uncertainty-related evaluation and remains limited to domain-specific applications. We propose a novel diffusion-based framework for probabilistic regression that learns predictive distributions in a nonparametric way. More specifically, we propose to model the full distribution of the diffusion noise, enabling adaptation to diverse tasks and enhanced uncertainty quantification. We investigate different noise parameterizations, analyze their trade-offs, and evaluate our framework across a broad range of regression tasks, covering low- and high-dimensional settings. For several experiments, our approach shows superior performance against existing baselines, while delivering calibrated uncertainty estimates, demonstrating its versatility as a tool for probabilistic prediction.

## 1 INTRODUCTION

Supervised regression concerns predicting a response variable $y \in \mathcal{Y}$ from covariates $c \in \mathcal{C}$. Classical approaches estimate the conditional mean $\mathbb{E}[y \mid c]$, whereas probabilistic regression models the full predictive distribution $p_{\mathcal{Y}}(y \mid c)$ (Bishop, 1994; Shen & Meinshausen, 2024). The latter provides calibrated uncertainty estimates and captures complex data-generating processes, including multimodal outcomes and nonlinear noise. We adopt a broad notion of regression, encompassing heterogeneous data types such as scalars, images, and trajectories. A key modeling choice is the specification of $p_{\mathcal{Y}}(\cdot \mid c)$: while Gaussian assumptions remain widespread (Lakshminarayanan et al., 2017; Nix & Weigend, 1994), recent work has emphasized more flexible non-parametric and distribution-free alternatives (Kelen et al., 2024; Shen & Meinshausen, 2024).

Diffusion-based generative models have emerged as state-of-the-art approaches for high-dimensional data generation, achieving remarkable results in tasks such as photorealistic image (Rombach et al., 2022) and video synthesis (Ho et al., 2022). These models are typically formulated as diffusion probabilistic models (Sohl-Dickstein et al., 2015; Ho et al., 2020), where data are gradually perturbed by Gaussian noise (the forward process) and new samples are generated via the time-reversed dynamics. Prior work has advanced diffusion models through improved noise schedules (Nichol & Dhariwal, 2021; Karras et al., 2022) and sampling algorithms (Song et al., 2021; Bortoli et al., 2025), where the focus has largely been on accelerating the diffusion process while achieving a high generational quality (Song & Ermon, 2019; Nichol & Dhariwal, 2021).

Recently, diffusion models have been applied to various regression tasks such as depth estimation (Ke et al., 2024), autoregressive flow prediction (Kohl et al., 2024; Finzi et al., 2023), and weather forecasting (Price et al., 2025; Couairon et al., 2024), often achieving state-of-the-art performance. Despite their inherently probabilistic nature, evaluations rarely emphasize uncertainty-related metrics or calibration. Recent efforts that aim to extract uncertainty estimates from diffusion models (Shu & Farimani, 2024; Chan et al., 2024; Berry et al., 2024) typically rely on training multiple networks, incurring substantial computational overhead. Furthermore, the intimate relation between uncertainty quantification and the noise modeling within the diffusion process remains underexplored.

**Contributions:** In this work, we address these limitations by adapting the diffusion process to yield calibrated probabilistic predictions. Building on recent advances from generative modeling (Bortoli et al., 2025), we introduce a novel loss for diffusion-based regression models that enables learning a flexible noise distribution, moving beyond optimization of the conditional mean. We derive several trainable parameterizations that offer task-specific trade-offs between expressivity and computational efficiency while admitting closed-form sampling at the same time. These extensions provide principled epistemic uncertainty estimation within diffusion models, and we extensively validate our method across diverse regression tasks—including UCI benchmarks, flow prediction, weather forecasting, and depth estimation—showing consistent improvement in predictive performance and substantial gains in uncertainty quantification and calibration.

## 2 BACKGROUND

### 2.1 PROBABILISTIC REGRESSION

Let $\boldsymbol{y} \in \mathcal{Y} \subseteq \mathbb{R}^{d_y}$ denote the response variables of interest and $\boldsymbol{c} \in \mathcal{C} \subseteq \mathbb{R}^{d_c}$ the corresponding conditioning variables. Classical regression focuses on estimating the conditional mean $\mathbb{E}[\boldsymbol{y} \mid \boldsymbol{c}]$, whereas probabilistic regression seeks to model the full conditional distribution $p_{\mathcal{Y}}(\boldsymbol{y} \mid \boldsymbol{c})$, thereby explicitly capturing predictive uncertainty. Given training data $\mathcal{D} = (\boldsymbol{c}_i, \boldsymbol{y}_i)_{i=1}^{N}$, the goal is to recover the predictive distribution $p_{\mathcal{Y}}(\boldsymbol{y} \mid \boldsymbol{c}, \mathcal{D})$.

The setting is deliberately chosen to be general, including classical regression approaches, but also autoregressive prediction tasks, for example, when choosing $\boldsymbol{y} = \boldsymbol{y}_{t+1}, \boldsymbol{c} = \boldsymbol{y}_t, \boldsymbol{y}_t \in \mathcal{Y}, \forall t = 1, \ldots, T$. A common strategy is to parameterize the conditional distribution via a generative mapping

$$f_\theta : \mathcal{C} \times \mathcal{Z} \to \mathcal{Y}, \quad \theta \in \Theta \subseteq \mathbb{R}^p, \tag{1}$$

where $\boldsymbol{z} \sim p_{\mathcal{Z}}$ is drawn from a source distribution, typically Gaussian. The parameters $\theta$ are then optimized such that $f_\theta(\boldsymbol{c}, \cdot) \approx p_{\mathcal{Y}}(\cdot \mid \boldsymbol{c})$.

In this work, we depart from classical parameterizations and instead employ diffusion-based generative models to represent $p_{\mathcal{Y}}(\boldsymbol{y} \mid \boldsymbol{c}, \mathcal{D})$. For notational simplicity, we omit the explicit dependence on $\mathcal{D}$ in the following.

### 2.2 DIFFUSION MODELS

Diffusion probabilistic models (DPMs) aim to learn a target distribution $p_0$ on $\mathbb{R}^d$ from samples by estimating the reverse dynamics of a diffusion process. We follow the non-Markovian formulation of denoising diffusion implicit models (DDIM) (Song et al., 2021).

Let $T \in \mathbb{N}, \boldsymbol{x}_0 \sim p_0$, and $\beta_{1:T} \in [0,1]^T$ denote a noise schedule. Define $\alpha_t := 1 - \beta_t$ and $\bar{\alpha}_t := \prod_{i=1}^{t} \alpha_i$. The *forward process* is specified as $p(\boldsymbol{x}_{1:T} \mid \boldsymbol{x}_0) := p(\boldsymbol{x}_T \mid \boldsymbol{x}_0) \prod_{t=2}^{T} p(\boldsymbol{x}_{t-1} \mid \boldsymbol{x}_t, \boldsymbol{x}_0)$, where, for $t > 1$,

$$p(\boldsymbol{x}_{t-1} \mid \boldsymbol{x}_t, \boldsymbol{x}_0) = \mathcal{N}\Big(\underbrace{\sqrt{\bar{\alpha}_{t-1}}\boldsymbol{x}_0 + \sqrt{1 - \bar{\alpha}_{t-1} - \sigma_t^2} \cdot \frac{\boldsymbol{x}_t - \sqrt{\bar{\alpha}_t}\boldsymbol{x}_0}{\sqrt{1 - \bar{\alpha}_t}}}_{=:\boldsymbol{\mu}(\boldsymbol{x}_0, \boldsymbol{x}_t)}, \sigma_t^2 \mathbf{I}\Big). \tag{2}$$

We set $p(\boldsymbol{x}_T \mid \boldsymbol{x}_0) := \mathcal{N}(\sqrt{\bar{\alpha}_T}\boldsymbol{x}_0, (1 - \bar{\alpha}_T)\mathbf{I})$ and require $\bar{\alpha}_T$ to be sufficiently small such that $p(\boldsymbol{x}_T \mid \boldsymbol{x}_0) \approx \mathcal{N}(\mathbf{0}, \mathbf{I})$. The variance schedule is parameterized as $\sigma_t := \eta\sqrt{\tilde{\beta}} := \eta\sqrt{\frac{1-\bar{\alpha}_{t-1}}{1-\bar{\alpha}_t}\beta_t}$ with $\eta \in [0,1]$ interpolating between the deterministic DDIM process ($\eta = 0$) and the stochastic DDPM process ($\eta = 1$) (Ho et al., 2020).

An important feature of this construction is that for all choices of $\sigma_t$ and for all $t = 1, \ldots, T$, we obtain the property

$$p(\boldsymbol{x}_t \mid \boldsymbol{x}_0) = \mathcal{N}\left(\sqrt{\bar{\alpha}_t}\boldsymbol{x}_0, (1 - \bar{\alpha}_t)\mathbf{I}\right), \tag{3}$$

which yields the identity

$$\boldsymbol{X}_t = \sqrt{\bar{\alpha}_t}\boldsymbol{X}_0 + \sqrt{1 - \bar{\alpha}_t}\epsilon_t, \quad \epsilon_t \sim \mathcal{N}(\mathbf{0}, \mathbf{I}), \tag{4}$$

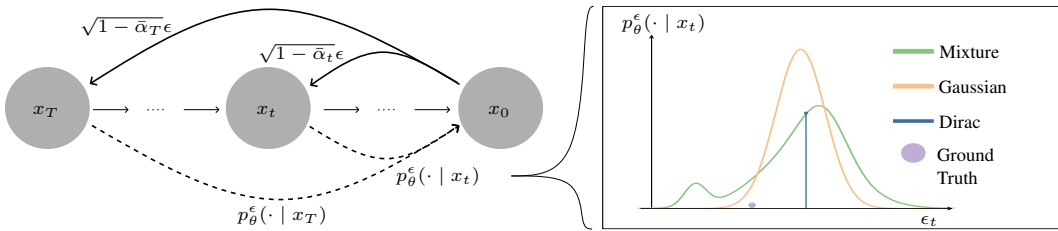

Figure 1: Method overview. At any given time $t = 1, \ldots, T$ in the diffusion process, we need a prediction of the noise $\epsilon_t$ given our current state $x_t$. Traditionally, this is achieved by a network that approximates the conditional mean $\mathbb{E}[\epsilon_t \mid x_t]$, which, when viewed under a probabilistic viewpoint, treats the distribution $p^\epsilon(\cdot \mid x_t)$ as a Dirac distribution. We propose to model this distribution by a Gaussian Mixture family.

for $\epsilon_t$ independent from $\boldsymbol{X}_0$, highlighting that $\boldsymbol{X}_t$ is defined by adding a specified amount of noise $\epsilon_t$ to the original input $\boldsymbol{X}_0$.

To generate samples, one must approximate the reverse process $p(\boldsymbol{x}_{t-1} \mid \boldsymbol{x}_t)$, which is intractable in general. DPMs approximate it with a latent-variable model

$$p_\theta(\boldsymbol{x}_{0:T}) \coloneqq p_\theta(\boldsymbol{x}_T) \prod_{t=1}^{T} p_\theta(\boldsymbol{x}_{t-1} \mid \boldsymbol{x}_t), \tag{5}$$

which is a Markov chain that samples from $\boldsymbol{x}_T$ to $\boldsymbol{x}_0$, that is referred to as the *generative process*. For sufficiently small $\beta_t$, the reverse transition $p(\boldsymbol{x}_{t-1} \mid \boldsymbol{x}_t)$ is well approximated by a Gaussian, thus allowing to set $p_\theta(\boldsymbol{x}_T) \sim \mathcal{N}(\boldsymbol{0}, \boldsymbol{I})$ and specifying the latent variable model as a neural network, $p_\theta(\boldsymbol{x}_{t-1}|\boldsymbol{x}_t) = \mathcal{N}(\boldsymbol{x}_{t-1}; \boldsymbol{\mu}_\theta(\boldsymbol{x}_t, t), \boldsymbol{\Sigma}_\theta(\boldsymbol{x}_t, t))$.

Training proceeds by minimizing the variational lower bound (VLB), which reduces to a KL divergence between $p_\theta(\boldsymbol{x}_{t-1} \mid \boldsymbol{x}_t)$ and the true posterior $p(\boldsymbol{x}_{t-1} \mid \boldsymbol{x}_t, \boldsymbol{x}_0)$. DDIM fixes the covariance to $\boldsymbol{\Sigma}_\theta = \sigma_t \mathbf{I}$, while DDPM considers $\boldsymbol{\Sigma}_\theta \in \{\beta_t \mathbf{I}, \tilde{\beta}_t \mathbf{I}\}$. With these simplifications, the VLB objective reduces (after reweighting) to a mean-squared error:

$$L_{\text{simple}}(\theta) = \mathbb{E}_{t,x_0,\epsilon_t} \left[ \|\epsilon_t - \epsilon_\theta(\boldsymbol{x}_t, t)\|_2^2 \right]. \tag{6}$$

To perform conditional modeling with covariates $\boldsymbol{c}$, the target distribution is set as $p_0 = p_{\mathcal{Y}}(\cdot \mid \boldsymbol{c})$, yielding an approximate predictive distribution $p_\theta(\cdot \mid \boldsymbol{c}) \approx p_{\mathcal{Y}}(\cdot \mid \boldsymbol{c})$. For simplicity, we omit explicit conditioning in the notation.

While DDPM and DDIM have shown great success in conditional and unconditional generative modeling, note that due to the objective in Equation (6), the latent variable model only learns to approximate $\epsilon_\theta(\boldsymbol{x}_t, t) \approx \mathbb{E}[\epsilon_t \mid \boldsymbol{x}_t]$ and does not capture information about the full distribution $p(\epsilon_t \mid \boldsymbol{x}_t)$.

## 2.3 SCORING RULES

Our methodology builds upon *scoring rules* (Gneiting & Raftery, 2007), which have been widely used for training neural networks in probabilistic prediction tasks (Pacchiardi et al., 2024; Bülte et al., 2025; Chen et al., 2022). Let $\mathcal{P}$ be a convex set of probability measures over a measurable space $\mathcal{Y}$, and let $\mathbb{P}, \mathbb{Q} \in \mathcal{P}$. A *scoring rule* is a function $S : \mathcal{P} \times \mathcal{Y} \to \mathbb{R}$, which assigns a numerical score $S(\mathbb{P}, y)$ to a predictive distribution $\mathbb{P}$ upon observing an outcome $y \in \mathcal{Y}$ (Gneiting & Raftery, 2007). This score desribes the discrepancy of observing $y$ under the predictive distribution $\mathbb{P}$. The corresponding *expected score* under a distribution $\mathbb{Q}$ is defined as

$$S(\mathbb{P}, \mathbb{Q}) \coloneqq \mathbb{E}_{Y \sim \mathbb{Q}}[S(\mathbb{P}, Y)].$$

A scoring rule is called *proper* if

$$S(\mathbb{Q}, \mathbb{Q}) \leq S(\mathbb{P}, \mathbb{Q}), \quad \forall \mathbb{P}, \mathbb{Q} \in \mathcal{P},$$

and *strictly proper* if equality holds only when $\mathbb{P} = \mathbb{Q}$. Strict propriety guarantees that the true data-generating distribution uniquely minimizes the expected score, making strictly proper scoring rules particularly suitable as loss functions for probabilistic models.

# 3 OUR METHODOLOGY

Our goal is to improve uncertainty quantification in regression tasks with diffusion models by generalizing the diffusion loss, thereby enabling more accurate approximation of the conditional distribution $p_{\mathcal{Y}}(\cdot \mid \boldsymbol{c})$.

## 3.1 LEARNING THE FULL DISTRIBUTION IN EACH DENOISING STEP

Most diffusion frameworks, following DDPM, fix the variance of the noise distribution in each denoising step. This design was originally motivated by two observations: (i) learning the variance often destabilized training, and (ii) variance modeling showed little benefit for image generation benchmarks, for example, with respect to the FID Heusel et al. (2018).

However, subsequent work (Nichol & Dhariwal, 2021) demonstrated that learning the variance improves likelihood estimates, indicating that recovering only the mean is insufficient for faithfully approximating the conditional distribution. Furthermore, the Gaussian approximation of $p(\boldsymbol{x}_{t-1} \mid \boldsymbol{x}_t)$ is only valid when the number of timesteps $T$ is large. Yet, large $T$ is computationally costly, and recent results suggest that using as few as 20–50 steps can yield superior performance in regression tasks (Kohl et al., 2024; Price et al., 2025), particularly with improved noise schedulers and solvers (Karras et al., 2022; Chung et al., 2022).

These observations motivate us to go beyond estimating the first two moments and instead learn the full distribution of $\epsilon_t$. Concretely, we reinterpret $\epsilon_\theta$: rather than treating it as a point estimate of $\epsilon_t$, we view it as a random variable. This perspective naturally suggests replacing the mean-squared error loss $L_{\text{VLB}}$ with a criterion that compares probability distributions rather than point predictions. To this end, we adopt the framework of strictly proper scoring rules (Dawid et al., 2016; Pacchiardi et al., 2024), see Section 2.3.

Thus, we propose the loss

$$L_{SR} = \mathbb{E}_{t,x_0,\epsilon_t}[S(p_\theta^\epsilon(\cdot \mid \boldsymbol{x}_t), \epsilon_t)] \tag{7}$$

where $p_\theta^\epsilon$ is a neural network–based model of the predictive distribution of $\epsilon_t$ and $S$ denotes a strictly proper scoring rule. This objective enables learning general noise distributions while ensuring that training accounts for the entire distribution rather than just its moments.

Section 3.2 connects this formulation to recent theoretical advances that justify our approach, and Section 3.3 details design choices for both the noise distribution $p_\theta^\epsilon$ and scoring rule $S$.

## 3.2 THEORETICAL JUSTIFICATION

Concurrent work by Bortoli et al. (2025) reached a similar conclusion, proposing the use of proper scoring rules as a principled way to learn the full posterior distribution over noisy samples. Consider the reverse transition

$$p(\boldsymbol{x}_{t-1} \mid \boldsymbol{x}_t) = \int_{\mathbb{R}^d} p(\boldsymbol{x}_{t-1} \mid \boldsymbol{x}_0, \boldsymbol{x}_t)\, p(\boldsymbol{x}_0 \mid \boldsymbol{x}_t)\, d\boldsymbol{x}_0, \tag{8}$$

where the intractable posterior $p(\boldsymbol{x}_0 \mid \boldsymbol{x}_t)$ must be approximated.

In standard DDPM/DDIM frameworks, this posterior is replaced by a point mass, $p(\boldsymbol{x}_0 \mid \boldsymbol{x}_t) \approx \delta_{\hat{\boldsymbol{x}}_0(t,\boldsymbol{x}_t)}$, where $\hat{\boldsymbol{x}}_0(t,\boldsymbol{x}_t)$ is the prediction of the denoising network. Trained with a regression loss, the denoiser recovers $\hat{\boldsymbol{x}}_0(t,\boldsymbol{x}_t) \approx \mathbb{E}[X_0 \mid X_t = \boldsymbol{x}_t]$, yielding the DDIM formulation. For $T \to \infty$, this procedure recovers the data distribution. However, when using only a small number of timesteps—a setting of practical interest—the approximation may no longer hold.

To address this, Bortoli et al. (2025) proposed learning the full posterior distribution $p(\boldsymbol{x}_0 \mid \boldsymbol{x}_t)$ via proper scoring rules. They mainly focus on *generalized kernel scores*, which are of the form

$$S_{\lambda,\rho}(p, \boldsymbol{y}) = \mathbb{E}_p[\rho(X, \boldsymbol{y})] - \frac{\lambda}{2}\mathbb{E}_{p \otimes p}[\rho(X, X')] - \frac{1}{2}\rho(\boldsymbol{y}, \boldsymbol{y}), \tag{9}$$

for kernel $\rho$. Special cases include the *energy score* (Gneiting & Raftery, 2007) for $\rho(\boldsymbol{x}', \boldsymbol{x}) = \|\boldsymbol{x}' - \boldsymbol{x}\|^\beta$, $\beta \in (0, 2)$, and the *(Gaussian) kernel score* with $\rho(\boldsymbol{x}', \boldsymbol{x}) = -\exp(-\|\boldsymbol{x} - \boldsymbol{x}'\|^2/\gamma^2)$. Importantly, Bortoli et al. (2025) show that both recover the classical diffusion regression loss in the limit when combining (7) with (9), a property termed *diffusion compatibility*.

A second design choice concerns the parametrization of $p(\boldsymbol{x}_0 \mid \boldsymbol{x}_t)$. Following Shen & Meinshausen (2024), Bortoli et al. (2025) propose to concatenate Gaussian noise to the neural network input, thereby generating $M$ samples of the target distribution. Training is then carried out using an unbiased estimator of Equation (9). This strategy provides a flexible, nonparametric estimate of $p(\boldsymbol{x}_0 \mid \boldsymbol{x}_t)$, but at significant computational cost: multiple samples $M$ increase both the input dimensionality and runtime. Empirically, training is reported to be $1.3\times$ to $7\times$ slower than standard diffusion models, even for moderate $M$ (Bortoli et al., 2025), and may become prohibitive for large-scale datasets.

## 3.3 Parametrization of $p_\theta^\epsilon(\cdot \mid \boldsymbol{x}_t)$

As an alternative to nonparametric sampling-based approaches, we propose to model $p_\theta^\epsilon(\cdot \mid \boldsymbol{x}_t)$ directly through a parametrized distribution. This aligns naturally with scoring rule minimization, since closed-form expressions for the training objective are available. A schematic overview is given in Figure 1. Depending on the choice of parametrization, one obtains a trade-off between computational efficiency and flexibility, which may vary across tasks: while simple Gaussian models may suffice in some applications, more complex structures are beneficial for multimodal or correlated noise patterns.

Specifically, we consider the general Gaussian mixture form

$$p_\theta^\epsilon(\epsilon_t \mid \boldsymbol{x}_t) = \sum_{k=1}^K \pi_{\theta,k} \mathcal{N}(\epsilon_t; \boldsymbol{\mu}_{\theta,k}^\epsilon(\boldsymbol{x}_t, t), \boldsymbol{\Sigma}_{\theta,k}^\epsilon(\boldsymbol{x}_t, t)), \tag{10}$$

with component means $\boldsymbol{\mu}_{\theta,k}^\epsilon(\boldsymbol{x}_t, t)$, positive-definite covariance matrices $\boldsymbol{\Sigma}_{\theta,k}^\epsilon(\boldsymbol{x}_t, t)$, and mixture weights $\pi_{\theta,k} \in [0, 1]$ satisfying $\sum_{k=1}^K \pi_{\theta,k} = 1$. From this general specification, we highlight three concrete parametrizations that offer a trade-off between the expressivity of the distribution and the simplicity of the model training:

**Univariate Gaussian:** For $K = 1$ and $\boldsymbol{\Sigma}_\theta(\boldsymbol{x}_t, t) = \mathrm{diag}(\sigma_\theta^2(\boldsymbol{x}_t, t))$, $\sigma_\theta^2(\boldsymbol{x}_t, t) \in \mathbb{R}_{>0}^{d_y}$, we obtain a univariate Gaussian parametrization for each coordinate in the noise space.

**Univariate Gaussian mixture.** Setting $\boldsymbol{\Sigma}_{\theta,k}(\boldsymbol{x}_t, t) = \mathrm{diag}(\sigma_{\theta,k}^2(\boldsymbol{x}_t, t))$ with $\sigma_{\theta,k}^2(\boldsymbol{x}_t, t) \in \mathbb{R}_{>0}^{d_y}$ and $K > 1$ enables multimodal modeling, as Gaussian mixtures can approximate arbitrary continuous densities to arbitrary precision under mild assumptions (Bishop, 1994; Plataniotis & Hatzinakos, 2000), offering significantly greater expressivity at moderate cost.

**Multivariate Gaussian:** Choosing $K = 1$ with full covariance $\boldsymbol{\Sigma}_\theta(\boldsymbol{x}_t, t)$ allows modeling correlations between components of $\epsilon_t$. Since a full parametrization is infeasible for high-dimensional tasks, we consider two approximations: *Cholesky-based* and *low-rank plus diagonal*, which are described in detail in Appendix A.1.

All three model variants are compatible with the scoring rule framework, where we focus on $\lambda = 1$ in Equation (9), corresponding to a strictly proper scoring rule. We utilize the energy score and (Gaussian) kernel score in (7), as they are strictly proper, diffusion compatible (Bortoli et al., 2025), show good performance in practice (Kelen et al., 2024; Bortoli et al., 2025), and admit closed-form expressions, which we derive in Appendix A.2. For completeness, we also compare performance against the log-score in Appendix H.1, which shows significantly worse performance. Beyond the scoring rule formulation, our parametrizations also admit a closed-form backward distribution, which enables straightforward sampling.

**Theorem 1.** *Let $p_\theta^\epsilon(\epsilon_t \mid \boldsymbol{x}_t)$ be given as in Equation* (10). *Then the reverse distribution in Equation* (8) *admits the closed form*

$$p_\theta(\boldsymbol{x}_{t-1} \mid \boldsymbol{x}_t) = \sum_{k=1}^{K} \pi_{\theta,k} \mathcal{N}\left(\boldsymbol{x}_{t-1}; \sqrt{\bar{\alpha}_{t-1}}\hat{\boldsymbol{x}}_0 + \lambda_t \boldsymbol{\mu}_{\theta,k}^\epsilon(\boldsymbol{x}_t, t), \gamma_t^2 \boldsymbol{\Sigma}_{\theta,k}^\epsilon(\boldsymbol{x}_t, t) + \sigma_t^2 \mathbf{I}\right),$$

*where $\hat{\boldsymbol{x}}_0 = \frac{\boldsymbol{x}_t - \sqrt{1-\bar{\alpha}_t}\boldsymbol{\mu}_{\theta,k}^\epsilon(\boldsymbol{x}_t,t)}{\sqrt{\bar{\alpha}_t}}$, $\lambda_t := \sqrt{1 - \bar{\alpha}_{t-1} - \sigma_t^2}$ and $\gamma_t := \lambda_t - \frac{\sqrt{1-\bar{\alpha}_t}}{\sqrt{\alpha_t}}$.*

The proof can be found in Appendix A.3.

Applying this theorem to our univariate Gaussian instantiation and setting $\sigma_t = \tilde{\beta}_t$, i.e. $\eta = 1$, we obtain a DDPM model that outputs the variance for each dimension, similar to what was proposed by Nichol & Dhariwal (2021). Remarkably, in both formulations, the variance is greater than $\tilde{\beta}_t$. However, note that their training objective is based on minimizing a KL divergence and relies on a fixed, state-independent noise variance. Our approach instead uses scoring rules as the loss function, providing a principled way to learn the underlying probability distribution directly from single samples. This not only aligns more naturally with our task but also avoids the restrictive variance assumptions in their method. In Appendix F we delve into this deeper and compare their method with ours.

## 4 NUMERICAL EXPERIMENTS

We now evaluate the proposed methodology across several regression tasks where diffusion models have previously demonstrated competitive or even state-of-the-art performance. Specifically, we consider (i) UCI regression benchmarks (Han et al., 2022), (ii) autoregressive prediction tasks (Rasul et al., 2021; Kohl et al., 2024; Price et al., 2025), and (iii) depth estimation (Ke et al., 2024).

For each task, we use our previously proposed instantiations: *Univariate Gaussian* ($\boldsymbol{\Sigma}_\theta^{\mathrm{diag}}$), *univariate Gaussian mixture* ($\boldsymbol{\Sigma}_\theta^{\mathrm{mix}}$), and *multivariate Gaussian* ($\boldsymbol{\Sigma}_\theta^{\mathrm{mv}}$) with *low-rank plus diagonal* due to its superior performance, see Appendix H. As a baseline, we use established diffusion models ($\boldsymbol{\delta}_\theta$) from the literature. In addition, we compare against the sample-based approach ($\boldsymbol{\epsilon}_t^{\mathrm{ES}}$) of Bortoli et al. (2025). Where possible, we apply minor hyperparameter tuning (Appendix H). Model performance is assessed using a variety of metrics, with particular emphasis on probabilistic calibration and distributional accuracy. Further experimental details are provided in Appendix C, and additional results and visualizations in Appendix D. A comparison with improved DDPM (Nichol & Dhariwal, 2021) is shown in Appendix F.

### 4.1 UCI REGRESSION

We first evaluate our approach on the UCI regression benchmarks (Dua & Graff, 2017), comparing it against CARD (Han et al., 2022), which serves as the baseline $\boldsymbol{\delta}_\theta$. Additional details on CARD are provided in Appendix B, and full experimental settings are given in Appendix C.3.

Table 1 reports test performance in terms of RMSE and continuous ranked probability score (CRPS) across datasets. All distributional variants perform at least on par with the baseline, with the exception of $\boldsymbol{\Sigma}_\theta^{\mathrm{mix}}$ and $\boldsymbol{\epsilon}_t^{\mathrm{ES}}$ on the *yacht* dataset. Still, even here, the simple diagonal Gaussian parameterization $\boldsymbol{\Sigma}_\theta^{\mathrm{diag}}$ consistently improves over $\delta_\theta$, indicating that modest extensions of the noise model can already yield measurable benefits.

Improvements are particularly strong for CRPS, which directly evaluates distributional fit. Among the competitors, the Gaussian mixture parameterization $\boldsymbol{\Sigma}_\theta^{\mathrm{mix}}$ achieves the best overall performance, closely followed by the diagonal Gaussian $\boldsymbol{\Sigma}_\theta^{\mathrm{diag}}$. In fact, both methods rank first and second on average across datasets, clearly outperforming all other approaches. This demonstrates that richer parameterized noise distributions can substantially improve probabilistic calibration without compromising predictive accuracy.

Table 1: RMSE ($\downarrow$) and CRPS ($\downarrow$) on the UCI datasets of the different methods. For better readability, the results for Naval and Kin8nm have been scaled by a factor of 10000 and 100, respectively.

| | RMSE | | | | CRPS | | | |
|---|---|---|---|---|---|---|---|---|
| | $\boldsymbol{\delta}_\theta$ | $\boldsymbol{\Sigma}_\theta^{\text{diag}}$ | $\boldsymbol{\Sigma}_\theta^{\text{mix}}$ | $\boldsymbol{\epsilon}_t^{\text{ES}}$ | $\boldsymbol{\delta}_\theta$ | $\boldsymbol{\Sigma}_\theta^{\text{diag}}$ | $\boldsymbol{\Sigma}_\theta^{\text{mix}}$ | $\boldsymbol{\epsilon}_t^{\text{ES}}$ |
| Concrete | 4.81 | **4.72** | 4.73 | 4.81 | 2.60 | 2.46 | **2.45** | 2.50 |
| Energy | 0.45 | 0.45 | **0.44** | 0.45 | 0.30 | 0.25 | **0.24** | 0.25 |
| Kin8nm | 6.91 | **6.88** | 6.89 | 6.90 | 3.93 | **3.81** | 3.82 | 3.83 |
| Naval | 1.35 | 1.24 | **1.23** | **1.23** | 0.79 | 0.56 | 0.54 | **0.53** |
| Power | 3.86 | 3.64 | **3.59** | 3.78 | 2.04 | 1.84 | **1.81** | 1.93 |
| Protein | 3.76 | **3.71** | 3.72 | 3.76 | 1.71 | 1.66 | **1.65** | 1.69 |
| Wine | 0.67 | 0.67 | **0.66** | **0.66** | 0.34 | 0.34 | 0.34 | **0.33** |
| Yacht | 0.74 | **0.70** | 0.78 | 0.80 | 0.36 | **0.28** | 0.31 | 0.32 |
| Avg. Rank | 3.44 | 1.94 | **1.75** | 2.875 | 3.875 | 2.0625 | **1.625** | 2.4375 |

## 4.2 AUTOREGRESSIVE PREDICTION TASKS

Autoregressive prediction is a key benchmark for probabilistic modeling, as small errors accumulate over rollout length, making uncertainty quantification particularly important. We next evaluate our approach on this setting, where the goal is to generate temporal trajectories $u_1, \ldots, u_S$ from an initial state $u_0$[1]. At each step, the diffusion model $f_\theta$ samples from a conditional distribution $u_s \sim f_\theta(\cdot \mid u_{s-1}, u_{s-2})$, predicting the dynamics $u_s - u_{s-1}$ for three different systems:

**1D PDEs.** We consider the Burgers' and Kuramoto–Sivashinsky (KS) equations,

$$\partial_s u(s,x) + \partial_x \big(u^2(s,x)/2\big) - \nu/\pi \, \partial_{xx} u(s,x) = 0 \qquad \text{(Burgers')}$$

$$\partial_s u(s,x) + u \, \partial_x u(s,x) + \partial_x^2 u(s,x) + \partial_x^4 u(s,x) = 0, \qquad \text{(KS)}$$

with random initial conditions and a spatial resolution of 256.

**Surface temperature prediction.**

As a more complex task, we predict 2-meter surface temperature (T2M) based on several meteorological input variables. We fix a 6-hour forecast horizon, initialized at 00UTC, and train on data from 2011 to 2020, evaluating on the period from 2021 to 2022.

For all tasks, we employ a U-Net diffusion backbone (Song et al., 2021; Karras et al., 2022), adapted to arbitrary input shapes and 1D convolutions. PDE experiments are averaged over five runs, whereas T2M is evaluated once due to computational cost. Full experimental details are provided in Appendix C.4.

Table 2: *Autoregressive prediction results*—showing RMSE ($\downarrow$), CRPS ($\downarrow$) and $\mathcal{C}_{0.95}$ only. The RMSE and CRPS are scaled by the factor 1000 (100) for the Burgers' (KS) equation. Best values per dataset are in bold. A lower average rank indicates better performance.

| | Burgers' | | | KS | | | T2M | | | |
|---|---|---|---|---|---|---|---|---|---|---|
| Model | RMSE | CRPS | $\mathcal{C}_{0.95}$ | RMSE | CRPS | $\mathcal{C}_{0.95}$ | RMSE | CRPS | $\mathcal{C}_{0.95}$ | Avg. Rank |
| $\boldsymbol{\delta}_\theta$ | 0.95 | 0.16 | **0.94** | 0.56 | 0.24 | 0.84 | 0.77 | 0.40 | 0.97 | 3.56 |
| $\boldsymbol{\Sigma}_\theta^{\text{diag}}$ | 0.81 | **0.12** | 1.00 | 0.39 | 0.21 | 1.00 | **0.71** | **0.35** | 0.84 | **2.39** |
| $\boldsymbol{\Sigma}_\theta^{\text{mix}}$ | 0.81 | 0.13 | 1.00 | **0.35** | **0.20** | 1.00 | **0.71** | **0.35** | 0.83 | **2.39** |
| $\boldsymbol{\Sigma}_\theta^{\text{mv}}$ | **0.70** | 0.24 | 1.00 | 0.49 | 0.34 | 0.99 | 0.76 | 0.38 | **0.94** | 2.94 |
| $\boldsymbol{\epsilon}_t^{\text{ES}}$ | 0.81 | 0.14 | 0.99 | 0.59 | 0.34 | **0.98** | - | - | - | 3.08 |

Table 2 summarizes the results. On Burgers', no single variant dominates all metrics, but every probabilistic model improves upon the deterministic baseline $\boldsymbol{\delta}_\theta$ for some metric, with particularly strong gains in the RMSE. For KS, the Gaussian mixture parameterization $\boldsymbol{\Sigma}_\theta^{\text{mix}}$ achieves the best overall performance, clearly outperforming $\boldsymbol{\delta}_\theta$. Across both PDEs, coverage remains close to one,

---

[1]To avoid confusion with diffusion timesteps $t \in 1, \ldots, T$, rollout times are denoted by $s \in 1, \ldots, S$.

indicating slight underconfidence and imperfect marginal calibration; we analyze this behavior further in Appendix E and provide a principled way to further improve performance for our method.

For T2M, all models improve upon $\delta_\theta$, with the strongest improvements by $\Sigma_\theta^{\mathrm{diag}}$ and $\Sigma_\theta^{\mathrm{mix}}$. Figure 2 highlights this performance difference: while both models capture the mean structure, $\delta_\theta$ leaves large residuals in the top-right region of the domain, which $\Sigma_\theta^{\mathrm{mix}}$ successfully resolves.

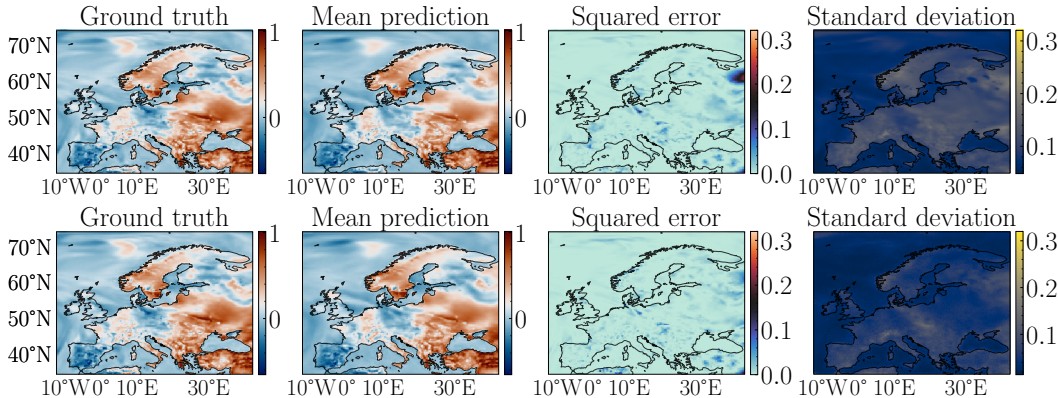

Figure 2: Comparison of the $\delta_\theta$ model (upper) and $\Sigma_\theta^{\mathrm{mix}}$ model (lower) predictions for a selected sample of the dynamics of the 2-meter surface temperature.

One additional advantage of our framework is that, by learning $p_\theta^\epsilon(\epsilon_t \mid x_t)$, it naturally provides a second-order distribution and hence estimates of epistemic (or model-) uncertainty (Hüllermeier & Waegeman, 2021)—something unavailable in standard diffusion models. Figure 3 demonstrates this for the KS equation: While aleatoric uncertainty grows with rollout length, due to the chaotic nature of the governing equation, epistemic uncertainty remains structurally consistent and could potentially be removed with more available data. More details and a theoretical derivation are given in Appendix G.

### 4.3 MONOCULAR DEPTH ESTIMATION

Finally, we evaluate our method on monocular depth estimation, where the goal is to recover a dense depth map from a single RGB image. Recent work has shown that diffusion models achieve state-of-the-art performance on this task (Ke et al., 2024). In particular, Marigold (Ke et al., 2024) adapts Stable Diffusion (Rombach et al., 2022) by finetuning its U-Net, concatenating image and depth embeddings, and modifying the first convolutional layer. We adopt this setup, additionally replacing the final convolution layer with our method-specific alternatives.

Following Marigold, we finetune on the synthetic datasets Hypersim (Roberts et al., 2021) and VKitti2 (Cabon et al., 2020), and evaluate on five real-world benchmarks: NYUv2 (Silberman et al., 2012), ScanNet (Dai et al., 2017), KITTI (Geiger et al., 2012), ETH3D (Schops et al., 2017), and DIODE (Vasiljevic et al., 2019). Dataset splits and further details are provided in Appendix C.5.

Table 3: *Depth estimation results* —showing AbsRel (↓) and CRPS (↓) only. Best values per dataset are in bold. Lower Avg. Rank indicates better overall performance.

| | $\delta_\theta$ | | $\Sigma_\theta^{\mathrm{diag}}$ | | $\Sigma_\theta^{\mathrm{mix}}$ | | $\Sigma_\theta^{\mathrm{mv}}$ | |
|---|---|---|---|---|---|---|---|---|
| Experiment | AbsRel | CRPS | AbsRel | CRPS | AbsRel | CRPS | AbsRel | CRPS |
| NYUv2 | 5.96 | 11.32 | 5.90 | 11.32 | 5.89 | 11.35 | **5.67** | **11.02** |
| KITTI | 10.32 | 142.92 | 10.07 | 138.24 | **9.89** | **137.60** | 10.14 | 142.28 |
| ETH3D | 6.82 | 29.23 | 6.57 | **27.89** | 6.72 | 28.66 | **6.47** | 29.43 |
| ScanNet | 7.10 | 9.19 | 6.84 | 8.86 | 6.96 | 8.99 | **6.79** | **8.85** |
| DIODE | 30.60 | 191.39 | 31.52 | 196.08 | 31.09 | 190.04 | **29.82** | **186.63** |
| Avg. Rank | 3.6 | 3.3 | 2.6 | 2.3 | 2.4 | 2.4 | **1.4** | **2.0** |

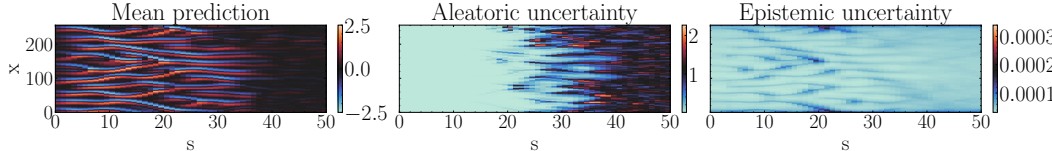

Figure 3: Comparison of aleatoric and epistemic uncertainty estimates for a sample trajectory of the $\Sigma_\theta^{\mathrm{diag}}$ model.

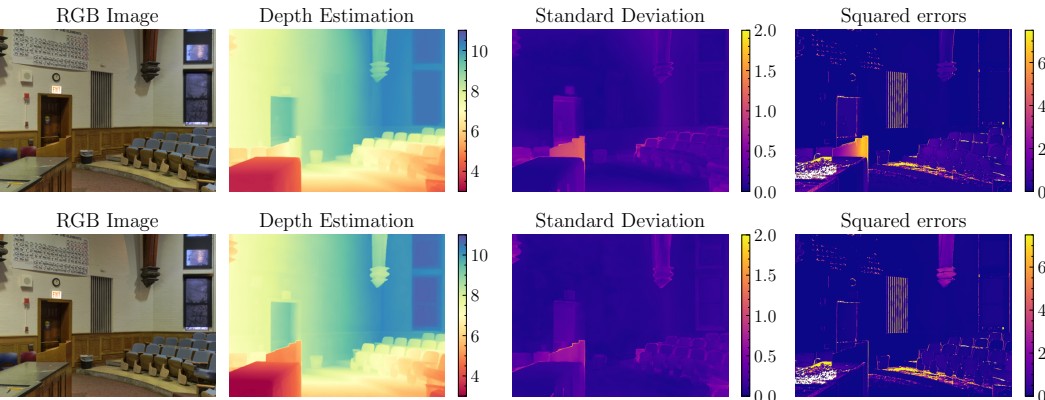

Figure 4: Comparison of the $\delta_\theta$ model (upper) and $\Sigma_\theta^{\mathrm{mv}}$ model (lower) predictions for the depth estimation task on DIODE (Vasiljevic et al., 2019).

Table 3 shows that the multivariate Gaussian parameterization $\Sigma_\theta^{\mathrm{mv}}$ achieves the best overall performance, consistently improving upon the deterministic Marigold baseline $\delta_\theta$ across nearly all datasets and metrics. Figure 4 illustrates the improvement in prediction quality and uncertainty estimation over $\delta_\theta$. Additional experiments, metrics, and further experiment details are reported in Appendix D.

## 5 RELATED WORK

**Approximation of the noise distribution.** This paper proposes to explicitly model the full predictive distribution of $\epsilon_t$ rather than only its conditional mean. Prior work has mainly focused on parameterizing the process variance as a diagonal Gaussian, using a variety of strategies (Ho et al., 2020; Nichol & Dhariwal, 2021; Bao et al., 2022b;a; Ou et al., 2025; Zhang et al., 2024). These methods, however, remain tailored to generative tasks and limited to the Gaussian assumption, since they rely on the standard diffusion loss. Other lines of work challenge the Gaussian assumption in the reverse process and propose to replace it with alternative parametric distributions (Nachmani et al., 2021; Guo et al., 2023; Pandey et al., 2025). Closest to our proposed approach is the recent work by Bortoli et al. (2025), who propose to learn the noise distribution non-parametrically via proper scoring rules to reduce the number of denoising steps in generative modeling. In contrast, we focus on regression tasks, where accurate uncertainty quantification is essential, and propose a parameterized framework for specifying the noise distribution, allowing practitioners to balance expressivity and computational cost.

**Regression with diffusion.** Diffusion-based models have also been adapted for regression tasks, ranging from classical UCI benchmarks (Han et al., 2022), to autoregressive prediction (Kohl et al., 2024; Price et al., 2025; Couairon et al., 2024; Larsson et al., 2025), and image-to-image regression such as depth estimation (Ke et al., 2024). In this context, several works have explored the connection between diffusion models and uncertainty quantification, for example via ensembling or Bayesian approaches (Shu & Farimani, 2024; Chan et al., 2024; Berry et al., 2024). Our work complements these existing approaches by showcasing that our proposed framework can improve performance across a variety of regression tasks, independent of the underlying model architecture, and allows for assessing predictive uncertainty.

## 6 CONCLUSION

We proposed a framework for extending diffusion models and explicitly parameterizing the full distribution of $\epsilon_t$ within the diffusion process. This yields a general approach to probabilistic regression that improves predictive performance across a diverse set of benchmarks. We suggested concrete parameterizations based on multivariate Gaussian mixtures and showed that the corresponding backward sampling distributions admit a closed form. Training with proper scoring rules, our method consistently outperforms deterministic diffusion baselines—state-of-the-art on some datasets—while additionally providing estimates of epistemic uncertainty, a capability unavailable in standard diffusion models. The training time and compute of our parameterizations stay similar to those of regular diffusion models due to our closed-form loss, whereas the sampling-based approach by Bortoli et al. (2025) requires significantly more resources. Additionally, we showed improved performance over their approach on a wide range of considered benchmark tasks and showed that our approach scales to large diffusion backbones such as Stable Diffusion. Furthermore, we demonstrated that existing diffusion architectures can be finetuned within our framework with minimal modification. Our results suggest that the univariate Gaussian method, which is hyperparameter-free, leads to improved performance out of the box and can therefore be utilized as a baseline for new data scenarios. Depending on the spatial structure of the task, the multivariate normal method might lead to additional performance gains, as highlighted for the depth estimation. The univariate Gaussian mixture can improve upon the regular Gaussian method but requires a careful finetuning of the hyperparameters.

**Limitations & future work**

Our approach introduces several design choices regarding the parameterization of the noise distribution. While experiments confirm consistent gains over standard diffusion models, the best-performing variant depends on the task. At present, there is no principled guidance for selecting a parameterization; the choice reflects a trade-off between distributional expressivity and computational cost. Apart from automating the selection process, future work could revolve around improving the different parameterizations and analyzing new ones, such as a multivariate Gaussian mixture. In addition, the combination with different diffusion frameworks and noise schedulers, such as those of Karras et al. (2022) could be developed. Another very interesting avenue for future research would be to dive deeper into the capability of our approach to estimate epistemic uncertainty and to analyze whether the model can successfully detect out-of-distribution shifts.

## REPRODUCIBILITY STATEMENT

In order to ensure reproducibility, we use publicly available datasets for our experiments. The UCI dataset (Dua & Graff, 2017), the PDEBench dataset (Takamoto et al., 2022), the py-pde package (Zwicker, 2020) and the depth estimation datasets (Roberts et al., 2021; Cabon et al., 2020; Silberman et al., 2012; Dai et al., 2017; Geiger et al., 2012; Schops et al., 2017; Vasiljevic et al., 2019).

Our own implementation of the models and experiments for Section 4.1 and 4.2 is available in an anonymous repository (`https://anonymous.4open.science/r/diff_anonymous-FBB6`). The code for running the monocular depth estimation experiments shown in Section 4.3 is available at `https://anonymous.4open.science/r/marigold_anonymous-ECC2`.

## USE OF LARGE LANGUAGE MODELS

Large language models (OpenAI's ChatGPT) were used to assist with improving grammar, style, and phrasing in the final stage of this manuscript.

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

# A APPROXIMATIONS OF THE NOISE DISTRIBUTION

## A.1 APPROXIMATIONS OF THE COVARIANCE MATRIX

As already mentioned, apart from only modeling the diagonal, we consider two efficient ways of parameterizing the full covariance matrix $\Sigma_{\theta,k} \in \mathbb{R}^{D \times D}$.

**Cholesky decomposition** First, we represent $\Sigma_{\theta,k}$ as

$$\Sigma_{\theta,k} = L_{\theta,k} L_{\theta,k}^{\top},$$

where $L_{\theta,k}$ is a lower triangular matrix with positive diagonal. This reduces the number of predicted parameters from $D^2$ to $D(D-1)/2$ while ensuring symmetry. To guarantee positive definiteness, we apply a softplus transformation to the diagonal of $L_{\theta,k}$, normalize the strictly lower part columnwise, and add a small stability constant $\varepsilon = 10e - 6$. In practice, we found this to be sufficient to ensure a positive definite covariance matrix. The Cholesky decomposition offers several advantages (Muschinski et al., 2024):

- Sampling is efficient and requires only a matrix-vector product.
- The determinant can be computed as $\det(\Sigma) = (\prod_{i=1}^{D} L_{ii})^2$.
- Although computing the full inverse $\Sigma^{-1}$ has cost $\mathcal{O}(D^3)$, inverse-vector products $\Sigma^{-1}v$ require only $\mathcal{O}(D^2)$.

Nevertheless, for very high-dimensional data, these computations may still be too expensive.

**Low-rank + diagonal** As a more efficient alternative, we consider the low-rank + diagonal structure (Rezende et al., 2014)

$$\Sigma_{\theta,k} = U_{\theta,k} U_{\theta,k}^{\top} + D_{\theta,k},$$

where $U_{\theta,k} \in \mathbb{R}^{D \times r}$, with $\mathrm{rank}(U_{\theta,k}) = r \ll D$ and $D_{\theta,k} \in \mathbb{R}^{D \times D}$ is a (positive) diagonal matrix. Here, $U_{\theta,k}$ and the diagonal $D_{\theta,k}$ are both predicted by the neural network. Using the matrix determinant lemma, we obtain $\det(\Sigma) = \det(D_{\theta,k}) \det(\mathbf{I}_r + U_{\theta,k}^{\top} D_{\theta,k}^{-1} U_{\theta,k})$, which can be computed in $\mathcal{O}(Dr^2 + r^3)$. Similarly, inverse-vector products $\Sigma^{-1}v$ cost $\mathcal{O}(Dr^2 + r^3 + Dr)$, while sampling is also efficient. When $r \ll D$, this parameterization is nearly as efficient as the diagonal case, though less expressive than the full Cholesky form.

Table 4 summarizes the computational complexity of common operations. The low-rank plus diagonal scales favorably when $r \ll D$, while Cholesky remains more costly. Figure 5 illustrates the different correlation structures for each method during different diffusion timesteps for a selected sample of the Kuramoto–Sivashinsky equation.

Table 4: Computational complexity for common operations on covariance matrices $\Sigma$ under different factorizations. $D$ is the data dimension, $r$ is the rank of $U$ in the low-rank+diagonal case.

| Method | Sampling from $\mathcal{N}(0, \Sigma)$ | $\det(\Sigma)$ | $\Sigma^{-1}$ | $\Sigma^{-1}v$ |
|---|---|---|---|---|
| Cholesky | $\mathcal{O}(D^2)$ | $\mathcal{O}(D)$ | $\mathcal{O}(D^3)$ | $\mathcal{O}(D^2)$ |
| Low-rank + diag | $\mathcal{O}(Dr)$ | $\mathcal{O}(Dr^2 + r^3)$ | $\mathcal{O}(D^2r)$ | $\mathcal{O}(Dr^2 + r^3)$ |
| Diagonal | $\mathcal{O}(D)$ | $\mathcal{O}(D)$ | $\mathcal{O}(D)$ | $\mathcal{O}(D)$ |

In practice, we found that even with these efficient approximations, training sometimes showed instabilities, especially for high-dimensional data. This is especially evident for specific hyperparameters, as can be seen in Table 16 in Appendix H. In addition, using the full Cholesky parameterization, even when converging, often led to significantly worse performance than the low-rank-plus-diagonal method, likely due to the substantially larger parameterization, and the resulting trade-offs discussed above.

## A.2 CLOSED-FORM EXPRESSIONS FOR SCORING RULES

In this section, we provide closed-form expressions for different scoring rules for the different instantiations of our proposed predictive distribution.

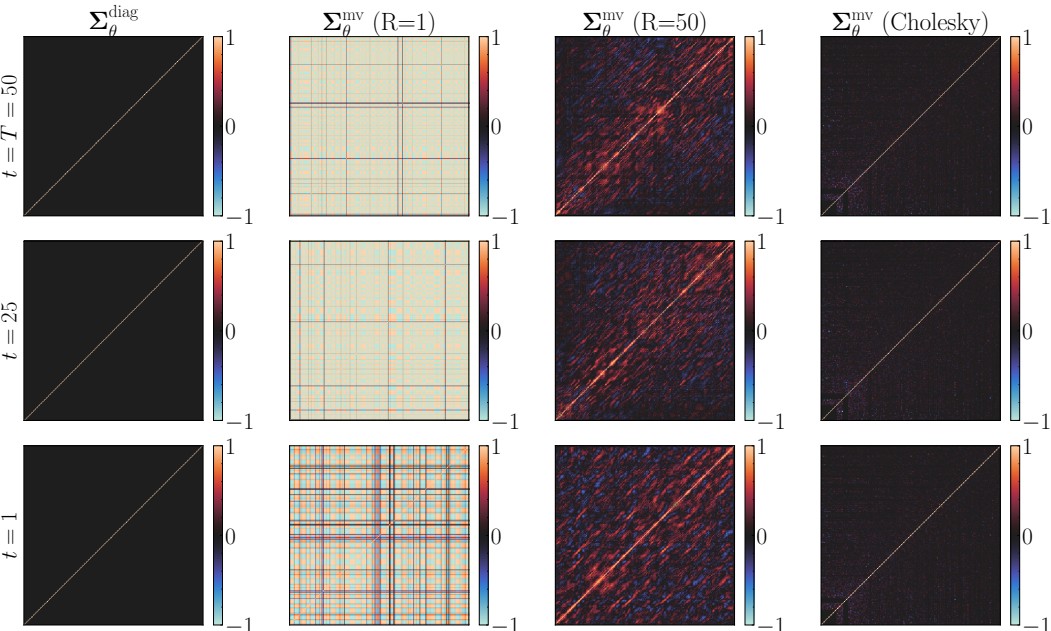

Figure 5: Correlation matrices for different covariance parameterizations across several time steps $t$ for a selected sample of the Kuramoto–Sivashinsky equation.

**Univariate Gaussians**  For univariate Gaussians, the energy score reduces to the continuous ranked probability score (CRPS), which admits a closed-form expression (Gneiting & Raftery, 2007):

$$\text{CRPS}(\mathcal{N}(\mu, \sigma^2), y) = \sigma\left(\frac{y-\mu}{\sigma}\left(\Phi\left(\frac{y-\mu}{\sigma}\right) - 1\right) + 2\varphi\left(\frac{y-\mu}{\sigma}\right) - \frac{1}{\sqrt{\pi}}\right), \quad (11)$$

where $\Phi$ and $\varphi$ denote the CDF and PDF of a standard normal distribution, respectively. For a Gaussian kernel with bandwidth $\gamma$, the corresponding kernel score admits a closed-form solution as

$$S_k(\mathcal{N}(\mu.\sigma^2), y) = -\frac{1}{\sqrt{1 + 2\sigma^2/\gamma^2}}\exp\left(-\frac{(y-\mu)^2}{\gamma^2 + 2\sigma^2}\right) + \frac{1}{2\sqrt{1 + 4\sigma^2/\gamma^2}} + \frac{1}{2}. \quad (12)$$

*Proof.* The proof follows a similar direction to Rustamov (2020). By definition, the Gaussian kernel score is given as

$$S_k(\mathcal{N}(\mu, \sigma^2), y) = \frac{1}{2}\mathbb{E}_{X,X'\sim\mathcal{N}(\mu,\sigma^2)}[k(X,X')] - \mathbb{E}_{X\sim\mathcal{N}(\mu,\sigma^2)}[k(X,y)] + \frac{1}{2}k(y,y),$$

with $k = \exp(-\|x - x'\|^2/2\gamma^2)$.

We start with the second term, which we can write as

$$\mathbb{E}_{\mathcal{N}(\mu,\sigma^2)}[k(X,y)] = \int_{\mathbb{R}} e^{-(x-y)^2/\gamma^2}(2\pi\sigma^2)^{-1/2}e^{-(x-\mu)^2/(2\sigma^2)}\,dx$$

$$= (\pi\gamma^2)^{1/2}\int_{\mathbb{R}}(\pi\gamma^2)^{-1/2}e^{-(x-y)^2/\gamma^2}(2\pi\sigma^2)^{-1/2}e^{(x-\mu)^2/(2\sigma^2)}\,dx.$$

Since $(x - y)^2 = (y - x)^2$, this can be expressed as the convolution of two densities

$$\mathbb{E}_{\mathcal{N}(\mu,\sigma^2)}[k(X,y)] = (\pi\gamma^2)^{1/2}(f_{\mu,\sigma^2} * f_{0,\gamma^2/2})(y),$$

where $f_{\mu,\sigma^2}$ denotes the pdf of a Gaussian. By the convolution theorem of Gaussian random variables, we know that $(f_{\mu,\sigma^2} * f_{0,\gamma^2/2})(y) = f_{\mu,\sigma^2+\gamma^2/2}(y)$. Therefore, we obtain

$$\mathbb{E}_{\mathcal{N}(\mu,\sigma^2)}[k(X,y)] = (\pi\gamma^2)^{1/2}(\pi(\gamma^2 + 2\sigma^2))^{-1/2}e^{-(y-\mu)^2/(2\sigma^2+\gamma^2)}$$

$$= \frac{1}{\sqrt{1 + 2\sigma^2/\gamma^2}}\exp\left(-\frac{(y-\mu)^2}{\gamma^2 + 2\sigma^2}\right).$$

The first term can be obtained by recognizing $X - X' \sim \mathcal{N}(0, 2\sigma^2)$ and following the same steps as above. $\qquad\square$

**Univariate Gaussian mixture**  Let $F_K(x) = \sum_{i=1}^K w_i \mathcal{N}(x; \mu_i, \sigma_i^2)$ denote a $K$-component normal mixture distribution. Then, a closed-form expression for the CRPS is given as

$$\text{CRPS}(F_K, y) = \sum_{i=1}^K w_i A(y - \mu_i, \sigma_i^2) - \frac{1}{2} \sum_{i=1}^K \sum_{j=1}^K w_j w_i A(\mu_i - \mu_j, \sigma_i^2 + \sigma_j^2), \tag{13}$$

where $A(\mu, \sigma^2) = \mu \left(2\Phi\left(\frac{\mu}{\sigma}\right) - 1\right) + 2\sigma\varphi\left(\frac{\mu}{\sigma^2}\right)$ (Grimit et al., 2006).

**Multivariate Gaussian**  In the case of the multivariate Gaussian $\mathcal{N}(\boldsymbol{\mu}, \boldsymbol{\Sigma})$, the energy score does not admit an analytical expression (Székely & Rizzo, 2005). However, the kernel score with a Gaussian kernel does admit a closed-form, given as

$$S_k(\mathcal{N}(\boldsymbol{\mu}, \boldsymbol{\Sigma}), \boldsymbol{y}) = -\frac{1}{\sqrt{\det\left(I + \frac{2}{\gamma^2}\boldsymbol{\Sigma}\right)}} \exp\left(-\frac{1}{\gamma^2}(\boldsymbol{y} - \boldsymbol{\mu})^\top \left(I + \frac{2}{\gamma^2}\boldsymbol{\Sigma}\right)^{-1} (\boldsymbol{y} - \boldsymbol{\mu})\right)$$
$$+ \frac{1}{2\sqrt{\det\left(I + \frac{4}{\gamma^2}\boldsymbol{\Sigma}\right)}} + \frac{1}{2}, \tag{14}$$

where $\gamma$ is the bandwidth of the kernel.

*Proof.* Let $X, X' \sim \mathcal{N}(\boldsymbol{\mu}, \boldsymbol{\Sigma})$ and $Z := X - \boldsymbol{y} \sim \mathbb{P}_Z := \mathcal{N}(\boldsymbol{\mu} - \boldsymbol{y}, \boldsymbol{\Sigma})$. Then $\|Z\|^2 = Z^\top Z = Z^\top A Z$ with $A = \boldsymbol{I}_d$ follows a generalized chi-squared distribution (Das & Geisler, 2024), i.e., $Z^\top A Z \sim \tilde{\chi}(\boldsymbol{I}_d, \boldsymbol{0}, 0, \boldsymbol{\mu} - \boldsymbol{y}, \boldsymbol{\Sigma})$. Then, we can express the kernel score in terms of the corresponding moment-generating function, via

$$\mathbb{E}_{X \sim \mathcal{N}(\boldsymbol{\mu}, \boldsymbol{\Sigma})}[k(X, \boldsymbol{y})] = \mathbb{E}_{\mathbb{P}_Z}\left[\exp\left(-\frac{Z^\top Z}{\gamma^2}\right)\right] = M_{\tilde{\chi}(\boldsymbol{I}_d, \boldsymbol{0}, 0, \boldsymbol{\mu} - \boldsymbol{y}, \boldsymbol{\Sigma})}\left(-\frac{1}{\gamma^2}\right),$$

with $t = -\frac{1}{\gamma^2}$. In the case of the generalized chi-squared distribution, the moment-generating function is given as

$$M_{\tilde{\chi}(\boldsymbol{I}_d, \boldsymbol{0}, 0, \boldsymbol{\mu} - \boldsymbol{y}, \boldsymbol{\Sigma})}(t) = \frac{1}{\sqrt{\det(\boldsymbol{I}_d - 2t\boldsymbol{\Sigma})}} \exp\left(t(\boldsymbol{\mu} - \boldsymbol{y})^\top (\boldsymbol{I}_d - 2t\boldsymbol{\Sigma})^{-1} (\boldsymbol{\mu} - \boldsymbol{y})\right),$$

from which we obtain

$$\mathbb{E}_{X \sim \mathcal{N}(\boldsymbol{\mu}, \boldsymbol{\Sigma})}[k(X, \boldsymbol{y})] = \frac{1}{\sqrt{\det\left(\boldsymbol{I}_d + \frac{2}{\gamma^2}\boldsymbol{\Sigma}\right)}} \exp\left(-\frac{1}{\gamma^2}(\boldsymbol{\mu} - \boldsymbol{y})^\top \left(\boldsymbol{I}_d + \frac{2}{\gamma^2}\boldsymbol{\Sigma}\right)^{-1} (\boldsymbol{\mu} - \boldsymbol{y})\right).$$

Repeating these steps for $Z := X - X' \sim \mathbb{P}_Z := \mathcal{N}(0, 2\boldsymbol{\Sigma})$ gives the expression for $\mathbb{E}_{X, X' \sim \mathcal{N}(\boldsymbol{\mu}, \boldsymbol{\Sigma})}[k(X, X')]$ and therefore the full expression for $S_k(\mathcal{N}(\boldsymbol{\mu}, \boldsymbol{\Sigma}), \boldsymbol{y})$. $\qquad\square$

### A.3 PROOF OF THEOREM 1

We start by stating a theorem concerning the distribution of a Gaussian random variable whose mean is the linear combination of another Gaussian distribution, which will serve as an important tool in many of the upcoming proofs.

**Theorem 2.** *Let $p(x) = \mathcal{N}(\mu_x, \Sigma_x)$, with mean $\mu_x \in \mathbb{R}^d$ and positive semi-definite covariance matrix $\Sigma_x \in \mathbb{R}^{d \times d}$. Further, let $y$ be a random vector with conditional distribution $p(y \mid x) = \mathcal{N}(Ax + b, \Sigma_y)$, for $A \in \mathbb{R}^{d \times d}, b \in \mathbb{R}^d$. Then $p(y) = \mathcal{N}(A\mu_x + b, A\Sigma_x A^T + \Sigma_y)$.*

For a proof, compare Theorem 4.4.1 in Murphy (2014), "Machine Learning: A Probabilistic Perspective". We now give the proof of Theorem 1.

*Proof.* First, we obtain an equivalent expression for (2) which is now conditioned on $\epsilon_t$ instead of $x_0$. Since $\epsilon_t = \frac{1}{\sqrt{1-\bar{\alpha}_t}}\left(X_t - \sqrt{\bar{\alpha}}X_0\right)$, we can write

$$\mu(x_0, x_t) = \sqrt{\bar{\alpha}_{t-1}}x_0 + \sqrt{1 - \bar{\alpha}_{t-1} - \sigma_t^2} \cdot \frac{x_t - \sqrt{\bar{\alpha}_t}x_0}{\sqrt{1 - \bar{\alpha}_t}}$$

$$= \sqrt{\bar{\alpha}_{t-1}} \cdot \frac{x_t - \sqrt{1 - \bar{\alpha}_t}\epsilon_t}{\sqrt{\bar{\alpha}_t}} + \sqrt{1 - \bar{\alpha}_{t-1} - \sigma_t^2}\epsilon_t$$

$$=: \tilde{\mu}(\epsilon_t, x_t) =: a(x_t, t)\epsilon_t + b(x_t, t)$$

where we introduce the functions $a(x_t, t)$ and $b(x_t, t)$ for notational simplicity. We obtain $p(x_{t-1} \mid x_t, \epsilon_t) = \mathcal{N}(\tilde{\mu}(\epsilon_t, x_t), \sigma_t^2\mathbf{I})$. Now,

$$p_\theta^{(t)}(x_{t-1} \mid x_t) = \int q(x_{t-1} \mid x_t, \epsilon_t)p_\theta(\epsilon_t \mid x_t)d\epsilon_t.$$

$$= \int \mathcal{N}(x_{t-1}; a(x_t, t)\epsilon_t + b(x_t, t), \sigma_t^2\mathbf{I})\left(\sum_{k=1}^K \pi_\theta^k \mathcal{N}(\epsilon_t; \mu_{\theta,k}^\epsilon(x_t), \Sigma_{\theta,k}^\epsilon(x_t))\right)d\epsilon_t$$

$$= \sum_{k=1}^K \pi_\theta^k \int \mathcal{N}(x_{t-1}; a(x_t, t)\epsilon_t + b(x_t, t), \sigma_t^2\mathbf{I})\mathcal{N}(\epsilon_t; \mu_{\theta,k}^\epsilon(x_t), \Sigma_{\theta,k}^\epsilon(x_t))d\epsilon_t$$

$$= \sum_{k=1}^K \pi_\theta^k \mathcal{N}(x_{t-1}; a(x_t, t)\mu_{\theta,k}^\epsilon(x_t) + b(x_t, t), a(x_t, t)^2\Sigma_{\theta,k}^\epsilon(x_t) + \sigma_t^2\mathbf{I})$$

$$= \sum_{k=1}^K \pi_\theta^k \mathcal{N}\left(x_{t-1}; \sqrt{\bar{\alpha}_{t-1}}\hat{x}_0 + \sqrt{1 - \bar{\alpha}_{t-1} - \sigma_t^2}\mu_{\theta,k}^\epsilon(x_t), \gamma_t^2\Sigma_{\theta,k}^\epsilon(x_t) + \sigma_t^2\mathbf{I}\right),$$

where the fourth equality follows by theorem 2 and the last by plugging in the constants. $\square$

## B    DETAILS ON CARD

CARD was introduced by Han et al. (2022) as a framework for classification and regression tasks using diffusion. They assume to have a pre-trained regressor $f_\phi : \mathcal{C} \to \mathcal{Y}$ available, which is trained on $\mathcal{D}$ and approximates the conditional expectation $\mathbb{E}[y \mid c]$ given covariates $c \in \mathcal{C}$. Its core idea is, that a diffusion process is used to interpolate between the noise distribution $p(y_t \mid c) = \mathcal{N}(f_\phi(c), \mathbf{I})$ and the data distribution $p(y_0 \mid c)$. We will provide a formulation of CARD embedded in the DDIM framework introduced earlier. To this end, we set the mean in (2) to

$$\mu(y_0, y_t) = \sqrt{\bar{\alpha}_{t-1}}y_0 + (1 - \sqrt{\bar{\alpha}_{t-1}}) f_\phi(c) + \sqrt{1 - \bar{\alpha}_{t-1} - \sigma_t^2} \cdot \frac{y_t - (\sqrt{\bar{\alpha}_t}y_0 + (1 - \sqrt{\bar{\alpha}_t}) f_\phi(c))}{\sqrt{1 - \bar{\alpha}_t}} \tag{15}$$

which yields a modified version of (3), namely

$$p(y_t \mid y_0, f_\phi(c)) = \mathcal{N}\left(\sqrt{\bar{\alpha}_t}y_0 + (1 - \sqrt{\bar{\alpha}_t}f_\phi(c))\right), (1 - \bar{\alpha}_t)\mathbf{I}). \tag{16}$$

We will prove this and the fact that we can recover the original CARD method by an appropriate choice of $\{\sigma_t\}_{t=1}^T$.

The original CARD algorithm for sampling $y_{t-1}$ given $y_t$ is specified as follows

$$\begin{cases} \hat{y}_0 = \frac{1}{\sqrt{\bar{\alpha}_t}}\left(y_t - (1 - \sqrt{\bar{\alpha}_t}) f_\theta(x) - \sqrt{1 - \bar{\alpha}_t}\epsilon_\theta\right) \\ y_{t-1} = \gamma_0\hat{y}_0 + \gamma_1 y_t + \gamma_2 f_\phi(x) + \sqrt{\tilde{\beta}}z, \quad z \sim \mathcal{N}(0, \mathbf{I}) \end{cases} \tag{17}$$

where

$$\gamma_0 := \frac{\beta_t\sqrt{\bar{\alpha}_{t-1}}}{1 - \bar{\alpha}_t}, \quad \gamma_1 := \frac{(1 - \bar{\alpha}_{t-1})\sqrt{\alpha_t}}{1 - \bar{\alpha}_t}, \quad \gamma_2 := \left(1 + \frac{(\sqrt{\bar{\alpha}_t} - 1)\left(\sqrt{\alpha_t} + \sqrt{\bar{\alpha}_{t-1}}\right)}{1 - \bar{\alpha}_t}\right). \tag{18}$$

We have

$$y_{t-1} = \gamma_0 \frac{1}{\sqrt{\bar{\alpha}_t}} \left( y_t - \left(1 - \sqrt{\bar{\alpha}_t}\right) f_\theta(x) - \sqrt{1 - \bar{\alpha}_t}\epsilon_\theta \right) + \gamma_1 y_t + \gamma_2 f_\phi(x) + \sqrt{\tilde{\beta}}z \qquad (19)$$

$$= \underbrace{\left( \frac{\gamma_0}{\sqrt{\bar{\alpha}_t}} + \gamma_1 \right)}_{=:A} y_t + \underbrace{\left( -\frac{\gamma_0 \left(1 - \sqrt{\bar{\alpha}_t}\right)}{\sqrt{\bar{\alpha}_t}} + \gamma_2 \right)}_{=:B} f_\phi(x) - \underbrace{\frac{\gamma_0 \sqrt{1 - \bar{\alpha}_t}}{\sqrt{\bar{\alpha}_t}}}_{=:C} \epsilon_\theta + \sqrt{\tilde{\beta}}z \qquad (20)$$

and calculate

$$A = \frac{\beta_t \sqrt{\bar{\alpha}_{t-1}}}{(1 - \bar{\alpha}_t) \sqrt{\bar{\alpha}_t}} + \frac{(1 - \bar{\alpha}_{t-1}) \sqrt{\alpha_t}}{1 - \bar{\alpha}_t} = \frac{\beta_t}{(1 - \bar{\alpha}_t) \sqrt{\alpha_t}} + \frac{(1 - \bar{\alpha}_{t-1}) \alpha_t}{(1 - \bar{\alpha}_t) \sqrt{\alpha_t}} \qquad (21)$$

$$= \frac{1 - \alpha_t + \alpha_t - \bar{\alpha}_t}{(1 - \bar{\alpha}_t) \sqrt{\alpha_t}} = \frac{1}{\sqrt{\alpha_t}} \qquad (22)$$

as well as

$$B = -\frac{\beta_t \sqrt{\bar{\alpha}_{t-1}} \left(1 - \sqrt{\bar{\alpha}_t}\right)}{\sqrt{\bar{\alpha}_t} \left(1 - \bar{\alpha}_t\right)} + 1 + \frac{\left(\sqrt{\bar{\alpha}_t} - 1\right) \left(\sqrt{\alpha_t} + \sqrt{\bar{\alpha}_{t-1}}\right)}{1 - \bar{\alpha}_t} \qquad (23)$$

$$= \frac{-(1 - \alpha_t) \left(1 - \sqrt{\bar{\alpha}_t}\right) + \left(\sqrt{\bar{\alpha}_t} - 1\right) \left(\sqrt{\alpha_t} + \sqrt{\bar{\alpha}_{t-1}}\right) \sqrt{\alpha_t}}{\sqrt{\alpha_t} \left(1 - \bar{\alpha}_t\right)} + 1 \qquad (24)$$

$$= \frac{-1 + \alpha_t + \sqrt{\bar{\alpha}_t} - \alpha_t \sqrt{\bar{\alpha}_t} + \left(\sqrt{\bar{\alpha}_t} - 1\right) \left(\alpha_t + \sqrt{\bar{\alpha}_t}\right)}{\sqrt{\alpha_t} \left(1 - \bar{\alpha}_t\right)} + 1 \qquad (25)$$

$$= \frac{-1 + \alpha_t + \sqrt{\bar{\alpha}_t} - \alpha_t \sqrt{\bar{\alpha}_t} + \sqrt{\bar{\alpha}_t}\alpha_t + \bar{\alpha}_t - \alpha_t - \sqrt{\bar{\alpha}_t}}{\sqrt{\alpha_t} \left(1 - \bar{\alpha}_t\right)} + 1 \qquad (26)$$

$$= \frac{\bar{\alpha}_t - 1}{\sqrt{\alpha_t} \left(1 - \bar{\alpha}_t\right)} + 1 \qquad (27)$$

$$= -\frac{1}{\sqrt{\alpha_t}} \left(1 - \sqrt{\alpha_t}\right) \qquad (28)$$

and lastly

$$C = \frac{\beta_t \sqrt{\bar{\alpha}_{t-1}} \sqrt{1 - \bar{\alpha}_t}}{(1 - \bar{\alpha}_t) \sqrt{\bar{\alpha}_t}} = \frac{1}{\sqrt{\alpha_t}} \cdot \frac{1 - \alpha_t}{\sqrt{1 - \bar{\alpha}_t}}. \qquad (29)$$

In total, we obtain

$$y_{t-1} = \frac{1}{\sqrt{\alpha_t}} \left( y_t - \frac{1 - \alpha_t}{\sqrt{1 - \bar{\alpha}_t}}\epsilon_\theta - \left(1 - \sqrt{\alpha_t}\right) f_\phi(x) \right) \qquad (30)$$

To finish the argument we substitute $\sigma = \sqrt{\frac{1-\bar{\alpha}_{t-1}}{1-\bar{\alpha}_t}\beta_t}$ in (15) and obtain

$$\sqrt{\bar{\alpha}_{t-1}}\frac{y_t - \left(\sqrt{1-\bar{\alpha}_t}\epsilon_\theta + \left(1-\sqrt{\bar{\alpha}_t}\right)f_\phi(x)\right)}{\sqrt{\bar{\alpha}_t}} + \sqrt{1 - \bar{\alpha}_{t-1} - \frac{1-\bar{\alpha}_{t-1}}{1-\bar{\alpha}_t}(1-\alpha_t)} \cdot \epsilon_\theta + \tag{31}$$

$$\left(1 - \sqrt{\bar{\alpha}_{t-1}}\right)f_\phi(x) \tag{32}$$

$$= \frac{1}{\sqrt{\alpha_t}}\left(y_t - \sqrt{1-\bar{\alpha}_t}\epsilon_\theta - \left(1-\sqrt{\bar{\alpha}_t}\right)f_\phi(x)\right) + \tag{33}$$

$$\sqrt{\frac{1 - \bar{\alpha}_{t-1} - \bar{\alpha}_t + \bar{\alpha}_{t-1}\bar{\alpha}_t - 1 + \alpha_t + \bar{\alpha}_{t-1} - \bar{\alpha}_t}{1 - \bar{\alpha}_t}} \cdot \epsilon_\theta + \left(1 - \sqrt{\bar{\alpha}_{t-1}}\right)f_\phi(x) \tag{34}$$

$$= \frac{1}{\sqrt{\alpha_t}}\left(y_t - \sqrt{1-\bar{\alpha}_t}\epsilon_\theta - \left(1-\sqrt{\bar{\alpha}_t}\right)f_\phi(x)\right) + \sqrt{\frac{-\bar{\alpha}_t + \bar{\alpha}_{t-1}\bar{\alpha}_t + \alpha_t - \bar{\alpha}_t}{1 - \bar{\alpha}_t}} \cdot \epsilon_\theta + \tag{35}$$

$$\left(1 - \sqrt{\bar{\alpha}_{t-1}}\right)f_\phi(x) \tag{36}$$

$$= \frac{1}{\sqrt{\alpha_t}}\left(y_t - \sqrt{1-\bar{\alpha}_t}\epsilon_\theta - \left(1-\sqrt{\bar{\alpha}_t}\right)f_\phi(x)\right) + \sqrt{\frac{(\alpha_t - \bar{\alpha}_t)\left(1 - \bar{\alpha}_{t-1}\right)}{1 - \bar{\alpha}_t}} \cdot \epsilon_\theta + \tag{37}$$

$$\left(1 - \sqrt{\bar{\alpha}_{t-1}}\right)f_\phi(x) \tag{38}$$

$$= \frac{1}{\sqrt{\alpha_t}}\left(y_t - \sqrt{1-\bar{\alpha}_t}\epsilon_\theta - \left(1-\sqrt{\bar{\alpha}_t}\right)f_\phi(x)\right) + \sqrt{\frac{(\alpha_t - \bar{\alpha}_t)^2}{\alpha_t\left(1 - \bar{\alpha}_t\right)}} \cdot \epsilon_\theta + \left(1 - \sqrt{\bar{\alpha}_{t-1}}\right)f_\phi(x) \tag{39}$$

$$= \frac{1}{\sqrt{\alpha_t}}\left(y_t + \left(\frac{(\alpha_t - \bar{\alpha}_t)}{\sqrt{1-\bar{\alpha}_t}} - \sqrt{1-\bar{\alpha}_t}\right)\epsilon_\theta - \left(1 - \sqrt{\bar{\alpha}_t} + \sqrt{\alpha_t} - \sqrt{\alpha_t\bar{\alpha}_{t-1}}\right)f_\phi(x)\right) \tag{40}$$

$$= \frac{1}{\sqrt{\alpha_t}}\left(y_t - \frac{1-\alpha_t}{\sqrt{1-\bar{\alpha}_t}}\epsilon_\theta - \left(1 - \sqrt{\bar{\alpha}_t}\right)f_\phi(x)\right) \tag{41}$$

We also want to show that (16) holds, which we will do by an induction argument, similar in style to Song et al. (2021).

*Proof.* Assume that

$$p(y_t \mid y_0, f_\phi(x)) = \mathcal{N}\left(\sqrt{\bar{\alpha}_t}y_0 + \left(1 - \sqrt{\bar{\alpha}_t}f_\phi(x)\right), (1 - \bar{\alpha}_t)\mathbf{I}\right). \tag{42}$$

holds for $t = 1, \ldots, T$. We will show that it is also satisfied for $t-1$. Since it is true by assumption for $t = T$, this would finish the proof.
We have

$$p(y_{t-1} \mid y_0, f_\phi(x)) = \int_{\mathbb{R}^d} p(y_{t-1} \mid y_0, y_t, f_\phi(x))p(y_t \mid y_0, f_\phi(x))dy_t, \tag{43}$$

where

$$p(y_{t-1} \mid y_0, y_t, f_\phi(x)) = \mathcal{N}(\sqrt{\bar{\alpha}_{t-1}}y_0 + \sqrt{1 - \bar{\alpha}_{t-1} - \sigma_t^2}\cdot\frac{y_t - \left(\sqrt{\bar{\alpha}_t}y_0 + \left(1-\sqrt{\bar{\alpha}_t}\right)f_\phi(x)\right)}{\sqrt{1-\bar{\alpha}_t}}, \sigma_t^2\mathbf{I}), \tag{44}$$

corresponds to (2) just using the mean defined in (15). Now,

$$p(y_t \mid y_0, f_\phi(x)) = \mathcal{N}\left(\sqrt{\bar{\alpha}_t}y_0 + \left(1 - \sqrt{\bar{\alpha}_t}f_\phi(x)\right), (1 - \bar{\alpha}_t)\mathbf{I}\right) \tag{45}$$

holds because of the induction hypothesis. By theorem 2, we have that $p(y_{t-1} \mid y_0, y_t, f_\phi(x))$ is Gaussian with mean $\mu_{t-1}$ and covariance matrix $\Sigma_{t-1}$, where

$$\mu_{t-1} = \sqrt{\bar{\alpha}_{t-1}}y_0 + \left(1 - \sqrt{\bar{\alpha}_{t-1}}\right)f_\phi(x) + \tag{46}$$

$$\sqrt{1 - \bar{\alpha}_{t-1} - \sigma_t^2} \cdot \frac{\sqrt{\bar{\alpha}_t}y_0 + \left(1-\sqrt{\bar{\alpha}_t}f_\phi(x)\right) - \left(\sqrt{\bar{\alpha}_t}y_0 + \left(1-\sqrt{\bar{\alpha}_t}\right)f_\phi(x)\right)}{\sqrt{1-\bar{\alpha}_t}} \tag{47}$$

$$= \sqrt{\bar{\alpha}_{t-1}}y_0 + \left(1 - \sqrt{\bar{\alpha}_{t-1}}\right)f_\phi(x) \tag{48}$$

and

$$\Sigma_{t-1} = \sigma_t^2\mathbf{I} + \frac{1 - \bar{\alpha}_{t-1} - \sigma_t^2}{1 - \bar{\alpha}_t}(1 - \bar{\alpha}_t)\mathbf{I} = (1 - \bar{\alpha}_{t-1})\mathbf{I} \tag{49}$$

which finishes the proof. $\square$

## C EXPERIMENT DETAILS

### C.1 EVALUATION METRICS

Consider the true target $\boldsymbol{y} \in \mathcal{Y} \subseteq \mathbb{R}^{d_y}$, the corresponding marginals $y^k$, $k = 1, \ldots, d_y$, the true predictive distribution $p_{\mathcal{Y}}(\cdot \mid \boldsymbol{c})$ and the learned approximate predictive distribution $p_\theta(\cdot \mid \boldsymbol{c})$. Furthermore, define $\hat{\boldsymbol{y}}_i \sim p_\theta(\cdot \mid \boldsymbol{c})$, $i = 1, \ldots, M$ and the empirical predictive distribution $p_\theta^M :=$ $\{\hat{\boldsymbol{y}}_m\}_{m=1}^M$. Finally, let $\bar{\boldsymbol{y}} := \frac{1}{M} \sum_{m=1}^M \hat{\boldsymbol{y}}_m$ denote the empirical mean, $\hat{\sigma}_k^2 := \frac{1}{M-1} \sum_{m=1}^M (\hat{\boldsymbol{y}}_m - \bar{\boldsymbol{y}})^2$ denote the empirical variance of the $k^{th}$ marginal and $(\hat{q}_\theta^\alpha)^k$ denote the empirical quantiles of the $k^{th}$ marginal of $p_\theta^M$ at the level $\alpha$. We use the following evaluation metrics:

$$\mathrm{RMSE}(p_\theta^M, \boldsymbol{y}) := \|\bar{\boldsymbol{y}} - \boldsymbol{y}\|_2, \tag{50}$$

$$\mathrm{ES}(p_\theta^M, \boldsymbol{y}) := \frac{1}{M} \sum_{m=1}^M \|\hat{\boldsymbol{y}}_i - \boldsymbol{y}\|_2 - \frac{1}{2M(M-1)} \sum_{\substack{m,h=1 \\ m \neq h}}^M \|\hat{\boldsymbol{y}}_m - \hat{\boldsymbol{y}}_h\|_2, \tag{51}$$

$$\mathrm{CRPS}(p_\theta^M, \boldsymbol{y}) := \frac{1}{d_y} \sum_{k=1}^{d_y} \left( \frac{1}{M} \sum_{m=1}^M |\hat{y}_i^k - y^k| - \frac{1}{2M(M-1)} \sum_{\substack{m,h=1 \\ m \neq h}}^M |\hat{y}_m^k - \hat{y}_h^k| \right), \tag{52}$$

$$\mathrm{NLL}(p_\theta^M, \boldsymbol{y}) := \frac{1}{d_y} \sum_{k=1}^{d_y} \log\left(2\pi \hat{\sigma}_k^2\right) + \frac{((\bar{\boldsymbol{y}})^k - y^k)^2}{\hat{\sigma}_k^2}, \tag{53}$$

$$\mathcal{C}_\alpha(p_\theta^M, \boldsymbol{y}) := \frac{1}{d_y} \sum_{k=1}^{d_y} \mathbb{1}\left\{ y^k \in [(\hat{q}_\theta^{\alpha/2})^k, (\hat{q}_\theta^{1-\alpha/2})^k] \right\}. \tag{54}$$

The RMSE evaluates the match between the mean prediction $\bar{\boldsymbol{y}}$ and the true observation, while the energy score (ES) evaluates the match for the predictive distribution as a whole. The continuous ranked probability score (CRPS) (Gneiting & Raftery, 2007) evaluates the predictive distribution at a pointwise level and assesses whether the predicted uncertainty fits the observations at all quantile levels for each marginal $y^k$, i.e., whether the predictive distribution is well-calibrated. Furthermore, we analyze the negative log-likelihood (NLL) of a predictive pointwise Gaussian distribution. Although the prediction can be inherently non-Gaussian, this criterion is commonly used and describes the fit of the predictive distribution in terms of the first two estimated moments. Finally, we report the coverage $\mathcal{C}_\alpha$ of the predictive distribution for selected $\alpha$-quantile levels, averaged over all marginals.

### C.2 TRAINING DETAILS

All of our proposed methods require some restricted output for the parameters, which we realize in the following way. For the marginal $\sigma_{\theta,k}^2(\boldsymbol{x}_t, t)$, we use one additional last layer with a softplus activation and add a threshold of $10^{-6}$ for numerical stability. For the mixture weights $\pi_{\theta,k}$, we apply a softmax activation to guarantee the summation constraint. For the $\epsilon_t^{\mathrm{ES}}$ method, we concatenate the random noise as an additional channel dimension. For the multivariate normal method, implementation is already described in Section A.1. The remaining training details, such as batch size or number of epochs, differ across experiments and are described in the next sections, respectively.

### C.3 UCI REGRESSION

To obtain a fair comparison, we take the train-test splits from Hernandez-Lobato & Adams (2015) and do not finetune our methods on a separate validation set but use sensible hyperparameters. We will copy the training process from the CARD method and apply it, except for some minor modifications to the number of training epochs used for our distributional methods. In particular, we train all of our methods for 5000 epochs except on the yacht dataset, where we train for 10000. We only deviate from this for the larger kin8nm and protein datasets, where we train for 1000 and 2000 epochs, respectively. We use the Adam optimizer (Kingma & Ba, 2017) with a learning rate of 0.001. For the diffusion process, we decided to use 50 timesteps, both during training and inference.

We use a linear noise schedule with $\beta_1 = 0.001$ and $\beta_{50} = 0.35$, which is the noise schedule used to train the CARD method for 50 timesteps. Lastly, we set $K = 3$, when training the $\Sigma_\theta^{\text{mix}}$ method and use for all of our distributional methods the CRPS score as scoring rule.

## C.4 Autoregressive prediction tasks

**Burgers' equation**    The Burgers' equation is given as

$$\partial_s u(s,x) + \partial_x(u^2(s,x)/2) = \nu/\pi \partial_{xx} u(s,x), \quad x \in (0,1), \ s \in (0,2] \tag{55}$$
$$u(0,x) = u_0(x), \quad x \in (0,1)$$

where $u \in C([0,S]; H_{\text{per}}^r((0,1); \mathbb{R}))$ for any $r > 0$, $u_0 \in L_{\text{per}}^2((0,1); \mathbb{R})$ is the initial condition and $\nu \in \mathbb{R}_+$ is the diffusion coefficient[2]. We utilize data from the PDEBENCH repository (Takamoto et al., 2022), which assumes a constant diffusion coefficient, which we choose as $\nu = 0.01$. The data is generated with a periodic boundary condition from a superposition of sinusoidal waves with the temporally and spatially 2nd-order upwind difference scheme for the advection term, and the central difference scheme for the diffusion term.

**Kuramoto–Sivashinsky equation**    Recall that the Kuramoto–Sivashinsky (KS-) equation in one spatial dimension is given as:

$$\partial_s u(x,s) + u\partial_x u(x,s) + \partial_x^2 u(x,s) + \partial_x^4 u(x,s) = 0, \qquad x \in \mathcal{D}, s \in (0,S]$$
$$u(x,0) = u_0(x), \qquad x \in \mathcal{D}$$

where $\mathcal{D} \subseteq \mathbb{R}$, $u \in C([0,S]; H_{\text{per}}^4(\mathcal{D}; \mathbb{R}))$, and $u_0 \in L_{\text{per}}^2(\mathcal{D}; \mathbb{R})$. We follow the setup in Bülte et al. (2025) and simulate the KS-equation from random uniform noise $\mathcal{U}(-1,1)$ on a periodic domain $\mathcal{D} = [0,100]$ using the py-pde package (Zwicker, 2020). We generate 10000 samples with a resolution of $256 \times 50$ and $\Delta s = 2$.

**Surface temperature prediction**    For the surface temperature prediction task, we utilize the ERA5 dataset (Hersbach et al., 2020) provided via the WeatherBench2 benchmark (Rasp et al., 2024). Similar to Bülte et al. (2025), we use data with a spatial resolution of $0.25° \times 0.25°$ and a time resolution of $6h$. For computational reasons, we restrict the data to a European domain, covering an area from $35°N - 75°N$ and $12.5°W - 42.5°E$ with selected user-relevant weather variables (u-component and v-component of 10-m wind speed (U10 and V10), temperature at 2m and 850 hPa (T2M and T850), geopotential height at 500 hPa (Z500), as well as land-sea mask and orography) that serve as input to the model. In contrast to other diffusion approaches (Larsson et al., 2025), we only predict the surface temperature T2M, as this is less computationally demanding. As the initial condition, we always use 00 UTC time and issue a prediction for 06 UTC time.

**Model details**    We adapt the U-Net used in Song et al. (2021); Karras et al. (2022) to allow for arbitrary input shapes and one-dimensional convolutions. We use 64 feature channels for the first layer and 128 channels for layers 2-4. The diffusion noise is encoded with Fourier embeddings (Karras et al., 2022), where the noise is transformed into sine/cosine features at 32 frequencies with base period 16. Afterwards, the features are passed through a 2-layer MLP, which results in a 256-dimensional noise encoding. Finally, this encoding is added to the network through various group norms of the U-Net. The final (regular) diffusion model has 6.5 million and 15.2 million parameters for the 1D and 2D tasks, respectively. While the temperature prediction task uses a fixed prediction horizon, for the PDE tasks, we sample random timesteps $s \in 1, \ldots, S$ for training and evaluation, but split the data such that the test data is unseen during training.

**Training details**    For the 1D PDE tasks, we use an effective batch size of 128, a learning rate of $5 \times 10^{-4}$ with the Adam optimizer, early stopping after 150 epochs and a learning rate scheduler that reduces the learning rate by the factor $0.5$ if the validation loss has not improved for 75 epochs. All models are trained on an NVIDIA RTX3090 with 24GB memory.

For the T2M task, we use an effective batch size of 64, a learning rate of $1 \times 10^{-5}$ with the AdamW optimizer, early stopping after 300 epochs, and a learning rate scheduler that reduces the learning

---

[2] $H_{\text{per}}^r(\mathcal{D}; \mathbb{R}), L_{\text{per}}^2(\mathcal{D}; \mathbb{R})$ denote the periodic Sobolev and $L^2$ spaces, respectively.

rate by the factor $0.5$ if the validation loss has not improved for $150$ epochs. All models are trained on two NVIDIA RTXA6000 with 48GB memory.

### C.5 MONOCULAR DEPTH ESTIMATION

We follow the setup of Marigold (Ke et al., 2024) in general. We only adapt the last layer of their model to our methods as described in Appendix C.2, the alignment of the inference ensemble, and the used metrics. Details on the Marigold setup and our adaptations are shown in the following. Due to the high computational load, we perform our experiments using only one seed.

**Model details**    The idea of Marigold (Ke et al., 2024) is to adapt Stable Diffusion (Rombach et al., 2022) to depth estimation. The frozen VAE is used to encode the image and the corresponding depth map into the latent space for training. As the encoder expects an image with 3 channels, the depth map is replicated into three channels. The embedded RGB image is used as the conditioning $c$, and the embedded depth map gets noise added to serve as $x_t$. To handle this, the first layer of the UNet is duplicated, and the weights are halved to keep the scale of the activations similar. As we need multiple output channels to model the standard deviation as well, we adapt the last layer depending on the used method, as explained in Section C.2. We initialize the variance head randomly, while we initialize the weights of the mean prediction with the weights of the last convolutional layer of the Stable Diffusion UNet.

**Training details**    We follow the exact training procedure of Marigold (Ke et al., 2024). Training is done on the synthetic datasets Hypersim (Roberts et al., 2021) and VKitti2 (Cabon et al., 2020). We use the same affine-invariant depth normalization and annealed multi-resolution noise as Marigold during training, while using Gaussian noise during inference. We perform 1000 diffusion steps with the DDPM scheduler (Ho et al., 2020). We train using an effective batch size of 32 with Adam (Kingma & Ba, 2017) and convergence takes about 3 days on a single Nvidia RTX GPU. We train all models with the same setup and also retrain Marigold ($\delta_\theta$).

**Evaluation datasets**    The evaluation is performed on five real-world datasets. NYUv2 (Silberman et al., 2012) and ScanNet (Dai et al., 2017) are both indoor scene datasets captured with an RGB-D Kinect sensor. NYUv2 has a designated test split with 654 images, which we use. For ScenNet, we sample 800 images from the 312 official validation scenes for testing. The street-scene dataset KITTI (Geiger et al., 2012) comes with a sparse metric depth captured by a LiDAR sensor and we use the Eigen test split (Eigen et al., 2014) made of 652 images. ETH3D (Schops et al., 2017) and DIODE (Vasiljevic et al., 2019) are high-resolution datasets. We utilize the entire ETH3D dataset and the entire validation split of DIODE, which comprises 325 indoor and 446 outdoor samples.

**Evaluation details**    We use $T = 50$ denoising steps and an ensemble size of $N = 10$ for every method. Marigold is tackling affine-invariant depth estimation and thus, the predictions have to be transformed before the evaluation. Furthermore, Marigold proposed to use an ensemble prediction to boost the performance and aligned the sample predictions $\{\hat{d}^1, ..., \hat{d}^N\}$ among each other by minimizing the following objective:

$$\min_{\substack{s_1,...,s_N \\ t_1,...,t_N}} \left( \sqrt{\frac{1}{b} \sum_{i=1}^{N-1} \sum_{j=i+1}^{N} \|\hat{d}^{i\prime} - \hat{d}^{j\prime}\|_2^2} + \lambda \mathcal{R} \right)$$

where $\hat{d}^{i\prime} = \hat{s}^i \hat{d}^i + \hat{t}^i$ are the scaled and shifted predictions, $\mathcal{R} = |\min(m)| + |1 - \max(m)|$ is a regularization term with $m(x, y) = \text{median}(\hat{d}^{1\prime}(x, y), ..., \hat{d}^{N\prime}(x, y))$. Furthermore, $b = \binom{N}{2}$ and $\lambda$ is a hyperparameter. Then, the median $m$ is taken and aligned with the target depth map by a least squares fit $a = sm + t$. We follow Marigold and perform the same steps for every method to get the point prediction $a$. As we are also interested in metrics that consider the whole ensemble, such as ES and CRPS, further align the ensemble by setting $a^i = s\hat{d}^{i\prime} + t$, i.e., we align each ensemble member with the same transformation that we computed for the median $\hat{d}$. We evaluate the prediction using the UQ metrics ES and CRPS and the metrics reported in Marigold ($d = (d_i)_{i=1,...,M}$ denotes the true depth map and $M$ the number of pixels):

- Absolute Mean Relativ Error (*AbsRel*): $\frac{1}{M}\Sigma_{i=1}^{M}|a_i - d_i|/d_i$
- $\delta_1$ Accuracy: Proportion of pixels satisfying $\max(a_i/d_i, d_i/a_i) < 1.25$

# D ADDITIONAL RESULTS

This section provides more detailed results of the different experiments. Tables 5, 6, 7 and 8 outline several considered metrics for the UCI dataset with standard deviations. Table 9 shows the results for the different autoregressive prediction tasks, averaged over five runs (except for T2M). While we report the average time per training epoch, the underlying algorithms are not optimized and might heavily depend on the used architecture and the availability of the compute cluster. They should therefore only be seen as a rough estimate for identifying methods that take a significantly longer time for training. Corresponding visualizations of the different methods can be found in Figures 6, 7, 8, 9, and 10. Table 10 shows more results of the depth estimation task.

Table 5: RMSE ($\downarrow$) on the test set for UCI datasets, averaged and with standard deviation.

|  | $\boldsymbol{\delta}_\theta$ | $\boldsymbol{\Sigma}_\theta^{\mathrm{diag}}$ | $\boldsymbol{\Sigma}_\theta^{\mathrm{mix}}$ | $\boldsymbol{\epsilon}_t^{\mathrm{ES}}$ |
|---|---|---|---|---|
| Concrete | 4.81 ± 0.63 | **4.72 ± 0.7** | 4.73 ± 0.63 | 4.81 ± 0.65 |
| Energy | 0.45 ± 0.06 | 0.45 ± 0.07 | **0.44 ± 0.06** | 0.45 ± 0.06 |
| Kin8nm | 6.91 ± 0.19 | **6.88 ± 0.19** | 6.89 ± 0.19 | 6.9 ± 0.19 |
| Naval | 1.35 ± 0.1 | 1.24 ± 0.09 | **1.23 ± 0.09** | **1.23 ± 0.09** |
| Power | 3.86 ± 0.17 | 3.64 ± 0.19 | **3.59 ± 0.17** | 3.78 ± 0.17 |
| Protein | 3.76 ± 0.04 | **3.71 ± 0.05** | 3.72 ± 0.05 | 3.76 ± 0.04 |
| Wine | 0.67 ± 0.08 | 0.67 ± 0.08 | **0.66 ± 0.07** | **0.66 ± 0.08** |
| Yacht | 0.74 ± 0.4 | **0.7 ± 0.31** | 0.78 ± 0.5 | 0.8 ± 0.49 |

Table 6: CRPS($\downarrow$) on the test set for UCI datasets, averaged and with standard deviation.

|  | $\boldsymbol{\delta}_\theta$ | $\boldsymbol{\Sigma}_\theta^{\mathrm{diag}}$ | $\boldsymbol{\Sigma}_\theta^{\mathrm{mix}}$ | $\boldsymbol{\epsilon}_t^{\mathrm{ES}}$ |
|---|---|---|---|---|
| Concrete | 2.6 ± 0.33 | 2.46 ± 0.37 | **2.45 ± 0.34** | 2.5 ± 0.34 |
| Energy | 0.3 ± 0.03 | 0.25 ± 0.03 | **0.24 ± 0.03** | 0.25 ± 0.03 |
| Kin8nm | 3.93 ± 0.11 | **3.81 ± 0.1** | 3.82 ± 0.1 | 3.83 ± 0.1 |
| Naval | 0.79 ± 0.1 | 0.56 ± 0.06 | 0.54 ± 0.05 | **0.53 ± 0.05** |
| Power | 2.04 ± 0.05 | 1.84 ± 0.06 | **1.81 ± 0.06** | 1.93 ± 0.07 |
| Protein | 1.71 ± 0.02 | 1.66 ± 0.03 | **1.65 ± 0.03** | 1.69 ± 0.02 |
| Wine | 0.34 ± 0.06 | 0.34 ± 0.06 | 0.34 ± 0.06 | **0.33 ± 0.06** |
| Yacht | 0.36 ± 0.17 | **0.28 ± 0.12** | 0.31 ± 0.18 | 0.32 ± 0.19 |

Table 7: $\mathcal{C}_{0.95}$ on the test set for UCI datasets, averaged and with standard deviation.

|  | $\boldsymbol{\delta}_\theta$ | $\boldsymbol{\Sigma}_\theta^{\mathrm{diag}}$ | $\boldsymbol{\Sigma}_\theta^{\mathrm{mix}}$ | $\boldsymbol{\epsilon}_t^{\mathrm{ES}}$ |
|---|---|---|---|---|
| Concrete | 0.57 ± 0.05 | 0.68 ± 0.05 | 0.68 ± 0.06 | **0.73 ± 0.05** |
| Energy | 0.28 ± 0.12 | **0.73 ± 0.06** | 0.72 ± 0.06 | 0.72 ± 0.07 |
| Kin8nm | 0.77 ± 0.02 | **0.86 ± 0.02** | **0.86 ± 0.01** | **0.86 ± 0.01** |
| Naval | 0.25 ± 0.09 | 0.93 ± 0.03 | **0.94 ± 0.02** | 0.92 ± 0.03 |
| Power | 0.82 ± 0.02 | 0.89 ± 0.01 | 0.87 ± 0.02 | **0.9 ± 0.01** |
| Protein | 0.86 ± 0.01 | **0.91 ± 0.0** | 0.89 ± 0.01 | **0.91 ± 0.01** |
| Wine | 0.53 ± 0.12 | 0.76 ± 0.05 | 0.7 ± 0.08 | **0.79 ± 0.04** |
| Yacht | 0.34 ± 0.12 | 0.78 ± 0.09 | 0.78 ± 0.09 | **0.79 ± 0.1** |

Table 8: NLL($\downarrow$) on the test set for UCI datasets, averaged and with standard deviation.

| | $\boldsymbol{\delta}_\theta$ | $\boldsymbol{\Sigma}_\theta^{\mathrm{diag}}$ | $\boldsymbol{\Sigma}_\theta^{\mathrm{mix}}$ | $\boldsymbol{\epsilon}_t^{\mathrm{ES}}$ |
|---|---|---|---|---|
| Concrete | 10.35 ± 5.99 | 9.63 ± 6.53 | 7.93 ± 4.1 | **4.97 ± 1.84** |
| Energy | 63.59 ± 85.17 | **1.6 ± 1.2** | 1.62 ± 1.28 | 1.63 ± 1.27 |
| Kin8nm | -0.99 ± 0.09 | **-1.22 ± 0.05** | **-1.22 ± 0.04** | **-1.22 ± 0.04** |
| Naval | 24.71 ± 26.16 | -7.94 ± 0.13 | **-7.98 ± 0.13** | -7.93 ± 0.17 |
| Power | 2.98 ± 0.12 | 2.84 ± 0.19 | 2.92 ± 0.3 | **2.77 ± 0.08** |
| Protein | 3.21 ± 0.36 | 3.19 ± 0.34 | 3.8 ± 0.45 | **2.91 ± 0.13** |
| Wine | 3903.17 ± 5036.57 | 173.79 ± 211.5 | 865.12 ± 880.86 | **116.0 ± 138.55** |
| Yacht | 55.75 ± 65.65 | 3.39 ± 7.32 | 6.34 ± 13.65 | **2.58 ± 4.93** |

Table 9: Results for the autoregressive prediction tasks. The RMSE, ES, and CRPS are scaled by the factor 1000 (100) for the Burgers' (KS) equation. The best model is highlighted in bold, and the standard deviation across the different runs is shown in brackets.

| Experiment | Model | RMSE$\downarrow$ | ES$\downarrow$ | CRPS$\downarrow$ | NLL$\downarrow$ | $\mathcal{C}_{0.95}$ |
|---|---|---|---|---|---|---|
| **Burgers'** | $\boldsymbol{\delta}_\theta$ | 0.95 (± 0.39) | 4.26 (± 0.71) | 0.16 (± 0.03) | -7.09 (± 0.17) | **0.94** (± 0.01) |
| | $\boldsymbol{\Sigma}_\theta^{\mathrm{diag}}$ | 0.81 (± 0.27) | 3.67 (± 0.39) | **0.12** (± 0.02) | **-7.12** (± 0.09) | 1.00 (± 0.00) |
| | $\boldsymbol{\Sigma}_\theta^{\mathrm{mix}}$ | 0.81 (± 0.27) | 3.82 (± 0.39) | 0.13 (± 0.02) | -6.99 (± 0.19) | 1.00 (± 0.00) |
| | $\boldsymbol{\Sigma}_\theta^{\mathrm{mv}}$ | **0.70** (± 0.20) | 6.35 (± 0.32) | 0.24 (± 0.02) | -6.10 (± 0.04) | 1.00 (± 0.00) |
| | $\boldsymbol{\epsilon}_t^{\mathrm{ES}}$ | 0.81 (± 0.49) | **3.59** (± 1.05) | 0.14 (± 0.03) | -7.00 (± 0.14) | 0.99 (± 0.00) |
| **KS** | $\boldsymbol{\delta}_\theta$ | 0.56 (± 0.06) | 5.93 (± 0.62) | 0.24 (± 0.02) | -3.99 (± 0.17) | 0.84 (± 0.05) |
| | $\boldsymbol{\Sigma}_\theta^{\mathrm{diag}}$ | 0.39 (± 0.06) | 5.23 (± 0.75) | 0.21 (± 0.03) | -4.04 (± 0.14) | 1.00 (± 0.00) |
| | $\boldsymbol{\Sigma}_\theta^{\mathrm{mix}}$ | **0.35** (± 0.04) | **4.91** (± 0.54) | **0.20** (± 0.02) | **-4.08** (± 0.09) | 1.00 (± 0.00) |
| | $\boldsymbol{\Sigma}_\theta^{\mathrm{mv}}$ | 0.49 (± 0.01) | 7.68 (± 0.08) | 0.34 (± 0.01) | -3.59 (± 0.02) | 0.99 (± 0.00) |
| | $\boldsymbol{\epsilon}_t^{\mathrm{ES}}$ | 0.59 (± 0.07) | 7.34 (± 1.51) | 0.34 (± 0.08) | -3.53 (± 0.28) | **0.98** (± 0.01) |
| **T2M** | $\boldsymbol{\delta}_\theta$ | 0.77 | 103.21 | 0.40 | 1.05 | 0.97 |
| | $\boldsymbol{\Sigma}_\theta^{\mathrm{diag}}$ | **0.71** | 96.48 | **0.35** | 0.97 | 0.84 |
| | $\boldsymbol{\Sigma}_\theta^{\mathrm{mix}}$ | **0.71** | **95.68** | **0.35** | 0.98 | 0.83 |
| | $\boldsymbol{\Sigma}_\theta^{\mathrm{mv}}$ | 0.76 | 101.39 | 0.38 | **0.93** | **0.94** |

Table 10: Full depth estimation results.

| Experiment | Model | AbsRel ↓ | $\delta 1 \uparrow$ | CRPS ↓ | ES ↓ |
|---|---|---|---|---|---|
| **NYUv2** | $\delta_\theta$ | 5.96 | 95.95 | 11.32 | 8324.67 |
| | $\Sigma_\theta^{\mathrm{diag}}$ | 5.90 | 95.94 | 11.32 | 8341.45 |
| | $\Sigma_\theta^{\mathrm{mix}}$ | 5.89 | 95.99 | 11.35 | 8397.20 |
| | $\Sigma_\theta^{\mathrm{mv}}$ | **5.67** | **96.15** | **11.02** | **8238.10** |
| **KITTI** | $\delta_\theta$ | 10.32 | 90.11 | 142.92 | 655.03 |
| | $\Sigma_\theta^{\mathrm{diag}}$ | 10.07 | 90.81 | 138.24 | **638.46** |
| | $\Sigma_\theta^{\mathrm{mix}}$ | **9.89** | **91.09** | **137.60** | 640.34 |
| | $\Sigma_\theta^{\mathrm{mv}}$ | 10.14 | 90.95 | 142.28 | 649.21 |
| **ETH3D** | $\delta_\theta$ | 6.82 | 95.62 | 29.23 | 906.37 |
| | $\Sigma_\theta^{\mathrm{diag}}$ | 6.57 | 95.59 | **27.89** | **886.84** |
| | $\Sigma_\theta^{\mathrm{mix}}$ | 6.72 | 95.66 | 28.66 | 887.65 |
| | $\Sigma_\theta^{\mathrm{mv}}$ | **6.47** | **95.99** | 29.43 | 916.28 |
| **ScanNet** | $\delta_\theta$ | 7.10 | **94.67** | 9.19 | 74.28 |
| | $\Sigma_\theta^{\mathrm{diag}}$ | 6.84 | **94.67** | 8.86 | **72.88** |
| | $\Sigma_\theta^{\mathrm{mix}}$ | 6.96 | 94.34 | 8.99 | 73.57 |
| | $\Sigma_\theta^{\mathrm{mv}}$ | **6.79** | 94.51 | **8.85** | 73.41 |
| **DIODE** | $\delta_\theta$ | 30.60 | 77.04 | 191.39 | 2328.79 |
| | $\Sigma_\theta^{\mathrm{diag}}$ | 31.52 | 76.41 | 196.08 | 2349.49 |
| | $\Sigma_\theta^{\mathrm{mix,}}$ | 31.09 | 77.01 | 190.04 | 2302.41 |
| | $\Sigma_\theta^{\mathrm{mv}}$ | **29.82** | **77.78** | **186.63** | **2284.49** |

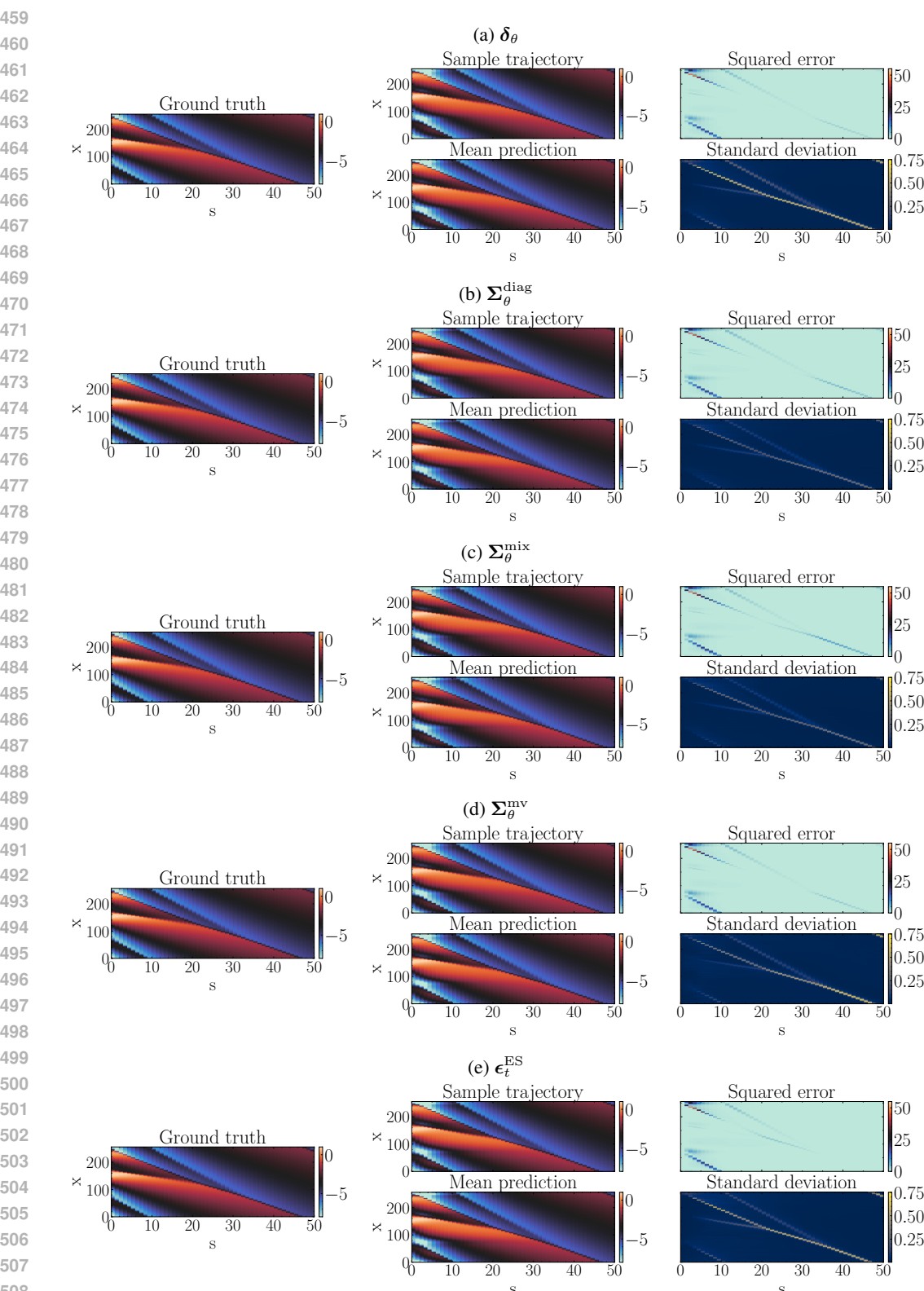

Figure 6: Comparison of the predicted autoregressive trajectories of the Burgers' equation for the different models.

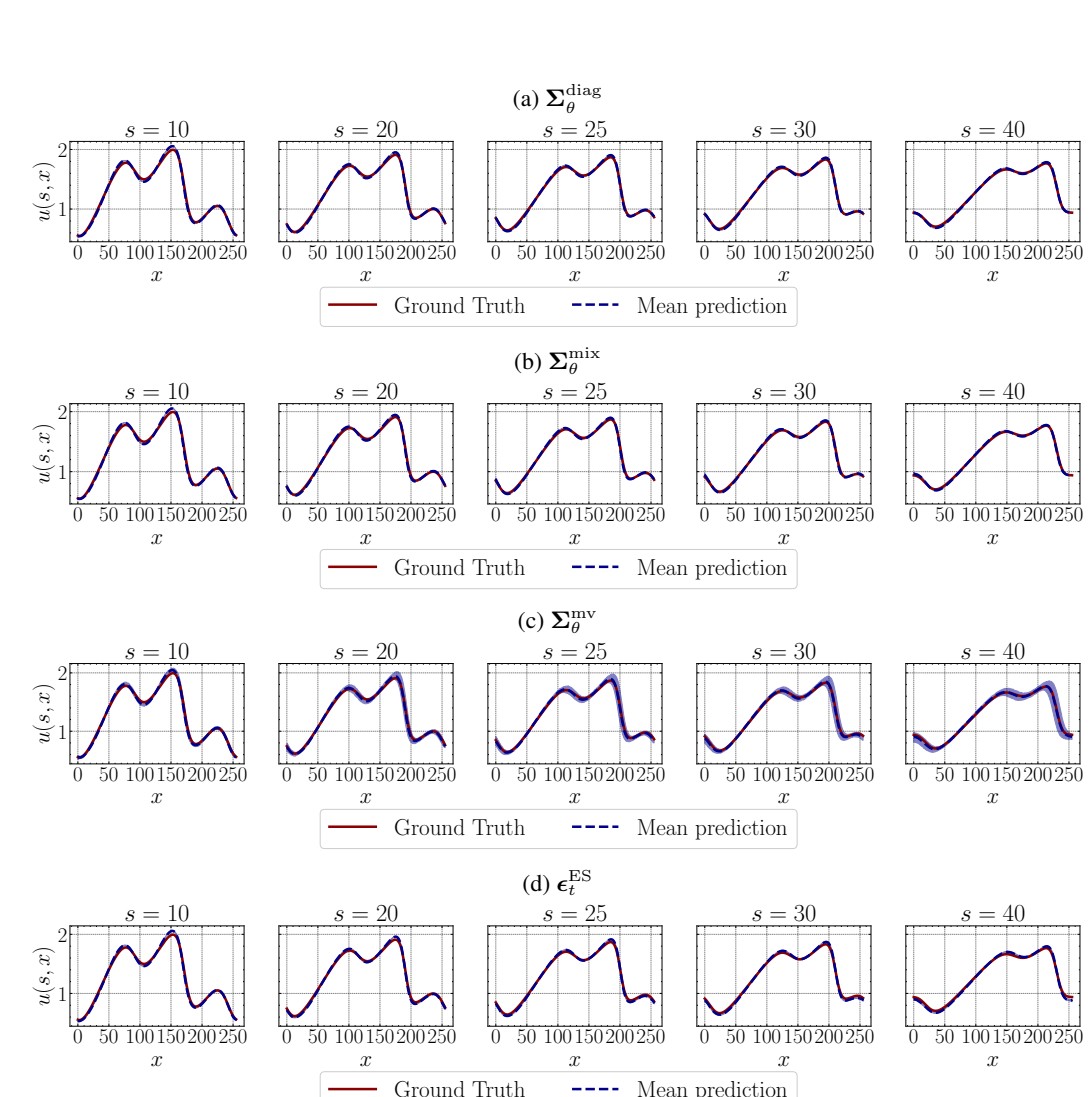

Figure 7: Comparison of the predicted solutions of the Burgers' equation for different steps $s$. Shown are the ground truth, mean prediction, and 95% confidence interval.

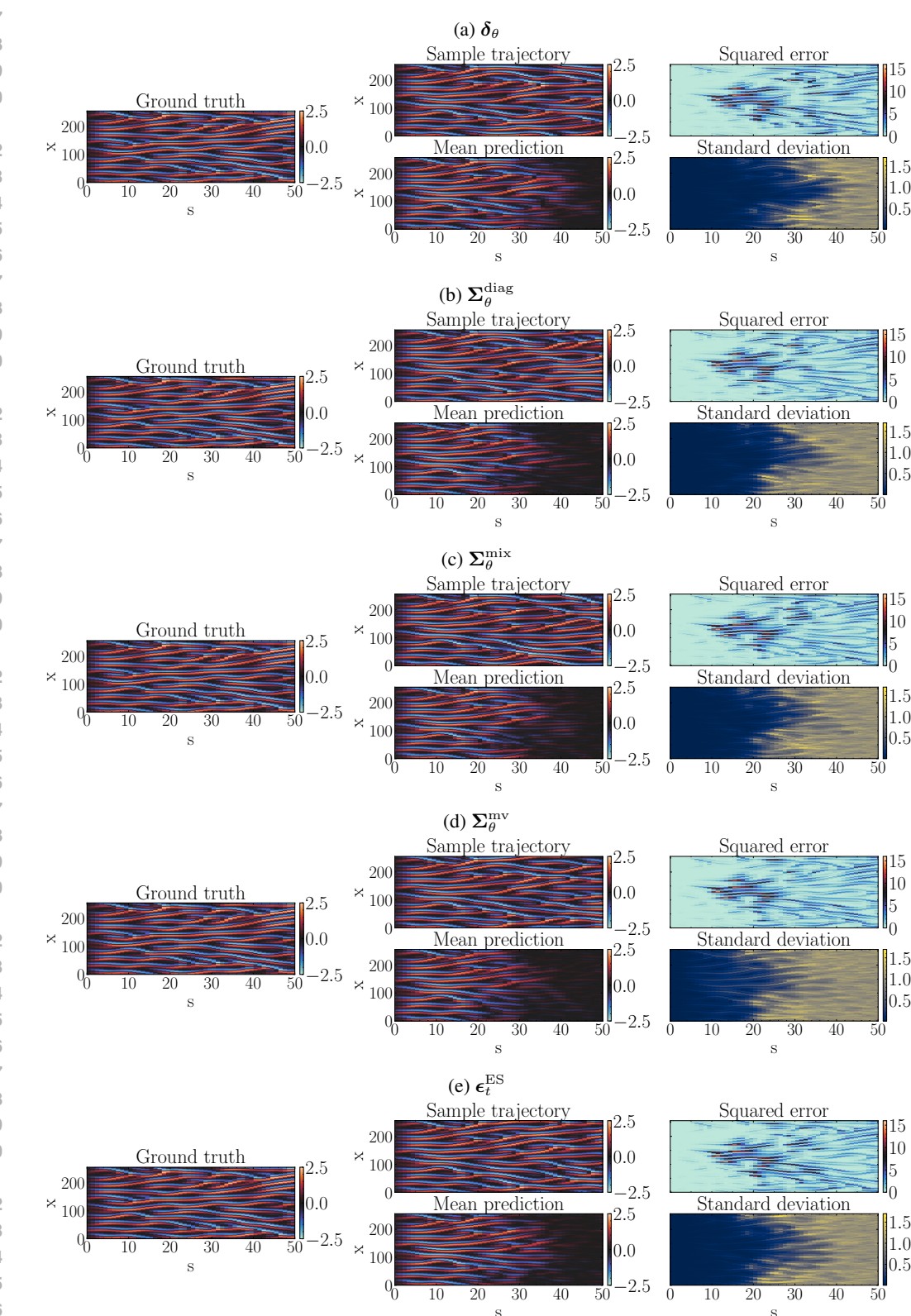

Figure 8: Comparison of the predicted autoregressive trajectories of the Kuramoto–Sivashinsky equation for the different models.

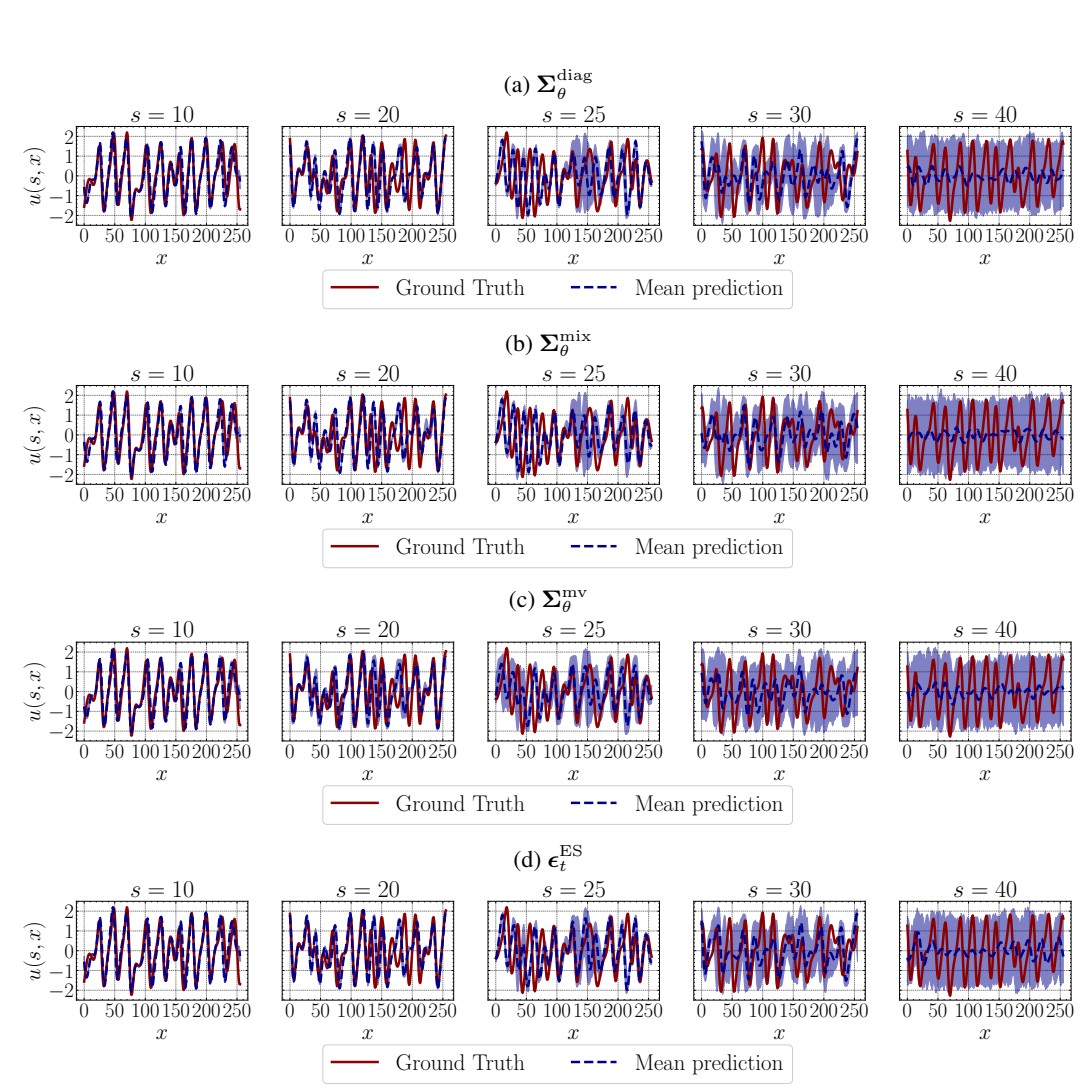

Figure 9: Comparison of the predicted solutions of the KS equation for different steps $s$. Shown are the ground truth, mean prediction, and 95% confidence interval.

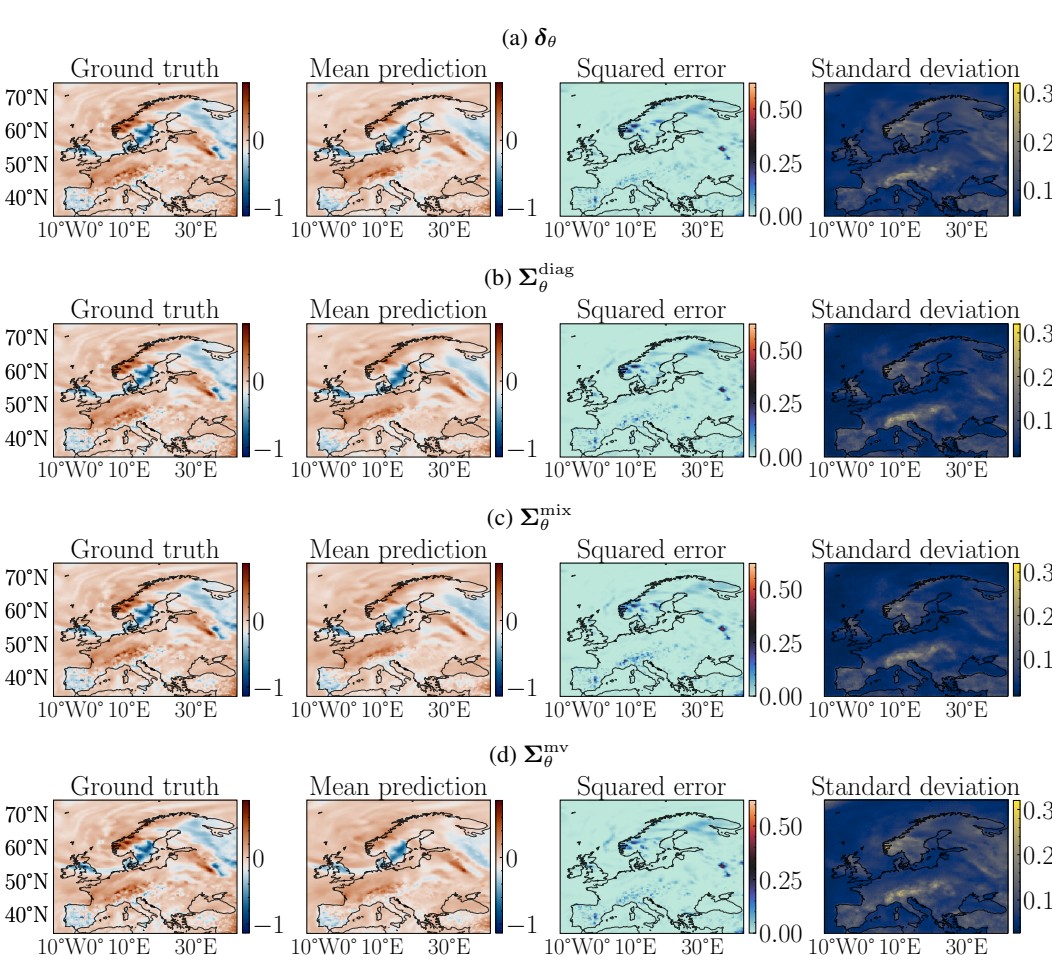

Figure 10: Comparison of the predicted dynamics of the two-meter surface temperature for the different models.

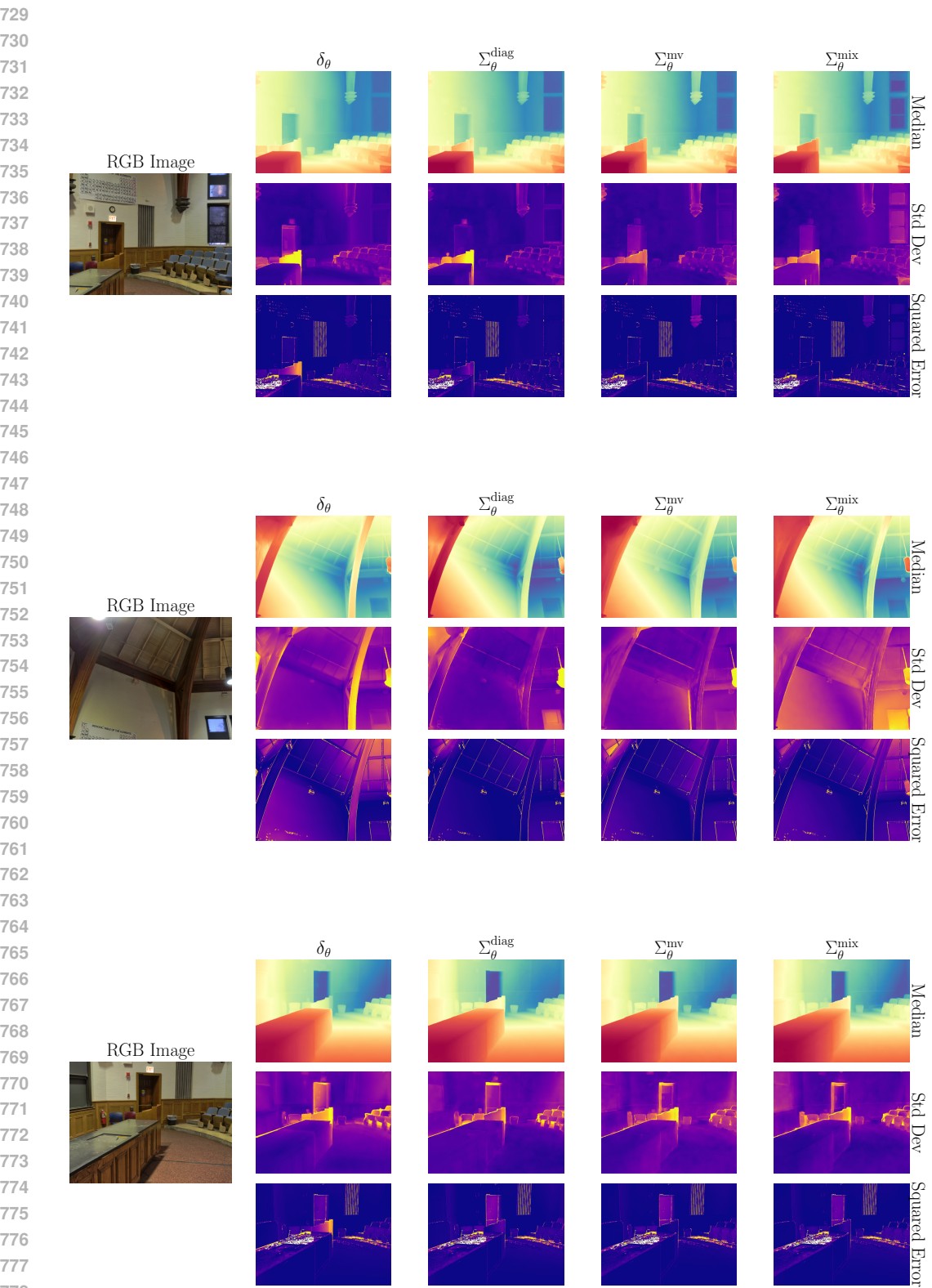

Figure 11: Samples from the Diode dataset (Vasiljevic et al., 2019).

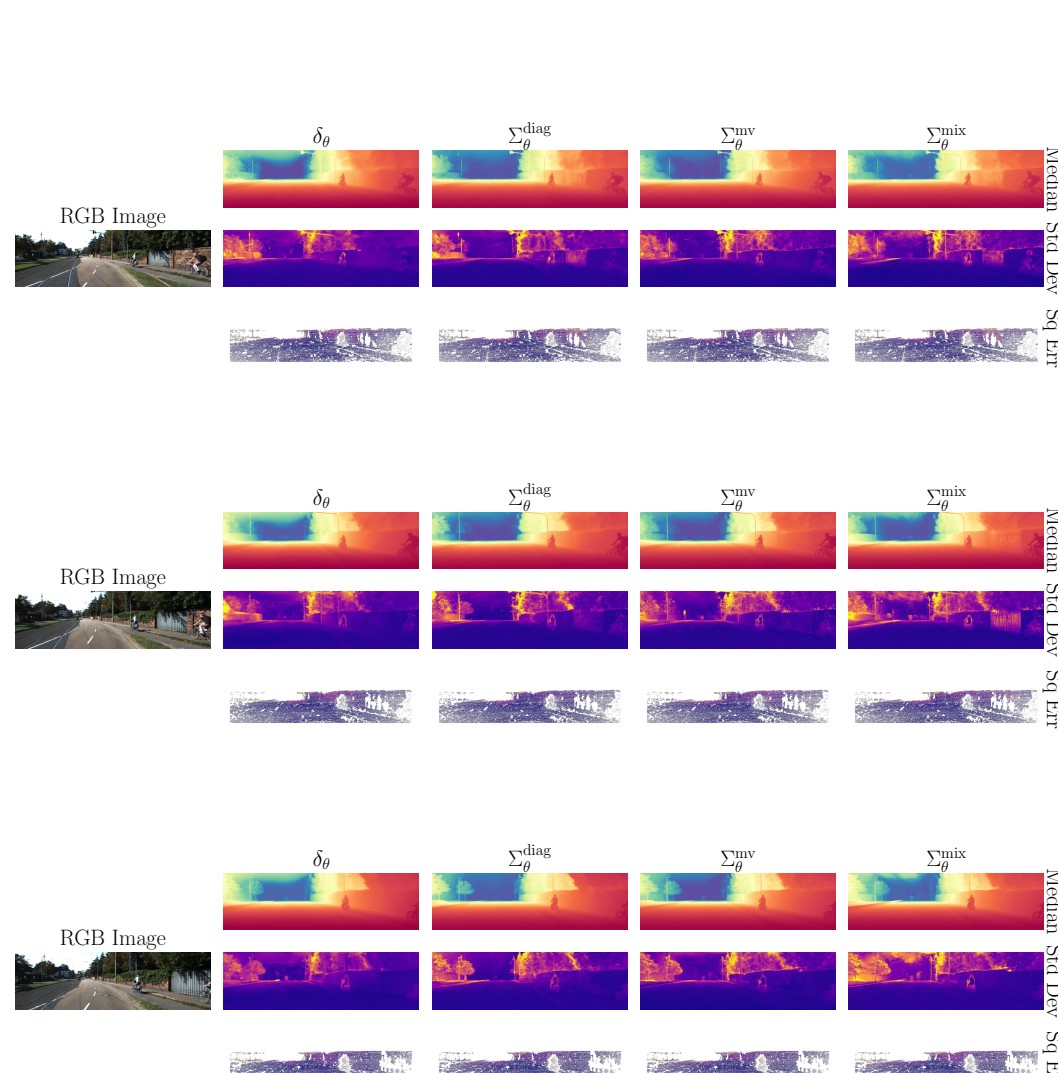

Figure 12: Samples from the Kitti dataset (Geiger et al., 2012).

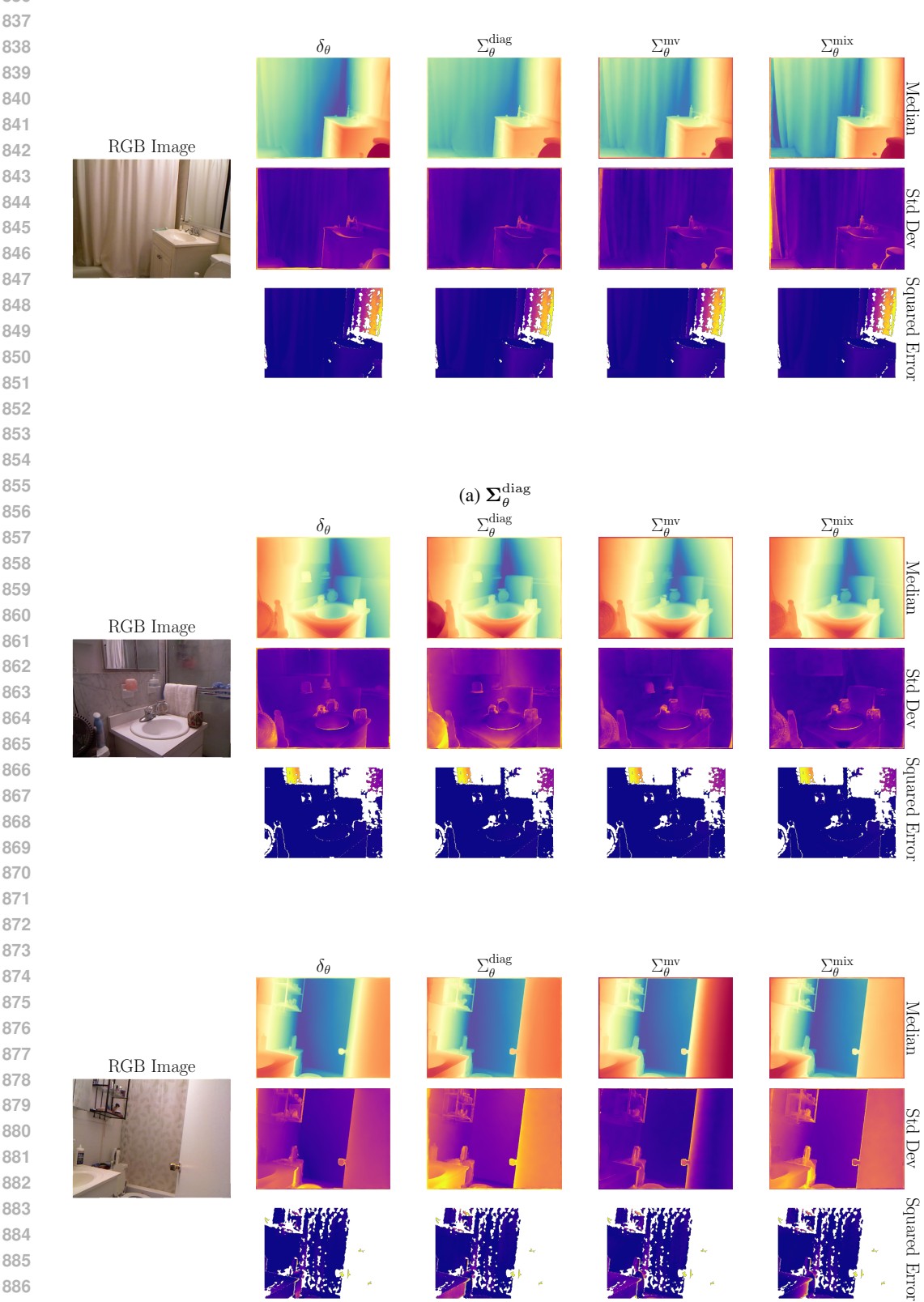

(a) $\mathbf{\Sigma}_\theta^{\text{diag}}$

Figure 13: Samples from the NYU dataset (Silberman et al., 2012).

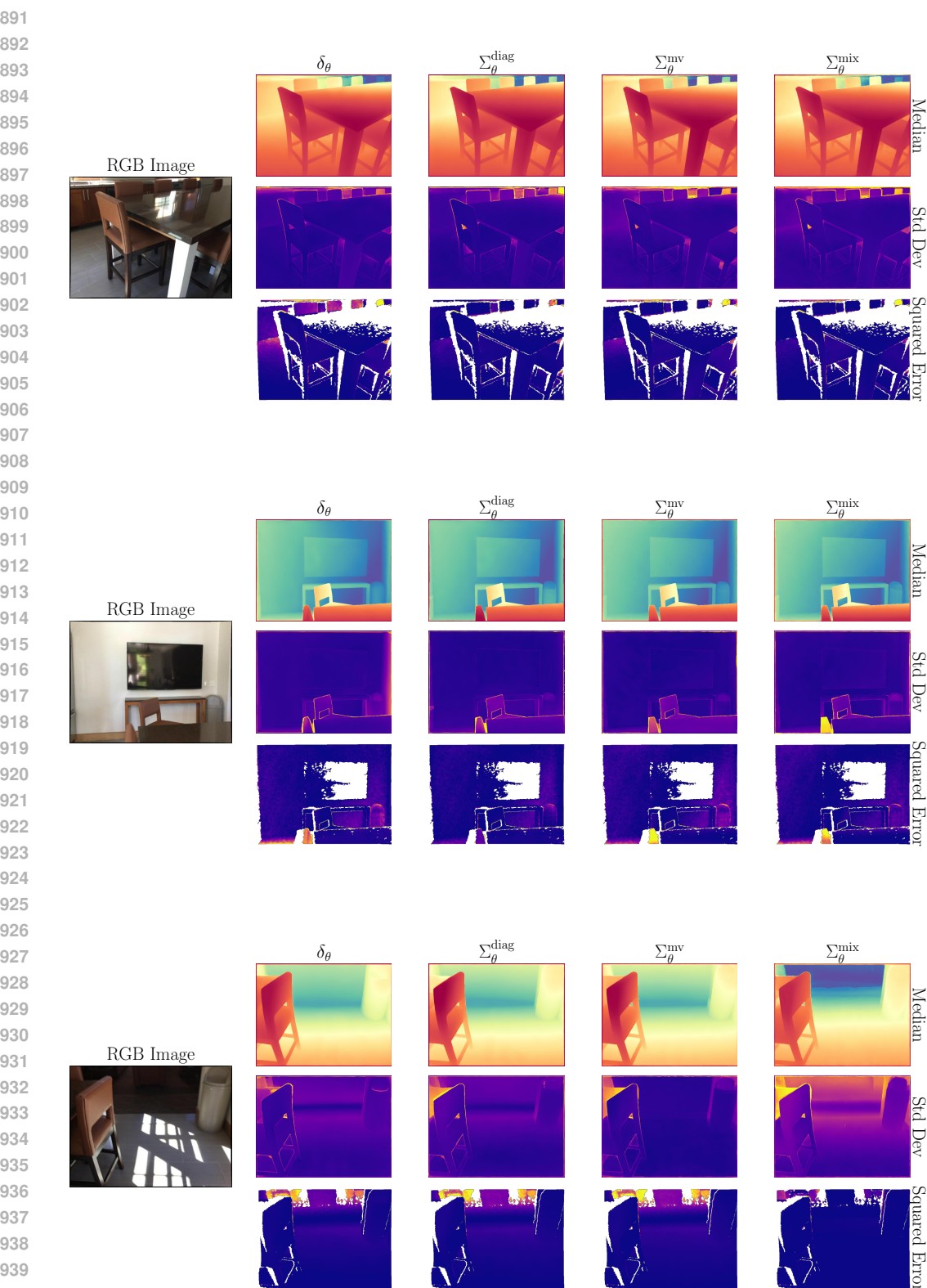

Figure 14: Samples from the ScanNet dataset (Dai et al., 2017).

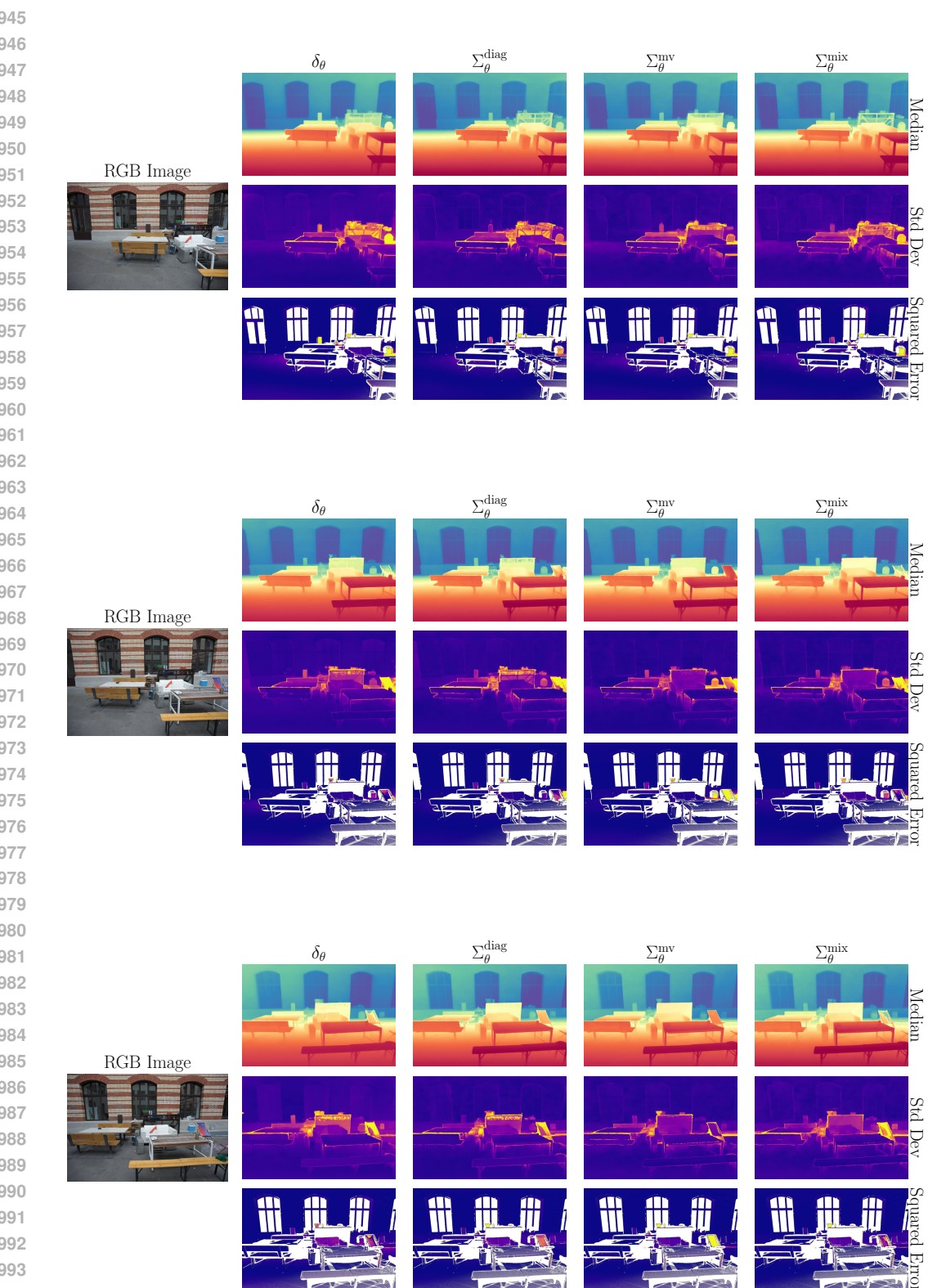

Figure 15: Samples from the ETH3D dataset (Schops et al., 2017).

## D.1 RUNTIME ANALYSIS

While we have already demonstrated improved predictive performance, we now briefly examine whether training with scoring rules incurs additional computational cost. As extensive efficiency evaluations for computationally demanding models such as Marigold ( 900M parameters) are impractical, we focus on the two one-dimensional PDE tasks. Since the architectural and training differences are minimal, these results transfer directly to the remaining experiments.

Table 11: Runtime per epoch [s] for the different methods and the two PDE experiments. The results are averaged over 1000 epochs and five independent experiment runs. The best model is highlighted in bold.

| Experiment | $\boldsymbol{\delta}_\theta$ | $\boldsymbol{\Sigma}_\theta^{\text{diag}}$ | $\boldsymbol{\Sigma}_\theta^{\text{mix}}$ | $\boldsymbol{\Sigma}_\theta^{\text{mv}}$ | $\boldsymbol{\epsilon}_t^{\text{ES}}$ |
|---|---|---|---|---|---|
| Burgers' | 7.20 ($\pm$ 0.60) | **7.13** ($\pm$ 0.61) | 7.73 ($\pm$ 0.77) | 8.01 ($\pm$ 0.67) | 10.53 ($\pm$ 0.13) |
| KS | **6.71** ($\pm$ 0.33) | 6.95 ($\pm$ 0.52) | 10.37 ($\pm$ 0.08) | 6.90 ($\pm$ 0.12) | 10.41 ($\pm$ 0.04) |

Table 11 shows the runtime per epoch (in seconds), averaged over 1000 epochs and five runs, using the same hyperparameters as the previous section, following the hyperparameter tuning in Appendix H. It is clear that there are no significant differences in the runtime of the $\boldsymbol{\delta}_\theta$, $\boldsymbol{\Sigma}_\theta^{\text{diag}}$, and $\boldsymbol{\Sigma}_\theta^{\text{mv}}$ methods. For $\boldsymbol{\Sigma}_\theta^{\text{mix}}$, the computational cost grows with the number of mixture components: For $K = 2$ (Burgers'), the runtime remains comparable to the other methods, whereas for $K = 50$ (KS), it increases noticeably, as the loss scales quadratically in $K$. The sample-based method $\boldsymbol{\epsilon}_t^{\text{ES}}$ by Bortoli et al. (2025) generally has significantly higher cost, even for a small number of samples, $M = 3$. These results show that while our method leads to improved performance, it comes at no additional training cost, except in cases with highly parameterized mixture models.

## D.2 STATISTICAL SIGNIFICANCE OF OUR RESULTS

In order to assess whether our methods significantly improve upon the baseline we are presenting a statistical test. In particular, we are performing a sign test Demšar (2006) over all datasets and tasks considered in this paper and count how many times a particular method proposed by us beats the diffusion baseline $\delta_\theta$. We have decided on the sign test as it has the lowest requirements on the metrics distribution, in particular, it does not assume any commensurability.

Table 12: Counts of wins of our methods compared to the baseline $\delta_\theta$ over all our 16 datasets on which we evaluated our methods. In parenthesis the $p$-value. We consider the RMSE or AbsRel for the computer vision tasks to assess classical regression performance and the CRPS for distributional fit.

| Metric | $\boldsymbol{\Sigma}_\theta^{\text{diag}}$ | $\boldsymbol{\Sigma}_\theta^{\text{mix}}$ | $\boldsymbol{\Sigma}_\theta^{\text{mv}}$ |
|---|---|---|---|
| RMSE/AbsRel | 13 (0.0106) | 14 (0.0022) | 15 (0.0002) |
| CRPS | 13 (0.0106) | 14 (0.0022) | 12 (0.0384) |

Table 12 shows the number of times that one of our models is better than the baseline. Since we have in total $n = 16$ datasets, a result is significant with $p = 0.05$ once it wins 12 or more times Demšar (2006). Thus, all of our methods are significantly better than the baseline.

## E IMPROVING MARGINAL CALIBRATION

Although our method generally outperforms the standard diffusion model, especially on PDEs, it tends to be over-conservative, leading to overly high coverage and thus suboptimal marginal calibration. A simple remedy is to rescale the covariance matrix of $p_\theta(\boldsymbol{x}_{t-1}|\boldsymbol{x}_t)$. Since the noise is always at least $\sigma_t^2\mathbf{I}$, adding a learned covariance can produce an overly dispersed distribution. Multiplying

the covariance by a scalar reduces this spread and improves calibration. While this deviates from the DDIM framework and warrants further theoretical analysis, we show empirically that it can enhance both calibration and predictive performance.

From Theorem 1, we have

$$p_\theta(\boldsymbol{x}_{t-1}|\boldsymbol{x}_t) = \sum_{k=1}^{K} \pi_{\theta,k} \mathcal{N}\left(\boldsymbol{x}_{t-1}; \sqrt{\bar{\alpha}_{t-1}}\hat{\boldsymbol{x}}_0 + \sqrt{1 - \bar{\alpha}_{t-1} - \sigma_t^2}\boldsymbol{\mu}_{\theta,k}^\epsilon(\boldsymbol{x}_t, t), \gamma_t^2 \boldsymbol{\Sigma}_{\theta,k}^\epsilon(\boldsymbol{x}_t, t) + \sigma_t^2 \mathbf{I}\right).$$

We define the rescaled covariance as

$$\tilde{\Sigma}_{\theta,k}^\epsilon := \tau\left(\gamma_t^2 \boldsymbol{\Sigma}_{\theta,k}^\epsilon(\boldsymbol{x}_t, t) + \sigma_t^2 \mathbf{I}\right), \quad \tau \in (0, 1].$$

Here, $\tau = 1$ recovers the original covariance, while smaller $\tau$ reduces marginal variances. This adjustment can be applied to all diffusion methods, though for $\boldsymbol{\delta}_\theta$ and $\boldsymbol{\epsilon}_t^{\text{ES}}$ only the diagonal term $\sigma_t^2 \mathbf{I}$ is rescaled. Figure 16 illustrates the effect of $\tau$ on different metrics.

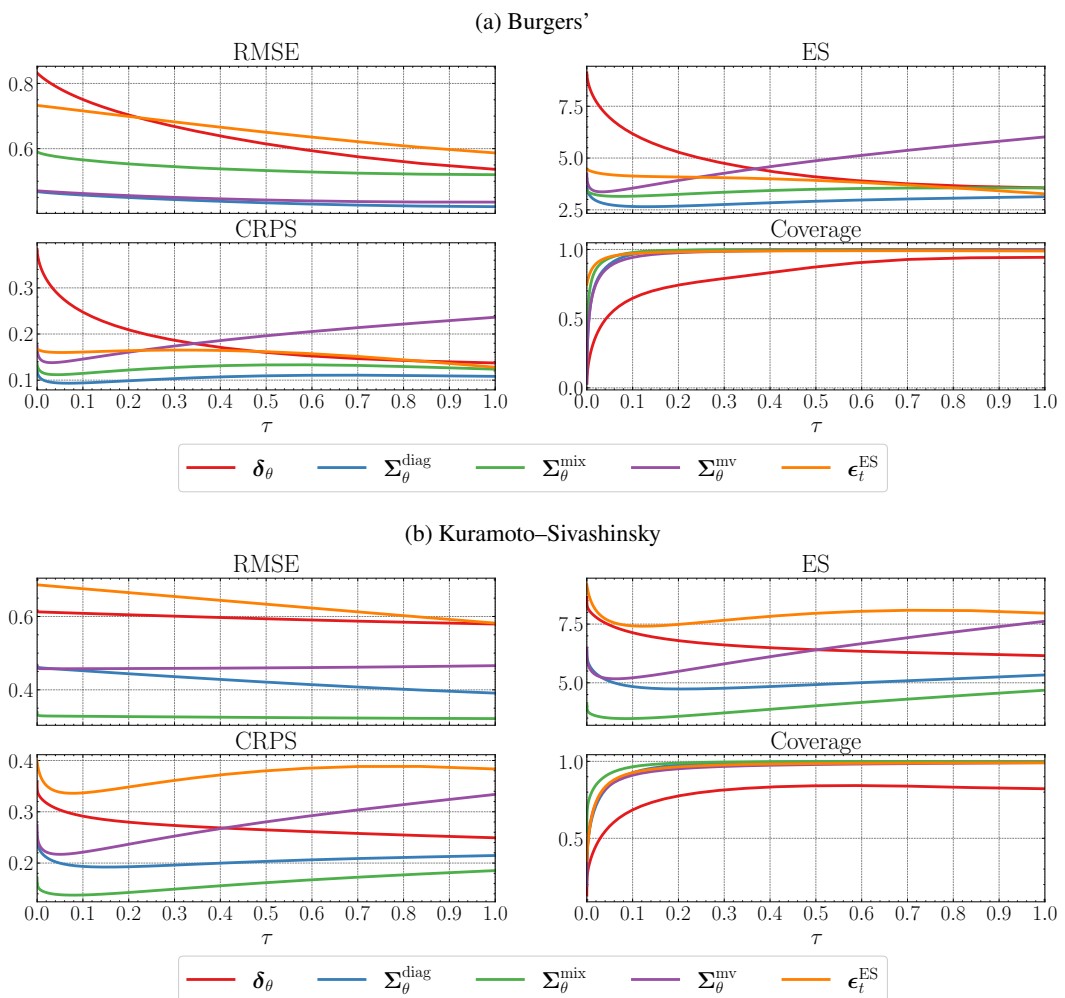

Figure 16: Performance of the different diffusion models in dependence on the scaling parameter $\tau$.

Results show that RMSE is typically minimized at $\tau = 1$, but for our proposed models ($\boldsymbol{\Sigma}_\theta^{\text{diag}}$, $\boldsymbol{\Sigma}_\theta^{\text{mix}}$, $\boldsymbol{\Sigma}_\theta^{\text{mv}}$), smaller values $\tau \approx 0.05$ improve coverage (close to 0.95) and yield substantial gains in CRPS and energy score—up to a factor of two. For $\boldsymbol{\delta}_\theta$ performance is best at $\tau = 1$, and for $\boldsymbol{\epsilon}_t^{\text{ES}}$, only CRPS and energy score show slight benefits for smaller $\tau$.

In summary, rescaling the covariance offers a simple way to improve marginal calibration and predictive performance, though not uniformly across metrics. Choosing $\tau$ optimally remains an open problem, and introducing this parameter moves the model away from the DDIM framework. A theoretical study of the rescaled diffusion process offers a promising direction for future work.

## F  COMPARISON WITH IMPROVED DDPM

In the following, we compare our new method with improved DDPM (iDDPM) (Nichol & Dhariwal, 2021), an extended version of DDPM that assumes the denoising distribution to be Gaussian and aims to learn its mean and variance. As a loss function, they use a composite loss

$$L_{\text{hybrid}} = L_{\text{simple}} + \lambda L_{\text{VLB}} \tag{56}$$

where $L_{\text{simple}}$ is the MSE loss given in Equation 6 and $L_{\text{VLB}}$ is the variational lower bound:

$$L_{\text{VLB}} \coloneqq L_0 + L_1 + ... + L_T$$
$$L_0 \coloneqq -\log p_\theta(\boldsymbol{x}_0|\boldsymbol{x}_1)$$
$$L_{t-1} \coloneqq D_{KL}\left(q(\boldsymbol{x}_{t-1}|\boldsymbol{x}_t, \boldsymbol{x}_0)||p_\theta(\boldsymbol{x}_{t-1}|\boldsymbol{x}_t)\right)$$
$$L_T \coloneqq D_{KL}\left(q(\boldsymbol{x}_T|\boldsymbol{x}_0)||p_\theta(\boldsymbol{x}_T)\right)$$

The VLB loss is necessary to incorporate variance information; however, training with it is very unstable, which is why they use the composite loss $L_{\text{hybrid}}$ and set $\lambda = 0.001$ in their experiments. The model architecture is the same as our univariate Gaussian, however, we use a very different loss function, the CRPS, as described in Appendix A.2. In Table 13, we show the results of iDDPM on the KS equation. While iDDPM outperforms $\boldsymbol{\delta}_\theta$ in every metric, it performs worse than $\boldsymbol{\Sigma}_\theta^{\text{diag}}$ and especially $\boldsymbol{\Sigma}_\theta^{\text{mix}}$ in all metrics but NLL.

Table 13: Results for the KS equation. The RMSE, ES, and CRPS are scaled by the factor 100. The best model is highlighted in bold, and the standard deviation across the different runs is shown in brackets.

| Experiment | Model | RMSE↓ | ES↓ | CRPS↓ | NLL↓ | $\mathcal{C}_{0.95}$ |
|---|---|---|---|---|---|---|
| **KS** | $\boldsymbol{\delta}_\theta$ | 0.56 ($\pm$ 0.06) | 5.93 ($\pm$ 0.62) | 0.24 ($\pm$ 0.02) | -3.99 ($\pm$ 0.17) | 0.84 ($\pm$ 0.05) |
| | $\boldsymbol{\Sigma}_\theta^{\text{diag}}$ | 0.39 ($\pm$ 0.06) | 5.23 ($\pm$ 0.75) | 0.21 ($\pm$ 0.03) | -4.04 ($\pm$ 0.14) | 1.00 ($\pm$ 0.00) |
| | $\boldsymbol{\Sigma}_\theta^{\text{mix}}$ | **0.35** ($\pm$ 0.04) | **4.91** ($\pm$ 0.54) | **0.20** ($\pm$ 0.02) | -4.08 ($\pm$ 0.09) | 1.00 ($\pm$ 0.00) |
| | $\boldsymbol{\Sigma}_\theta^{\text{mv}}$ | 0.49 ($\pm$ 0.01) | 7.68 ($\pm$ 0.08) | 0.34 ($\pm$ 0.01) | -3.59 ($\pm$ 0.02) | 0.99 ($\pm$ 0.00) |
| | $\boldsymbol{\epsilon}_t^{\text{ES}}$ | 0.59 ($\pm$ 0.07) | 7.34 ($\pm$ 1.51) | 0.34 ($\pm$ 0.08) | -3.53 ($\pm$ 0.28) | **0.98** ($\pm$ 0.01) |
| | iDDPM | 0.53 ($\pm$ 0.07) | 5.56 ($\pm$ 0.59) | 0.22 ($\pm$ 0.02) | **-4.13** ($\pm$ 0.06) | 0.86 ($\pm$ 0.03) |

## G  CAPTURING EPISTEMIC UNCERTAINTY

When considering predictive uncertainty, one typically considers two sources of uncertainty (Hüllermeier & Waegeman, 2021): *epistemic uncertainty* (EU) and *aleatoric uncertainty* (AU). While aleatoric uncertainty describes the inherent randomness in the data-generating process, for example, due to measurement errors and is often referred to as *irreducible* uncertainty, epistemic uncertainty arises from a lack of knowledge or information about the data-generating process and is also referred to as *reducible* uncertainty. While aleatoric uncertainty is naturally represented by a probability distribution, epistemic uncertainty usually requires a higher-order representation, such

as a second-order distribution (distribution of a distribution). Although some works propose specialized diffusion architectures for estimation of EU(Berry et al., 2024; Shu & Farimani, 2024), standard diffusion models generally lack such access.

Our method addresses this by introducing an additional distribution $p_\theta^\epsilon(\epsilon_t \mid \boldsymbol{x}_t)$ over the noise variable $\epsilon_t$, which serves as a second-order distribution and enables direct estimation of epistemic uncertainty. Recall that for DDPMs, we have

$$p_\theta(\boldsymbol{x}_{t-1} \mid \boldsymbol{x}_t) = \mathcal{N}(\boldsymbol{x}_{t-1}; \boldsymbol{\mu}_\theta(\boldsymbol{x}_t, t), \boldsymbol{\Sigma}_\theta(\boldsymbol{x}_t, t)), \tag{57}$$

with $\boldsymbol{\mu}_\theta(\boldsymbol{x}_t, t) = \frac{1}{\sqrt{\alpha_t}}\left(\boldsymbol{x}_t - \frac{1-\alpha_t}{\sqrt{1-\bar{\alpha}_t}}\boldsymbol{\epsilon}_\theta(\boldsymbol{x}_t, t)\right)$ and a fixed diagonal $\boldsymbol{\Sigma}_\theta(\boldsymbol{x}_t, t)$. Here, $\boldsymbol{\epsilon}_\theta(\boldsymbol{x}_t, t)$ is a deterministic denoiser. Iterative sampling then leads to the (approximate) predictive conditional distribution $p_{\mathcal{Y}}(\cdot \mid \boldsymbol{c})$, from which aleatoric uncertainty can be assessed (Shu & Farimani, 2024). Furthermore, by modeling $p_\theta^\epsilon(\epsilon_t \mid \boldsymbol{x}_t)$, our approach induces a distribution over $\boldsymbol{\mu}_\theta(\boldsymbol{x}_t, t)$ and hence a second-order distribution over the Gaussian mean in the transition kernel in (57).

As an example, consider the diagonal covariance approximation, $\boldsymbol{\Sigma}_\theta^{\mathrm{diag}}$. In this case, EU can be estimated using the variance of the predictive mean:

$$\underbrace{\mathbb{V}[\mathbb{E}[X]]}_{\mathrm{EU}} = \mathbb{V}[\boldsymbol{\mu}_\theta(\boldsymbol{x}_t, t)] \propto \mathrm{diag}(\sigma_\theta^2(\boldsymbol{x}_t, t)).$$

Here, we focus on marginal epistemic uncertainty, i.e., one estimate per dimension ($i = 1, \ldots, D$), and average across timesteps $T$ to obtain a single value for EU. With this construction, both AU and EU are expressed in the same space as the prediction target, provided that the diffusion process and input/output domain coincide.

Figure 17 shows selected sample predictions and their corresponding uncertainty estimates for the three autoregressive regression tasks. For the Kuramoto–Sivashinsky equation, AU grows over time—consistent with its chaotic dynamics—while EU correlates with different solution branches, especially at early timesteps. In the following, we provide a quantitative evaluation of the uncertainty decomposition. Due to the chaotic nature of the KS equation, the systems become unpredictable after some timesteps. Aleatoric uncertainty should be significantly higher after that time, while epistemic uncertainty should not be affected.

For that purpose, we compare the (Spearman-) correlation between the mean prediction and the true trajectory, averaged over the test dataset, to the measures of AU and EU. Figure 18 shows that the model behaves as expected: With decreasing predictability, measured by the correlation, the aleatoric uncertainty increases, while epistemic uncertainty remains unaffected.

For the Burgers' equation, both uncertainties remain small but vary with spatial location. Finally, for surface temperature prediction, AU is elevated in high-altitude regions due to fine-scale variability, while EU highlights these regions even more strongly, with large values over the Alps where predicted temperatures are low. This suggests the model is less certain in these regimes, which could be mitigated by using additional training data.

Overall, our model yields sensible EU estimates, but further theoretical and empirical study is needed—for example, exploring timestep aggregation strategies or alternative EU measures. These remain interesting directions for future work.

Figure 17: Estimates of aleatoric and epistemic uncertainty for the $\Sigma_\theta^{\mathrm{diag}}$ model for the three different autoregressive prediction tasks.

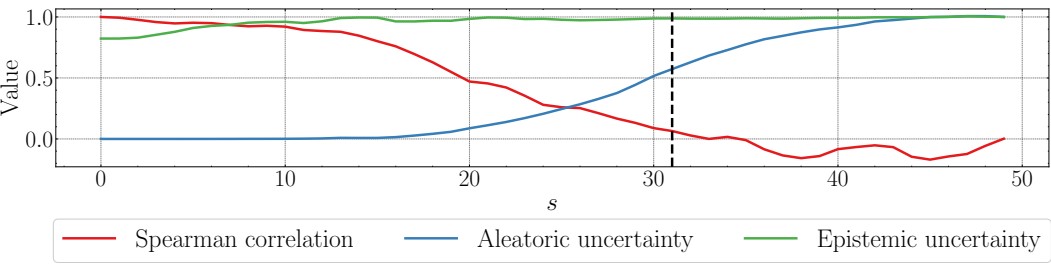

Figure 18: The figure shows AU, EU and the Spearman correlation between the mean prediction and true trajectory, across the PDE timesteps $s$. For visibility purposes, AU and EU are normalized, as they are different by order of magnitude.

## H  HYPERPARAMETERS

Due to the high computational cost of the models, we are not able to provide an extensive statistical evaluation of all hyperparameters. However, since the proposed methods mainly affect the diffusion process, we can use the same model backbone for all methods and therefore still make them comparable. For some of the proposed methods, though, we introduce additional hyperparameters that need to be tuned. Therefore, we provide some minor hyperparameter evaluation for the prediction task of the Burgers' and the Kuramoto–Sivashinsky equations. Table 14 shows the performance of the mixed normal model in dependence on the number of components $K$, Table 15 shows the performance of the sample-based model in dependence on the number of samples $M$, and Table 16 shows the performance of the multivariate normal model in dependence on the covariance approximation and the kernel bandwidth $\gamma$. Each table shows the different metrics, as well as the average training time of one epoch. The training process is similar to the main experiment, but with more strict early stopping, to remain computationally feasible. All metrics are evaluated on an (unseen) validation set, and as a selection criterion, we choose the energy score.

Table 14: Effect of the number of components in the (univariate) mixture normal model on the performance on the validation set. For readability, the metrics RMSE, ES, and CRPS are scaled by the factor 100 (10) for the Burgers' (KS) data.

| Experiment | $K$ | $t_{epoch}[s]$ | RMSE↓ | ES↓ | CRPS↓ | NLL↓ | $\mathcal{C}_{0.95}$ |
|---|---|---|---|---|---|---|---|
| **Burgers'** | 2 | 7.42 (0.18) | **0.0680** | **0.4790** | **0.0170** | -6.6114 | 0.9996 |
| | 3 | 7.40 (0.16) | 0.1450 | 0.8400 | 0.0280 | -6.2805 | 0.9993 |
| | 5 | 7.44 (0.16) | 0.0710 | 0.5030 | 0.0170 | **-6.6847** | **0.9992** |
| | 10 | 7.46 (0.17) | 0.1040 | 0.5960 | 0.0200 | -6.5018 | 0.9993 |
| | 25 | 7.61 (0.14) | 0.0730 | 0.5230 | 0.0180 | -6.5867 | 0.9995 |
| | 50 | 10.71 (0.22) | 0.0800 | 0.5210 | 0.0180 | -6.6183 | 0.9996 |
| **KS** | 2 | 7.20 (0.18) | 0.0432 | 0.6043 | 0.0240 | -3.9073 | 0.9998 |
| | 3 | **7.12 (0.15)** | 0.0545 | 0.7386 | 0.0301 | -3.6915 | 0.9997 |
| | 5 | 7.39 (0.19) | 0.0640 | 0.8095 | 0.0328 | -3.6345 | **0.9995** |
| | 10 | 7.50 (0.19) | 0.0446 | 0.6356 | 0.0264 | -3.7839 | 0.9997 |
| | 25 | 7.65 (0.16) | 0.0446 | 0.6557 | 0.0265 | -3.7612 | 0.9999 |
| | 50 | 10.78 (0.18) | **0.0409** | **0.5790** | **0.0237** | **-3.9075** | 0.9998 |

Table 15: Effect of the number of samples drawn in the training of the sample-based model on the performance on the validation set. For readability, the metrics RMSE, ES, and CRPS are scaled by the factor 100 (10) for the Burgers' (KS) data.

| Experiment | $M$ | $t_{epoch}[s]$ | RMSE↓ | ES↓ | CRPS↓ | NLL↓ | $\mathcal{C}_{0.95}$ |
|---|---|---|---|---|---|---|---|
| **Burgers'** | 3 | **10.46 (0.25)** | 0.1020 | 0.4870 | **0.0190** | -6.6803 | 0.9912 |
| | 5 | 15.65 (0.28) | 0.1840 | 0.8440 | 0.0300 | -6.4734 | **0.9878** |
| | 10 | 30.18 (0.45) | 0.1610 | 0.6250 | 0.0240 | -6.5091 | 0.9939 |
| | 25 | 71.31 (0.31) | **0.0860** | **0.4560** | **0.0190** | -6.6150 | 0.9957 |
| **KS** | 3 | **10.47 (0.24)** | **0.0618** | **0.6824** | **0.0304** | **-3.6989** | **0.9700** |
| | 5 | 15.68 (0.20) | 0.0668 | 0.7579 | 0.0347 | -3.5424 | 0.9785 |
| | 10 | 29.82 (0.44) | 0.2015 | 2.1613 | 0.0939 | -2.7003 | 0.9105 |
| | 25 | 72.56 (0.33) | 0.0635 | 0.7364 | 0.0346 | -3.5335 | 0.9731 |

In a second stage, given the previous optimal hyperparameters, we tested for each model, whether performance would improve when changing the parameter $\beta_T$ of the noise scheduler. We tested two different values, based on recent studies; $\beta_T = 0.35$ (Han et al., 2022) and $\beta_T = 0.2$ (Kohl et al., 2024). Each model was run for two different seeds, the averaged results can be found in Table 17. As for the weather forecasting task hyperparameter optimization is computationally too demanding, we choose values based on the previous results, selecting values based on a trade-off between performance and computational complexity.

Table 16: Effect of the covariance matrix approximation and the kernel bandwidth $\gamma$ on the performance on the validation set. The numerical values for $R$ denote the rank in the low-rank approximation, while C denotes the full Cholesky approximation. For the KS data, training with $\gamma = 5$ did not converge. For readability, the metrics RMSE, ES, and CRPS are scaled by the factor 100 (10) for the Burgers' (KS) data.

| Experiment | $\gamma$ | $R$ | $t_{epoch}[s]$ | RMSE↓ | ES↓ | CRPS↓ | NLL↓ | $\mathcal{C}_{0.95}$ |
|---|---|---|---|---|---|---|---|---|
| **Burgers'** | 5 | 1 | 7.39 (0.34) | **0.0480** | **0.7040** | **0.0270** | **-5.8702** | 0.9979 |
| | | 5 | 7.56 (0.11) | 0.1470 | 1.1290 | 0.0480 | -5.4215 | 0.9991 |
| | | 10 | 7.64 (0.20) | 0.2320 | 1.5920 | 0.0630 | -5.3654 | 0.9961 |
| | | 25 | 7.57 (0.14) | 0.3130 | 1.9480 | 0.0700 | -5.3428 | 0.9967 |
| | | 50 | 7.28 (0.37) | 0.0790 | 0.9950 | 0.0390 | -5.6229 | 0.9995 |
| | | C | 9.10 (0.22) | 0.1650 | 5.0860 | 0.1710 | -4.9388 | 0.9937 |
| | 10 | 1 | 7.57 (0.45) | 0.0600 | 0.8280 | 0.0330 | -5.7380 | 0.9977 |
| | | 5 | 7.76 (0.25) | 0.0600 | 0.9130 | 0.0420 | -5.4361 | 0.9996 |
| | | 10 | 7.66 (0.24) | 0.6410 | 3.4170 | 0.1230 | -4.7906 | 0.9938 |
| | | 25 | 7.92 (0.27) | 0.0630 | 0.9490 | 0.0370 | -5.6301 | 0.9997 |
| | | 50 | 7.17 (0.40) | 0.0790 | 0.9790 | 0.0380 | -5.6346 | 0.9996 |
| | | C | 9.02 (0.17) | 0.1790 | 2.9850 | 0.0810 | -5.3116 | **0.9859** |
| | 25 | 1 | 7.62 (0.18) | 0.0970 | 0.9870 | 0.0410 | -5.5773 | 0.9952 |
| | | 5 | **7.06 (0.19)** | 0.0990 | 1.0520 | 0.0470 | -5.3813 | 0.9985 |
| | | 10 | **7.06 (0.09)** | 0.0650 | 0.9450 | 0.0390 | -5.5638 | 0.9997 |
| | | 25 | 7.88 (0.71) | 0.1040 | 1.1440 | 0.0450 | -5.5535 | 0.9993 |
| | | 50 | 7.85 (0.62) | 0.1160 | 1.1010 | 0.0430 | -5.5743 | 0.9990 |
| | | C | 8.95 (0.08) | 0.2200 | 7.7430 | 0.2500 | -4.5096 | 0.9955 |
| **KS** | 10 | 1 | 7.51 (0.33) | 0.0505 | **0.7791** | **0.0340** | **-3.5624** | 0.9921 |
| | | 5 | 7.54 (0.32) | 0.6869 | 7.5898 | 0.3386 | -1.4127 | **0.9719** |
| | | 10 | 7.26 (0.29) | 0.0620 | 0.9294 | 0.0405 | -3.2973 | 0.9987 |
| | | 25 | 7.18 (0.16) | 0.0549 | 0.8870 | 0.0374 | -3.3786 | 0.9991 |
| | | 50 | 7.15 (0.12) | **0.0493** | 0.8546 | 0.0365 | -3.3813 | 0.9993 |
| | | C | 9.01 (0.10) | 0.1147 | 4.3415 | 0.1133 | -2.9118 | 0.9797 |
| | 25 | 1 | **7.13 (0.12)** | 0.2035 | 2.3734 | 0.1039 | -2.5079 | 0.9851 |
| | | 5 | 7.38 (0.34) | 0.0578 | 0.8802 | 0.0412 | -3.1876 | 0.9990 |
| | | 10 | 7.53 (0.34) | 0.0612 | 0.9406 | 0.0421 | -3.2333 | 0.9989 |
| | | 25 | 7.54 (0.32) | 0.0549 | 0.8918 | 0.0395 | -3.2972 | 0.9991 |
| | | 50 | 7.54 (0.32) | 0.0768 | 1.0469 | 0.0467 | -3.1729 | 0.9978 |
| | | C | 9.07 (0.11) | 0.1773 | 7.8864 | 0.2815 | -1.9158 | 0.9958 |

Table 17: For readability, the metrics RMSE, ES, and CRPS are scaled by the factor 100 (10) for the Burgers' (KS) data.

| Experiment | Model | $\beta_T$ | RMSE↓ | ES↓ | CRPS↓ | NLL↓ | $\mathcal{C}_{0.95}$ |
|---|---|---|---|---|---|---|---|
| **Burgers'** | $\boldsymbol{\delta}_\theta$ | 0.2 | 0.1856 | 0.9785 | 0.0371 | -6.2573 | 0.8240 |
| | | **0.35** | 0.0640 | 0.4133 | 0.0155 | -7.0553 | 0.9131 |
| | $\boldsymbol{\Sigma}_\theta^{\text{diag}}$ | **0.2** | 0.1098 | 0.7814 | 0.0314 | -6.1350 | 0.9944 |
| | | 0.35 | 0.1688 | 1.1605 | 0.0452 | -5.7837 | 0.9906 |
| | $\boldsymbol{\Sigma}_\theta^{\text{mix}}$ | **0.2** | 0.0581 | 0.4219 | 0.0146 | -6.8363 | 0.9994 |
| | | 0.35 | 0.0784 | 0.5105 | 0.0172 | -6.6333 | 0.9995 |
| | $\boldsymbol{\Sigma}_\theta^{\text{mv}}$ | **0.2** | 0.0524 | 0.6523 | 0.0255 | -6.0578 | 0.9970 |
| | | 0.35 | 0.0806 | 0.8573 | 0.0342 | -5.7823 | 0.9967 |
| | $\boldsymbol{\epsilon}_t^{\text{ES}}$ | **0.2** | 0.0637 | 0.3675 | 0.0148 | -6.8602 | 0.9903 |
| | | 0.35 | 0.0806 | 0.4386 | 0.0176 | -6.6754 | 0.9922 |
| **KS** | $\boldsymbol{\delta}_\theta$ | 0.2 | 0.1896 | 2.1152 | 0.0976 | -3.0426 | 0.8237 |
| | | **0.35** | 0.0636 | 0.6545 | 0.0263 | -3.9476 | 0.8643 |
| | $\boldsymbol{\Sigma}_\theta^{\text{diag}}$ | **0.2** | 0.0549 | 0.7443 | 0.0324 | -3.5836 | 0.9985 |
| | | 0.35 | 0.0718 | 0.9108 | 0.0390 | -3.4225 | 0.9982 |
| | $\boldsymbol{\Sigma}_\theta^{\text{mix}}$ | **0.2** | 0.0475 | 0.6448 | 0.0258 | -3.8172 | 0.9996 |
| | | 0.35 | 0.0556 | 0.7336 | 0.0300 | -3.6820 | 0.9995 |
| | $\boldsymbol{\Sigma}_\theta^{\text{mv}}$ | 0.2 | 0.0987 | 1.1955 | 0.0515 | -3.3161 | 0.9903 |
| | | **0.35** | 0.0584 | 0.8510 | 0.0376 | -3.5145 | 0.9893 |
| | $\boldsymbol{\epsilon}_t^{\text{ES}}$ | 0.2 | 0.1145 | 1.2534 | 0.0558 | -3.2355 | 0.9432 |
| | | **0.35** | 0.0814 | 0.9029 | 0.0410 | -3.4059 | 0.9620 |

## H.1 COMPARISON OF SCORING RULES

In principle, our framework allows for arbitrary scoring rule loss functions for training. While Bortoli et al. (2025) show that certain choices of $k$ for the kernel score lead to loss functions that can recover the original diffusion loss in the limit, they also show empirical findings that scoring rules without this property can lead to similar results. To further motivate our use of the energy (and kernel) score, we show a brief comparison against a very common scoring rule, the log-score (or negative log-likelihood), which is defined as

$$S_{\log}(p, \boldsymbol{y}) \coloneqq -\log p(\boldsymbol{y}),$$

for a probability density $p$. The log-score is commonly used to train neural networks for parametric distributions such as predictive Gaussians (Nix & Weigend, 1994; Lakshminarayanan et al., 2017). We compare $S_{\log}$ against the energy score $S_{\mathrm{ES}}$ for the two PDE prediction tasks and the $\boldsymbol{\Sigma}_\theta^{\mathrm{diag}}$ model, for which closed-form expressions are available for both loss functions. Table 18 shows that the energy score leads to significantly better results across all metrics for both PDEs. The performance of the model trained with $S_{\log}$ is up to a factor of 20 times worse, highlighting that training using the energy score can be advantageous, which goes along recent literature (Rasp & Lerch, 2018; Shen & Meinshausen, 2024; Alet et al., 2025).

Table 18: Comparison of different scoring rules for the $\boldsymbol{\Sigma}_\theta^{\mathrm{diag}}$ model. The best model is highlighted in bold and the standard deviation across different model runs is given in brackets.

| Experiment | $S$ | $t_{epoch}[s]$ | RMSE↓ | ES↓ | CRPS↓ | NLL↓ | $\mathcal{C}_{0.95}$ |
|---|---|---|---|---|---|---|---|
| **Burgers'** | ES | **6.81** ($\pm$ 0.16) | **0.81** ($\pm$ 0.27) | **3.67** ($\pm$ 0.39) | **0.12** ($\pm$ 0.02) | **-7.12** ($\pm$ 0.09) | 1.00 ($\pm$ 0.00) |
| | log | 7.25 ($\pm$ 0.22) | 17.31 ($\pm$ 6.47) | 65.88 ($\pm$ 28.98) | 1.32 ($\pm$ 0.52) | -5.79 ($\pm$ 0.28) | 1.00 ($\pm$ 0.00) |
| **KS** | ES | 7.54 ($\pm$ 0.28) | **0.39** ($\pm$ 0.06) | **5.23** ($\pm$ 0.75) | **0.21** ($\pm$ 0.03) | **-4.04** ($\pm$ 0.14) | 1.00 ($\pm$ 0.00) |
| | log | **7.47** ($\pm$ 0.66) | 2.11 ($\pm$ 1.57) | 23.19 ($\pm$ 15.69) | 0.50 ($\pm$ 0.14) | -3.35 ($\pm$ 0.23) | 1.00 ($\pm$ 0.00) |

