# OpenReview forum: "Improved probabilistic regression using diffusion models"
_ICLR.cc/2026/Conference — Submitted to ICLR 2026_

### Official Review · Reviewer_vvhz · 2025-10-31

**Soundness:** 3
**Presentation:** 3
**Contribution:** 3
**Rating:** 4
**Confidence:** 4

**Summary:**

To enable better uncertainty quantification and calibrated probabilistic predictions across regression tasks, this paper introduces a diffusion-based probabilistic regression framework that learns the full distribution of diffusion noise instead of predicting only its mean (as in standard DDPM/DDIM).

The key points of this paper includes:

1. Reformulate diffusion regression as learning the distribution $p_\epsilon(\epsilon_t | x_t)$, using strictly proper scoring rules (e.g., energy score or kernel score) to ensure consistency and calibration.
2. Propose several parameterizations of the diffusion noise distribution with:
- Univariate Gaussian
- Gaussian Mixture
- Multivariate Gaussian (low-rank + diagonal)
3. Demonstrate that this formulation yields closed-form reverse sampling, preserving the tractability of DDPM.
4. Empirically validate across UCI benchmarks, autoregressive prediction, and monocular depth estimation, showing consistent performance and better uncertainty calibration.

**Strengths:**

The paper is conceptually novel and rigororous, and it connects diffusion modeling with the broader literature on probabilistic scoring rules and distributional regression. The proposed method is versatile and general, which just need no or at least minimal architectural modification: existing diffusion backbones (e.g., DDPM, DDIM, U-Net) can be reused. Results across multiple domains support generality and robustness, and overall the paper is well-written, with clear derivations and equations linking the proposed loss, parametrizations, and reverse process formulation.

**Weaknesses:**

The paper still exposes several limitations, which is listed below and hope the authors could address:

1. While three variants are explored (univariate, mixture, multivariate), the paper lacks deeper insight into when and why each performs best. It would benefit if the authors provide with guidance to the readers on how to choose these parameterizations, maybe according to the data distribution?

2. Although claimed efficient, explicit runtime comparisons (e.g., against DDPM or nonparametric) are limited.

3. It remains unclear whether these advantages persist at scale (e.g., large diffusion backbones, complex regression scenarios).

4. The theoretical results and contribution are incremental relative to Bortoli et al. (2025) and is conceptually close to their work. Although the authors categorize it as concurrent work, Bortoli et al. (2025) appears in ICML 2025, which is hardly considered as concurrent. The paper could more sharply differentiate its contributions beyond being a computationally efficient parameterized alternative.

**Questions:**

1. Why specifically use the energy and kernel scores?

2. Are there any parameterization trade-offs, e.g., how does performance scale with K?

3. Any observed training instabilities not reported due to modeling the full noise distribution?

4. Could this approach extend naturally to discrete or hybrid data (e.g., language, tabular categorical features), like done in CARD (Han et al., 2022)?

---

> ### Author Response · Authors · 2025-11-21
>
> [1/3]
>
> We thank Reviewer vvhz for their detailed review and their helpful suggestions, which significantly improved the quality of our paper. We have uploaded a revised version that incorporates the reviewers’ feedback.
>
> **Weaknesses:**
>
> 1. **While three variants are explored (univariate, mixture, multivariate), the paper lacks deeper insight into when and why each performs best. It would benefit if the authors provide with guidance to the readers on how to choose these parameterizations, maybe according to the data distribution?**
>
> We agree that clearer practical guidance is helpful, and we have added some to the main text (compare line 477). Our results indicate that the univariate Gaussian variant—being hyperparameter-free—already performs strongly and often outperforms DDPM. Therefore, this provides a good default model choice.
> Choosing more expressive parameterizations based on data characteristics is reasonable and consistent with our observations. For example, in the depth-estimation task, the multivariate Normal performs best, likely due to spatial structure in the latent space. In contrast, for autoregressive residual modeling, much of the correlation is already encoded in the inputs, so additional parameterization seems to bring limited benefit. The mixture model also performs well, but is more sensitive to tuning (especially the choice of $K$).
> In summary, we recommend starting with the univariate Gaussian method and considering the multivariate or mixture variants when the data exhibit meaningful spatial dependencies. A systematic study of these relationships is an interesting direction for future work.
>
> 2. **Although claimed efficient, explicit runtime comparisons (e.g., against DDPM or nonparametric) are limited.**
>
> Running extensive efficiency ablations for computationally demanding models such as Marigold (~900M parameters) is challenging, so we initially provided only minor studies.
> To provide a comparison of the efficiency (and computational advantages) of the methods, we ran an additional detailed experiment for the PDE tasks. We added the results in Appendix D.1 and also include them in our comment that describes our revision changes, please compare for more details. Essentially, we find that there is no significant runtime difference between DDPM and our univariate and multivariate Normal method. For the univariate mixture small $K \approx 5$ also lead to the same runtime, while large $K \approx 50$ take significantly longer. The sample-based method by Bortoli et. al. (2025) has significantly higher computational cost, even for a small amount of samples.
>
> 3. **It remains unclear whether these advantages persist at scale (e.g., large diffusion backbones, complex regression scenarios).**
>
> Depth estimation and weather forecasting are both very complex regression tasks and recent progress relies heavily on large neural network architectures. For depth estimation, we use Marigold [1], which is based on the Stable Diffusion backbone with ~900M parameters, demonstrating that our method scales and leads to good performance, even with large diffusion backbones.
>
> 5. **The theoretical results and contribution are incremental relative to Bortoli et al. (2025) and is conceptually close to their work. Although the authors categorize it as concurrent work, Bortoli et al. (2025) appears in ICML 2025, which is hardly considered as concurrent. The paper could more sharply differentiate its contributions beyond being a computationally efficient parameterized alternative.**
>
> That is a great point. We believe our work differs from Bortoli et. al. (2025) [2] (ICML, July 2025) in several ways:
>
> Their work focuses on generative tasks and is motivated by accelerating the diffusion sampling process.
> In contrast, our work is motivated by using scoring rules as (distributional) loss functions to improve predictive performance and uncertainty quantification in high-dimensional conditional regression tasks.
> While both approaches make use of scoring rules—and this connection is indeed not novel—our contribution is not merely a parametrized instantiation of their framework. Beyond proposing different noise approximations and providing an analytical form of the sampling distribution, we empirically show improved performance over Bortoli et al. (2025) across all benchmark tasks considered.
>
> Therefore, the main difference is our focus on deriving the scoring rule models as to improve conditional regression objectives and corresponding uncertainty quantification, which requires a different approach than that of Bortoli et. al. (2025), which focuses on computational sampling efficiency. We have revised the corresponding sections in the main text to make this distinction clearer

---

> > ### Author Response · Authors · 2025-11-21
> >
> > [2/3]
> >
> > **Questions:**
> >
> > 1. **Why specifically use the energy and kernel scores?**
> >
> > Thank you for the question. Kernel scores (including the energy score as a special case) are strictly proper for our chosen kernels and correspond directly to the Maximum Mean Discrepancy (MMD) [1], a well-studied divergence with useful theoretical guarantees [2].
> > Compared to the log-score, kernel scores place fewer assumptions on the underlying distributions and remain well-defined for degenerate cases, improving training stability (e.g., no issues from variance collapse), which is consistent with our empirical comparison to the log-score in Appendix H.1.
> > Moreover, recent work [3] shows that both the energy score and Gaussian kernel score recover the classical diffusion loss in the distributional diffusion framework, providing a unified view in diffusion settings.
> > Finally, they offer simple unbiased estimators and closed-form expressions for the predictive families we consider (Normal, Normal-mixture, multivariate Normal), as shown in Appendix A.2.
> >
> >
> > 2. **Are there any parameterization trade-offs, e.g., how does performance scale with K?**
> >
> > Yes. Our hyperparameter search (Table 11) shows that while small $K$ can work well, larger $K$ often improves performance (e.g., for KS), consistent with findings in MDNs [4]. However, using the CRPS introduces a computational cost of $\mathcal{O}(K^2)$, making very large $K$ more expensive despite improving stability compared to the log-score. Overall, selecting $K$ remains task-dependent and is an open problem in MDNs and Gaussian mixture approaches more generally.
> >
> > 3. **Any observed training instabilities not reported due to modeling the full noise distribution?**
> >
> > Indeed, modeling the full noise distribution with a multivariate Normal can introduce numerical instabilities. All relevant issues appear in Appendix A and Appendix H, but we summarize them here: Using the Cholesky and LoRA parameterizations already avoids many instabilities of full covariance modeling, yet certain hyperparameter settings still fail to converge—for example, specific kernel bandwidth $\gamma$ choices or combinations such as  $(\gamma=10, R=10)$ for Burgers’ and ($\gamma=10, R=5$) for KS (Table 13). Furthermore, the full Cholesky parameterization, while convergent, performs significantly worse than LoRA—likely due to its substantially larger parameterization, linking back to the trade-offs discussed above.
> >
> > 4. **Could this approach extend naturally to discrete or hybrid data (e.g., language, tabular categorical features), like done in CARD (Han et al., 2022)?**
> >
> > Great question. Since we already implement CARD [5] and evaluate on the corresponding UCI benchmarks, we demonstrate that our model performs well on tabular data, (including categorical features), see Table 1 for comparison. For language models, the extension highly depends on how diffusion is performed. If the diffusion process is defined in a continuous latent space (of tokens or text), our method can be applied directly. If discrete diffusion over the tokens is used [6], the model already learns the full distribution over each token. Whether scoring rule training could lead to further improvements is not directly clear, but would be a very interesting avenue of future research.

---

> > > ### Author Response · Authors · 2025-11-21
> > >
> > > [3/3]
> > >
> > > **References**
> > >
> > > [1] Ke, B., Obukhov, A., Huang, S., Metzger, N., Daudt, R.C., Schindler, K., 2024. Repurposing Diffusion-Based Image Generators for Monocular Depth Estimation. [https://doi.org/10.48550/arXiv.2312.02145](https://doi.org/10.48550/arXiv.2312.02145)
> > >
> > > [2] Bortoli, V.D., Galashov, A., Guntupalli, J.S., Zhou, G., Murphy, K.P., Gretton, A., Doucet, A., 2025. Distributional diffusion models with scoring rules, in: Forty-Second International Conference on Machine Learning.
> > >
> > > [3] Gretton, Arthur, Karsten M. Borgwardt, Malte J. Rasch, Bernhard Schölkopf, and Alexander Smola. “A Kernel Two-Sample Test.” _The Journal of Machine Learning Research_ 13, no. null (2012): 723–73.
> > >
> > > [4] Waghmare, Kartik, and Johanna Ziegel. “Proper Scoring Rules for Estimation and Forecast Evaluation.” arXiv:2504.01781. Preprint, arXiv, April 2, 2025. [https://doi.org/10.48550/arXiv.2504.01781](https://doi.org/10.48550/arXiv.2504.01781).
> > >
> > > [5] Bortoli, Valentin De, Alexandre Galashov, J Swaroop Guntupalli, et al. "Distributional Diffusion Models with Scoring Rules." *The Forty-second International Conference on Machine Learning*, 2025.
> > >
> > > [6] Kelen, Domokos M., Ádám Jung, Péter Kersch, and Andras A. Benczur. “Distribution-Free Data Uncertainty for Neural Network Regression.” Paper presented at The Thirteenth International Conference on Learning Representations. October 4, 2024. [https://openreview.net/forum?id=pDDODPtpx9](https://openreview.net/forum?id=pDDODPtpx9).
> > >
> > > [7] Han, X., Zheng, H., Zhou, M., 2022. CARD: Classification and Regression Diffusion Models. [https://doi.org/10.48550/arXiv.2206.07275](https://doi.org/10.48550/arXiv.2206.07275)
> > >
> > > [8] Lou, A., Meng, C., Ermon, S., 2024. Discrete Diffusion Modeling by Estimating the Ratios of the Data Distribution. [https://doi.org/10.48550/arXiv.2310.16834](https://doi.org/10.48550/arXiv.2310.16834)

---

> ### Comment · Reviewer_vvhz · 2025-11-28
> **Thank you for the response**
>
> I appreciate the authors providing detailed point-by-point responses in addressing my questions. I will update my score and I'd appreciate if the authors could incorprate these points into the revision and final version, e.g., iDDPM results, distinction compared to Bortoli et. al. (2025), the risk of training instabilities.

---

> > ### Author Response · Authors · 2025-11-28
> >
> > We thank the reviewer for their feedback and their willingness to raise the score. We have uploaded a revised version in which we (i) discuss the training instabilities of the multivariate normal method in Appendix A.1, (ii) clarify the connection and differences to Bortoli et al. (2025) in the Conclusion, (iii) highlight the scalability of our approach to large diffusion backbones (e.g., Stable Diffusion) in the conclusion, and (iv) justify our use of the energy score and Gaussian kernel score in Section 3.2.
> >
> > Regarding the iDDPM results: the KS equation results are already included in Appendix F and referenced in the main text. We will add the full iDDPM results for all datasets to the camera-ready version (due to time and compute constraints).

---

### Official Review · Reviewer_wwWQ · 2025-11-01

**Soundness:** 3
**Presentation:** 3
**Contribution:** 2
**Rating:** 4
**Confidence:** 3

**Summary:**

The paper proposes a diffusion based framework for probabilistic regression that, instead of predicting only a point estimate, learns the full conditional distribution of the per step diffusion noise using strictly proper scoring rules. Concretely, it parameterizes the noise distribution (e.g., diagonal Gaussian, Gaussian mixtures, and multivariate Gaussian with efficient approximations) so the model can balance expressivity and compute, while still admitting closed form reverse transitions for straightforward sampling. Across diverse tasks—UCI tabular regression, autoregressive flow/weather prediction, and monocular depth—the approach reports better predictive accuracy and notably improved calibration/uncertainty estimates compared to standard diffusion baselines. Overall, it reframes diffusion regression as learning a flexible noise distribution to produce calibrated predictive distributions end to end.

**Strengths:**

1. Flexible noise modeling with proper scoring rules yields calibrated predictive distributions instead of mere point estimates.
2. Closed-form reverse transitions allow efficient sampling and easy integration with standard diffusion samplers.
3. Modular parameterizations (diag Gaussian, mixtures, multivariate/low-rank) trade off accuracy vs. compute without redesigning the pipeline.
4. Broad empirical scope (tabular, autoregressive flow/weather, depth) shows consistent CRPS/calibration gains over diffusion baselines.

**Weaknesses:**

1. The paper's core premise—learning the full noise distribution $ p_{\theta}^{\epsilon}(\cdot|x_t) $ via proper scoring rules—was concurrently proposed by Bortoli et al. (2025). This work's primary contribution is thus the specific instantiation with (mixture) Gaussian heads. This is further narrowed by the fact that the simplest case (univariate Gaussian) is, as the authors note, conceptually equivalent to prior work on variance learning.

2. The reported improvements are inconsistent and, in some cases, negligible. On UCI benchmarks, CRPS gains vary widely, from substantial (e.g., Naval: $ -32 % $) to nonexistent (e.g., Wine: $ 0 % $). The method can also underperform the baseline in RMSE on some datasets (e.g., Yacht). Likewise, on depth estimation, the CRPS gains are marginal (e.g., $ \approx -0.45 % $ on KITTI, $ \approx -2.5\% $ on DIODE) and even show a slight regression on ETH3D ($ \approx +0.7\% $). This mixed evidence weakens the claim of general applicability.

3.  The method's calibration claims are undermined by the reliance on a post-hoc hyperparameter, $ \tau $. The authors concede the model is "over-conservative" and that achieving near-nominal coverage—as well as the largest CRPS/ES gains—requires a small $ \tau \approx 0.05 $. This is a significant, unprincipled deviation from the underlying DDIM framework, and no method for selecting this critical parameter is proposed.

4. The empirical evaluation lacks the rigor expected for a high-impact venue. The authors admit to only "minor hyperparameter tuning" for comparators and an inability to conduct an "extensive statistical evaluation" of their own hyperparameters due to cost. Furthermore, computational comparisons are dismissed as "rough estimates". Without a rigorous and fair benchmark, the reported performance margins are difficult to interpret confidently.

**Questions:**

See the weakness section and the following:

1.  Given the method is demonstrably "over-conservative" at $\tau=1$, how can the reliance on an unprincipled, post-hoc $\tau \approx 0.05$ to correct calibration be justified? Can the authors provide a principled selection rule for $\tau$ and re-evaluate all key metrics using it, rather than presenting ad-hoc results?

2.  How do the authors explain the highly variable, and in some cases negligible or negative, empirical gains (e.g., 0% CRPS on Wine, RMSE regression on Yacht, marginal/negative CRPS on KITTI/ETH3D)? Can rigorous ablations be provided to demonstrate that these gains are not mere artifacts of added parameters, especially for the depth estimation tasks?

3.  Given the admission of "minor hyperparameter tuning" and "not optimized" timing, how can the claims of superiority be validated? Can the authors strengthen the paper by providing comparisons against properly tuned, strong UQ baselines (e.g., variance-learning DDPMs, MDNs) under a fair, matched-compute framework?

4.  How can the framework be considered "principled" when it currently lacks practical guidance for model selection and the claims of epistemic uncertainty separation are purely qualitative? Can the authors substantiate these claims by providing a quantitative heuristic for head selection and either a formal analysis or a quantitative benchmark for the UQ decomposition?

---

> ### Author Response · Authors · 2025-11-21
>
> [1/4]
>
> We thank Reviewer wwWQ for their detailed review and help in improving our paper. We uploaded a revised version of the paper to follow the ideas and recommendations of the reviewers.
>
> ### Weaknesses
>
> 1. **The paper's core premise—learning the full noise distribution via proper scoring rules—was concurrently proposed by Bortoli et al. (2025). This work's primary contribution is thus the specific instantiation with (mixture) Gaussian heads. This is further narrowed by the fact that the simplest case (univariate Gaussian) is, as the authors note, conceptually equivalent to prior work on variance learning.**
>
> Thank you for bringing this up. First, we want to discuss our difference with De Bortoli et al.: De Bortoli et al. only consider the generative setting and focus on reducing the number of time steps, while we work in the probabilistic regression setting and want to improve the (distributional) performance; reducing the number of diffusion sampling time steps is a nice property, but not our motivation. Further, we derive closed-form expressions for several of our parameterizations combined with specific scoring rules, which make training on a single sample possible in the first place. On the other hand, the method proposed by De Bortoli et al. requires multiple forward processes in order to compute the scoring rule, which increases the training time and compute significantly, making it unusable for complicated tasks.
>
> Second, we want to clarify that even in the simplest case (univariate Gaussian), we do not reduce to prior work on variance learning, as our loss is still very different and is an important part of our work. We realized that this is not presented clearly enough and have adapted our paper. Thank you very much for pointing this out.
>
> 2. **The reported improvements are inconsistent and, in some cases, negligible. On UCI benchmarks, CRPS gains vary widely, from substantial (e.g., Naval: ) to nonexistent (e.g., Wine: ). The method can also underperform the baseline in RMSE on some datasets (e.g., Yacht). Likewise, on depth estimation, the CRPS gains are marginal (e.g.,  on KITTI,  on DIODE) and even show a slight regression on ETH3D. This mixed evidence weakens the claim of general applicability.**
>
> This is an important question.
> - Regarding UCI benchmarks: Yes the gains vary; however, this is not a unique artifact of our methods, but can also be observed elsewhere in the literature, for example, in the original paper on CARD [1] .
> - Regarding Depth Estimation: Indeed, our method is slightly worse than the baseline in terms of CRPS in the ETH3D dataset; however, we want to stress that it beats the baseline in all other nine instances of our experiments.
> - In general, our methods are applicable where standard DPMs are and generally improve upon those. To undermine this claim, we follow the recommendations from  Demsar [6] and run a sign test in order to assess the statistical significance of our results, as shown in Appendix D.2. This supports our claim that representing $\epsilon_t$ by a parameterized distribution significantly improves upon the baseline. Thank you for bringing this to our attention, we believe that this is an important analysis.

---

> > ### Author Response · Authors · 2025-11-21
> >
> > [2/4]
> >
> > 3. **The method's calibration claims are undermined by the reliance on a post-hoc hyperparameter tau. The authors concede the model is "over-conservative" and that achieving near-nominal coverage—as well as the largest CRPS/ES gains—requires a small tau. This is a significant, unprincipled deviation from the underlying DDIM framework, and no method for selecting this critical parameter is proposed.**
> >
> > This is a very important remark. We have observed the $\tau$-scaling phenomenon during our experiments and wanted to share them with the research community.
> > Let us try to address your concern by stating the following:
> > - All of our results, as stated in the main part of our paper, are **without** finetuning $\tau$.
> > - We have to admit that this $\tau$ might not fit perfectly into the proposed framework of ours. However, we believe this can be justified as
> > 	- $\tau$ scaling simply controls the stochasticity in the sampling of our method. The effects of it can be theoretically explained by the fact that in/decreasing the variance of the transition distribution in/decreases the final output variance of the diffusion process. Why we see that the functions of CRPS or energy score in terms of $\tau$ sometimes have a minimum at $\tau \approx 0.05$  is something that we do not understand yet and want to investigate in future work.
> > 	- We also want to refer to the paper by Karras et al. [2], specifically to the section about Stochastic Sampling. There, the authors arrive at the conclusion that "the optimal amount of stochasticity should be determined empirically" and "on a case-by-case basis", highlighting the complicated role of stochasticity in diffusion models as it improves empirical performance, but no general applicable theory is known. Indeed, Karras et al [2] used a grid search to find the optimal sampling parameters regarding stochasticity and reported that they obtained completely different values depending on the dataset and the diffusion model.
> > - As argued above, $\tau$ can only be selected post-hoc, since it does not appear during training and is a parameter that can heavily depend on the underlying task, similar to other diffusion-process parameters [2]. The optimal $\tau$ is best chosen using a held-out validation set and an appropriate decision criterion. This is precisely what we already propose in Appendix E: Select $\tau$ based on minimizing (for example) the energy score as a selection criterion. As this does not require additional training, but only inference of the diffusion model, the selection process is computationally inexpensive. We are happy to elaborate on this in the paper if you believe it would strengthen the presentation.
> >
> > 4. **The empirical evaluation lacks the rigor expected for a high-impact venue. The authors admit to only "minor hyperparameter tuning" for comparators and an inability to conduct an "extensive statistical evaluation" of their own hyperparameters due to cost. Furthermore, computational comparisons are dismissed as "rough estimates". Without a rigorous and fair benchmark, the reported performance margins are difficult to interpret confidently.**
> >
> > We agree that a rigorous and fair benchmark is essential for good academic work. With that being said, we believe we can elaborate to clarify how we make sure that our comparisons do follow these principles. First, we admit that we do only "minor hyperparameter tuning" for **our distributional methods**. In particular, we investigate and tune the hyperparameters that are being introduced with our parameterizations, i.e., the rank of the covariance for the multivariate Gaussian and the number of mixtures for the mixture of Gaussians, but take the training, model, and diffusion hyperparameters from existing models in the literature (SongUNet, CARD, Marigold). Our framework is designed to be applicable whenever one has a diffusion model in a regression task.
> > More particularly:
> > - For the UCI experiments, we use CARD [1] as our baseline, which was optimized for these datasets, and we train with the same configuration as in CARD.
> > - Regarding the depth estimation experiments, the situation is the same, where we compare to the Marigold model as a baseline. We take the same setup, hyperparameters, and compute budget in order to obtain fair and comparable results
> > - We also want to clarify what we mean when we write that "we are not able to provide an extensive statistical evaluation". This only refers to the hyperparameters of our methods; as already mentioned, we build on top of established (and fine-tuned) models in the literature and do not "extensively" tune our backbone model, diffusion solvers, or sampling schemes to our parameterizations. We believe it is also more interesting to see that our models are able to improve upon finetuned baselines, without extensive hyperparameter search.

---

> ### Author Response · Authors · 2025-11-21
>
> [3/4]
>
> Regarding the "rough estimates" of the computation times, you are right. We have added Appendix D.1, addressing the runtime of our methods more carefully. Our results show that while our method leads to improved performance, it comes at no significantly increased computational cost (compared to standard diffusion models), except for very highly parameterized mixture models.
>
> ### Questions
>
> 1. **Given the method is demonstrably "over-conservative" at $\tau = 1$, how can the reliance on an unprincipled, post-hoc to correct calibration be justified? Can the authors provide a principled selection rule for and re-evaluate all key metrics using it, rather than presenting ad-hoc results?**
>
> We agree that the confidence intervals are over-conservative—though mainly for the two PDE tasks, not for UCI or T2M—and this may indicate suboptimal quantile calibration.
> However, the other evaluation metrics also assess distributional fit and thus measure calibration quality. For instance, the CRPS can be written as the integral of the quantile loss across all quantile levels [3], therefore assessing average quantile calibration over all quantile levels. In contrast, our defined coverave focuses solely on the commonly-used 95% interval. Likewise, [4] argue that the energy score or other proper scoring rules can be used as a summary statistic to assess calibration.
> This also touches on the broader question of what “calibration’’ should mean in high-dimensional or distributional settings, which remains an active research area with different notions of calibration [5].
> We also refer to our answer to weakness 3, especially regarding the "principled selection rule".
>
>
> 2. **How do the authors explain the highly variable, and in some cases negligible or negative, empirical gains (e.g., 0% CRPS on Wine, RMSE regression on Yacht, marginal/negative CRPS on KITTI/ETH3D)? Can rigorous ablations be provided to demonstrate that these gains are not mere artifacts of added parameters, especially for the depth estimation tasks?**
>
> For the explanations of the variable empirical gains, please refer to weaknesses 2 and 4.
> Regarding the question of whether the gains are only artifacts of added parameters, we want to highlight that the Stable Diffusion UNet has more than 800 million trainable parameters. By adding one more output head, such as in the univariate Gaussian parameterization, we are only adding around 10,000 parameters to the model and already improve the performance.
> What rigorous ablation studies do you have in mind? We are currently working on training a model that has the same structure as our backbones but averages the outputs of the different heads. This way, we have a classical model with the same number of parameters as our distributional methods.
>
> 3. **Given the admission of "minor hyperparameter tuning" and "not optimized" timing, how can the claims of superiority be validated? Can the authors strengthen the paper by providing comparisons against properly tuned, strong UQ baselines (e.g., variance-learning DDPMs, MDNs) under a fair, matched-compute framework?**
>
> Please refer to weakness 4 regarding the "minor hyperparameter tuning" and the timing issue.
> Thank you for providing suggestions on how we can strengthen our paper. We agree with you and are grateful for your idea of comparing to variance-learning DDPMs. For this reason, we use improved DDPM (iDDPM) [7] as it is an established diffusion model, that learns the variance as well. Furthermore, it is of high interest due ot the similarity of the architecture to our univariate Gaussian. They differ in their loss function (see Weakness 1), so this experiment also works as a kind of ablation study for the loss function. We report the results on the KS equation in Appendix F, which shows that iDDPM is superior to the previous baseline; however, it underperforms against our univariate and mixture models.

---

> > ### Author Response · Authors · 2025-11-21
> >
> > [4/4]
> >
> > 4. **How can the framework be considered "principled" when it currently lacks practical guidance for model selection and the claims of epistemic uncertainty separation are purely qualitative? Can the authors substantiate these claims by providing a quantitative heuristic for head selection and either a formal analysis or a quantitative benchmark for the UQ decomposition?**
> >
> > By principled, we are referring to our derivation of using scoring rules as loss functions as well as the derivation of the closed-form sampling formulas using our proposed parameterizations of the distribution $p(\epsilon_t \mid x_t)$.
> > Regarding the model selection: It is difficult to come up with a quantitative rule, but we can share our heuristic for deciding what method to use.
> > - The univariate Gaussian is a hyperparameter-free method that provides solid results out of the box
> > - We can see that on the vision task the mvnormal method performs best. This is likely due to the spatial structure of the image latents that can be learned with the covariance matrix.
> > - The mixednormal method requires the most finetuning as its performance heavily depends on the number of mixture components and can in theory represent any distributions with enough components.
> > If you think this is helpful we can add it to the paper.
> >
> > Regarding UQ decomposition, we emphasize that this work constitutes a first step toward a more comprehensive treatment of uncertainty-aware modeling. While applications such as active learning are indeed promising directions, they lie beyond the scope of the present study. Our primary contribution is the introduction of a novel method with tractable, analytically defined sampling distributions. As we demonstrate improved performance across probabilistic metrics, we provide evidence that our approach enhances uncertainty quantification in the sense of better fitting the predictive distribution.
> >
> > Nevertheless, to further support this claim, we added a quantitative analysis of predicted epistemic and aleatoric uncertainty for the KS equation in Appendix G. In particular, we examine the behavior of AU and EU alongside the predictability of the solution—measured via the Spearman correlation between the mean prediction and the ground truth—as the number of autoregressive steps $s$ increases. An additional figure, averaged over the full test set, illustrates these trends and confirms that the uncertainty estimates behave consistently with expectations for the KS equation (see lines 2167–2190).
> >
> >
> > **References**
> >
> > [1] Xizewen Han, Huangjie Zheng, Mingyuan Zhou. "CARD: Classification and Regression Diffusion Models".  Advances in Neural Information Processing Systems, 2022.
> >
> > [2] Tero Karras, Miika Aittala, Timo Aila, Samuli Laine. "Elucidating the Design Space of Diffusion-Based Generative Models". Advances in Neural Information Processing Systems, 2022.
> >
> > [3] Arnold, S., Walz, E-M., Ziegel, J., Gneiting, T. "Decompositions of the mean continuous ranked probability score," Electronic Journal of Statistics, Electron. J. Statist. 18(2), 4992-5044, (2024)
> >
> > [4] Knüppel, M., Krüger, F., Pohle, M-O., Score-based calibration testing for multivariate forecast distributions, arXiv:2211.16362v3
> >
> > [5] Gneiting, T., Balabdaoui, F., Raftery, A.E., Probabilistic Forecasts, Calibration and Sharpness, Journal of the Royal Statistical Society Series B: Statistical Methodology, Volume 69, Issue 2, April 2007, Pages 243–26We thank Reviewer wwWQ for their detailed review and help in improving our paper. We uploaded a revised version of the paper to follow the ideas and recommendations of the reviewers.
> >
> > [6] Demšar, J., 2006. Statistical comparisons of classifiers over multiple data sets. Journal of Machine learning research, 7(Jan), pp.1-30.
> >
> > [7] Nichol, A.Q. and Dhariwal, P., 2021, July. Improved denoising diffusion probabilistic models. In International conference on machine learning (pp. 8162-8171). PMLR.

---

> > > ### Comment · Reviewer_wwWQ · 2025-11-26
> > > **Response to Authors and Page Limit Reminder**
> > >
> > > I appreciate the authors' efforts in thoroughly addressing the concerns raised by all reviewers.
> > >
> > > With respect to my own comments, I find that the current version shows clear improvement over the original submission.
> > >
> > > I am willing to raise my score, provided the proposed revisions can be incorporated within the original page limit guidelines.

---

> > > > ### Author Response · Authors · 2025-11-26
> > > >
> > > > We appreciate the reviewer’s feedback and their willingness to raise their score. We also believe that the manuscript has clearly improved as a result of the reviewer’s comments and suggestions.
> > > > Regarding the page limit, we would like to point out that ICLR 2026 permits a ten-page limit during the discussion period and for the camera-ready submission: “At the time of submission, the main text should be 9 pages or fewer. During the discussion/rebuttal phase and for the camera ready, the page limit will be increased to 10 pages to allow for new results/discussions.” (https://iclr.cc/Conferences/2026/AuthorGuide).

---

> > > > > ### Comment · Reviewer_wwWQ · 2025-11-27
> > > > >
> > > > > Apologies for my oversight. I have raised the score. All the best.

---

### Official Review · Reviewer_QhqQ · 2025-11-01

**Soundness:** 3
**Presentation:** 1
**Contribution:** 4
**Rating:** 4
**Confidence:** 4

**Summary:**

This paper extends diffusion models to handle the probabilistic regression setting - that is, when one models the full conditional distribution of the outputs given the inputs, not just the conditional mean. This is done by proposing a proper scoring rule based objective, and adapting the diffusion model accordingly using a certain Gaussian mixture component. The framework is evaluated on UCI datasets as well as some PDE/weather type benchmarks, and for depth estimation.

**Strengths:**

This paper is somewhat tangential to my own research, so some comments may not be perfectly informed. With this said, there are several things I really like about this paper:
* **Very important problem.** The ability to model the full distribution of inputs automatically is a major advantage of classification setups as opposed to regression, as cross-entropy loss learns the full softmax scores and not just the largest score, and is critical to LLM success. In fact, I am surprised by is that it is not solved, as I would have thought that donig this knd of modeling successfully would be necessary to do video generation to the kind of fidelity we have today.
* **Evaluations on multiple qualitatively different domains.** The paper is much stronger by evaluating on toy datasets, climate modeling, and computer vision simultaneously.
* **Approach via scoring rules.** This is not the most obvious thing to do, which to me would have been something that leverages distribution matching via optimal transport.

**Weaknesses:**

My main concerns are:
* ***No available code.*** The authors claim to have attached code the the submission, but I am unable to access any of the two anonymized repositories because they say "The requested file is not found." for every single file except the readme.
* **Diffusion presentation is too heavy.**, especially related to diffusion model aspects. There were multiple times where I though details could have been moved to an appendix. In particular 2.2 jumps straight into formulation and could be eased, with details moved to an appendix.
* **Not enough review on scoring rules.** To many readers of this paper, this part will be new, and it needs to be reviewed in a lot more detail. I know their definition and setup by memory, and for me these aspects were barely comprehensible. For many others coming from a diffusion background this will be the new and exciting part.
* **Use of VI and mixture models.**.In other settings, it is well known that variational inference is hard to get working with mixture models, which exhibit all kinds of things like mode collapse and other problems. This is why VAEs for instance are rarely used with mixture model components. Does this aspect of the modeling really work?
* **Not obvious early-enough that mixture models will be used.** Readers should know this from the abstract and introduction, because it's a very important signal about potential performance.
* **Not enough ablations,** I would have appreciated it if it was easier to understand what parts of the pipelines are necessary, and what is qualitatively lost if one for instance drops the mixture model component, or if one keeps it but trains without scoring rules. Right now the best I can see is numbers related to performance, which is not rich enough to tell what is going on, see next point.
* **Way too much reliance on table-based evaluation and relative comparisons.** Tables give information about relative performance of models, but reveal nothing about how well they work in an absolute sense and can therefore hide serious performance problems.
* **Evaluation details are hard to read.** For example, the acronym CRPS is never defined, and is not as widespread as RMSE so it should be defined.
* **No sanity checks on visualizable toy examples.** I would have liked to see, for instance, something like a a 1D time series example that can be plotted, to sanity check the model's behavior.
* **Tables contain no +- error bars.** This makes it impossible to assess how noisy experiments are.
* **Unclear how many random seeds.** This is only revealed in the sea surface example, which uses 5 runs, which is rather small. This can be acceptable on basis of limited compute, but not otherwise - so a justification should be included if this is the reason.
* **Unclear how plots (as opposed to tables) actually evaluate how good the distribution is.** We should be comparing the true distribution with the learned distribution, for instance via Wasserstein distance or some other metric, or by plotting both side-by-side in a 1D example where this is possible (and listing something like a Kolmogorov-Smirnov distance).

**Questions:**

Please see weaknesses and address as appropriate, taking extra care to point out anywhere I might have an error so I can take another look.

In addition, below I include not just questions, but also some detailed comments:
* Typo: "the UCI benchmark" -> "UCI benchmarks"
* In what sense does p_z represent a prior? What's the likelihood here?
* In (2) parentheses are not the right size and should be made larger
* In 2.2, why opt for a discrete presentation rather than write down the SDE, which is quite a bit cleaner?
* I am confused about why the forward process, as written, is Markovian. For T=3 it reads like p(x_3 | x_0) p(x_1 | x_2, x_0) p(x_2 | x_3, x_0). Why is it that we can write  p(x_1 | x_2, x_0) and don't need additional dependence on x_3? Note that a factorization like this is not true generically, as can be seen by considering for instance the Kalman smoothing equations. I don't necessarily think there is an error here, instead I think I am confused and misunderstanding the details of the setup, so if there is a standard reference you can point me to I would appreciate this so I can better follow. Relatedly, it is also not clear to me whether this level of detail is actually needed here.
* Why not formulate the method in terms of calibration and refinement, as opposed to scoring rules? There should be an equivalent way to think about the work from this perspective, and I am guessing it would come out cleaner and easier to follow.

---

> ### Author Response · Authors · 2025-11-21
>
> [1/3]
>
> We thank Reviewer QhqQ for their detailed review and their helpful suggestions, which significantly improved the quality of our paper. We have uploaded a revised version that incorporates the reviewers’ feedback.
>
> ### Weaknesses
>
> 1. **No available code/anonymous github not working**
>
> We are surprised to hear this, as both anonymous GitHub repositories are accessible on our end. Could you please try again? To ensure reliable access for all reviewers, we additionally included the repositories as zip files in the supplementary material. We apologize for the inconvenience.
>
> 2. **Diffusion presentation is too heavy.**
>
> We acknowledge that the diffusion background is mathematically involved, especially for readers less familiar with diffusion models. Unfortunately, this level of detail is difficult to avoid, as our method directly modifies the diffusion training objective. To make our approach understandable, we must introduce the forward process, the backward process, and a short derivation of the training objective, including the DDIM formulation (necessary for its connection to Bortoli et al.).
> We attempted to keep the presentation as concise as possible and note that its complexity is comparable to standard diffusion literature (e.g., [3–7]). If the reviewer has concrete suggestions for simplification, we would be happy to revise further.
>
> 3. **Not enough review on scoring rules. I know their definition and setup by memory, and for me these aspects were barely comprehensible. For many others coming from a diffusion background this will be the new and exciting part.**
>
> Thank you for pointing this out. We agree that the short introduction directly inside the methodology section might be too short and hard to follow. Therefore, we moved the introduction of scoring rules to 2.3, where we hope that the presentation is a lot cleaner. In case you are still missing additional background, feel free to let us know what we should add.
>
> 4. **Use of VI and mixture models: Variational inference is hard to get working with mixture models, which exhibit all kinds of things like mode collapse and other problems. Does this aspect of the modeling really work?**
>
> We agree that mixture models can be challenging for variational inference, especially under the log-score, where components often collapse to Dirac masses ($\sigma = 0$ or $w = 0$), leading to numerical instabilities. While stabilized MDN training exists [2], in our setting kernel scores circumvent these issues entirely: they impose fewer distributional restrictions and remain well-defined even for collapsed components. Empirically, this led to stable training without noticeable mode-collapse problems.
>
> 5. **Not obvious early-enough that mixture models will be used. Readers should know this from the abstract and introduction, because it's a very important signal about potential performance.**
>
> We are unsure why the use of mixture models would be a major determinant of potential performance, as the mixture is applied over the noise distribution, not over the model’s output space. Thus, the diffusion model can represent multimodal target distributions regardless of the mixture model. If the reviewer can elaborate, we would be open to emphasizing this earlier in the paper.
>
> 6. **Not enough ablations, I would have appreciated it if it was easier to understand what parts of the pipelines are necessary, and what is qualitatively lost if one for instance drops the mixture model component, or if one keeps it but trains without scoring rules. Right now the best I can see is numbers related to performance, which is not rich enough to tell what is going on, see next point.**
>
> We would like to clarify that a mixture model cannot be trained with the standard diffusion loss (Eq. 6), which provides signal only for the mean. Therefore, a mixture model without scoring rules is not trainable unless one introduces a non-standard alternative diffusion loss.
> The converse is possible: scoring rules without a mixture model correspond exactly to our Gaussian models (both univariate and multivariate), which are included in our experiments. We hope this addresses the reviewer’s concern.
>
> 7. **Way too much reliance on table-based evaluation and relative comparisons. Tables give information about relative performance of models, but reveal nothing about how well they work in an absolute sense and can therefore hide serious performance problems.**
>
> Our tables mainly compare against state-of-the-art diffusion models within each task (e.g., Marigold for depth estimation). Improving over these baselines demonstrates strong absolute performance. Additionally, we included numerous qualitative results in both the main paper and the appendix to complement the quantitative comparisons. Are there additional plots that the reviewer would be interested in?

---

> > ### Author Response · Authors · 2025-11-21
> >
> > [2/3]
> >
> > 8. **No sanity checks on visualizable toy examples. I would have liked to see, for instance, something like a a 1D time series example that can be plotted, to sanity check the model's behavior.**
> >
> > Thank you for this suggestion. We have added visualizations for fixed time-steps of the 1D PDEs in Appendix D (Figures 7 and 9).
> >
> > 9. **Evaluation details are hard to read. For example, the acronym CRPS is never defined, and is not as widespread as RMSE so it should be defined.**
> >
> > Thank you for catching this. We now define CRPS in the main paper (it was previously only explained in the appendix).
> >
> > 10. **Tables contain no +- error bars. This makes it impossible to assess how noisy experiments are.**
> >
> > Due to space constraints, we placed the standard deviations in the appendix. In the revised version, we now have more space and can move them into the main paper if the reviewer prefers.
> >
> > 11. **Unclear how many random seeds. This is only revealed in the sea surface example, which uses 5 runs, which is rather small. This can be acceptable on basis of limited compute, but not otherwise - so a justification should be included if this is the reason.**
> >
> > Thank you for flagging this. We now state the number of seeds throughout. Due to the high computational cost of diffusion models in high-dimensional PDE settings, we used five seeds for the PDE tasks. The temperature dataset has a single test set with temporal structure, making multiple runs infeasible. The same holds for the Marigold datasets. For CARD, the seeds are determined by the prescribed data splits (5 for protein, 20 for the remaining datasets), consistent with the CARD paper.
> >
> > 12. **Unclear how plots (as opposed to tables) actually evaluate how good the distribution is. We should be comparing the true distribution with the learned distribution, for instance via Wasserstein distance or some other metric, or by plotting both side-by-side in a 1D example where this is possible (and listing something like a Kolmogorov-Smirnov distance).**
> >
> > Our setup provides only a single ground-truth sample per condition rather than a full ground-truth distribution. Thus, metrics such as the Wasserstein distance are not applicable. Instead, we rely on proper scoring rules, such as the CRPS, precisely because they evaluate predictive distributions under this one-sample setting.
> >
> > ### Questions
> >
> > 1. **Typo: "the UCI benchmark" -> "UCI benchmarks"**
> >
> > Fixed, thank you.
> >
> > 2. **In what sense does p_z represent a prior? What's the likelihood here?**
> >
> > Good catch. This was imprecise and we changed it to source distribution. Thank you!
> >
> > 3. **In (2) parentheses are not the right size and should be made larger**
> >
> > Thank you, we fixed it!
> >
> > 4. **In 2.2, why opt for a discrete presentation rather than write down the SDE, which is quite a bit cleaner?**
> >
> > Thank you, this is a valid point. While the SDE formulation of [3] is elegant and widely used, many works (e.g., [4, 5]) continue to present diffusion models in the discrete form. We chose the discrete version because we do not rely on the explicit SDE representation, and it keeps the exposition closer to standard training formulations.
> >
> > 5. **I am confused about why the forward process, as written, is Markovian. For T=3 it reads like p(x_3 | x_0) p(x_1 | x_2, x_0) p(x_2 | x_3, x_0). Why is it that we can write p(x_1 | x_2, x_0) and don't need additional dependence on x_3? Note that a factorization like this is not true generically, as can be seen by considering for instance the Kalman smoothing equations. I don't necessarily think there is an error here, instead I think I am confused and misunderstanding the details of the setup, so if there is a standard reference you can point me to I would appreciate this so I can better follow. Relatedly, it is also not clear to me whether this level of detail is actually needed here.**
> >
> > You are right. It is true that such factorization is not true in general. The forward process is part of the DDIM formulation introduced in [7] and is *non-Markovian* in general as we write in section 2.2. This formulation generalizes the previous framework coined DDPM which is Markovian [6]. DDIM becomes Markovian only for the specific choice of $\eta = 1.0$ and we recover the DDPM formulation exactly. The DDIM formulation is nowadays found in many works, for example in [5], which is the reason why we build on top of it in our work. For standard references, the seminal papers [6] and [7] are recommended and the recent overview [8]. There are also many helpful blog posts, such as https://lilianweng.github.io/posts/2021-07-11-diffusion-models.

---

> > > ### Author Response · Authors · 2025-11-21
> > >
> > > [3/3]
> > >
> > > 6. **Why not formulate the method in terms of calibration and refinement, as opposed to scoring rules? There should be an equivalent way to think about the work from this perspective, and I am guessing it would come out cleaner and easier to follow.**
> > >
> > > We are unsure what the reviewer means. We interpret calibration as the agreement between predictive distribution and observation, and sharpness as the concentration of the predictive distribution. Under these definitions, proper scoring rules already encode the calibration–sharpness trade-off: they maximize sharpness subject to achieving optimal calibration [1]. For the energy score, $ES= \mathbb{E}[\|X-y\|]- \frac{1}{2} \mathbb{E}[\|X-X'  \|],$ the first term reflects calibration/accuracy, while the second reflects sharpness. Thus, viewing our method through proper scoring rules is equivalent to the calibration–refinement perspective. Does that answer your question?
> > >
> > > ### References
> > >
> > >
> > > [1] Tilmann Gneiting, Fadoua Balabdaoui, Adrian E. Raftery, Probabilistic Forecasts, Calibration and Sharpness, _Journal of the Royal Statistical Society Series B: Statistical Methodology_, Volume 69, Issue 2, April 2007, Pages 243–268, [https://doi.org/10.1111/j.1467-9868.2007.00587.x](https://doi.org/10.1111/j.1467-9868.2007.00587.x)
> > >
> > > [2] Kelen, Domokos M., Ádám Jung, Péter Kersch, and Andras A. Benczur. “Distribution-Free Data Uncertainty for Neural Network Regression.” Paper presented at The Thirteenth International Conference on Learning Representations. October 4, 2024. [https://openreview.net/forum?id=pDDODPtpx9](https://openreview.net/forum?id=pDDODPtpx9).
> > >
> > > [3] Yang Song, Jascha Sohl-Dickstein, Diederik P. Kingma, Abhishek Kumar, Stefano Ermon, Ben Poole. "Score-Based Generative Modeling through Stochastic Differential Equations". Paper presented at ICLR 2021. [https://openreview.net/forum?id=PxTIG12RRHS](https://openreview.net/forum?id=PxTIG12RRHS).
> > >
> > > [4] Bingxin Ke, Anton Obukhov, Shengyu Huang, Nando Metzger, Rodrigo Caye Daudt, and Konrad Schindler. Repurposing diffusion-based image generators for monocular depth estimation. In 2024 IEEE/CVF Conference on Computer Vision and Pattern Recognition (CVPR), pp. 9492–9502, 2024. doi: 10.1109/CVPR52733.2024.00907.
> > >
> > > [5] Valentin De Bortoli, Alexandre Galashov, J Swaroop Guntupalli, Guangyao Zhou, Kevin PatrickMurphy, Arthur Gretton, and Arnaud Doucet. Distributional diffusion models with scoringrules. In Forty-second International Conference on Machine Learning, 2025. URL [https://openreview.net/forum?id=N82967FcVK](https://openreview.net/forum?id=N82967FcVK).
> > >
> > > [6] Ho, J., Jain, A. and Abbeel, P., 2020. Denoising diffusion probabilistic models. Advances in neural information processing systems, 33, pp.6840-6851. URL [https://proceedings.neurips.cc/paper/2020/file/4c5bcfec8584af0d967f1ab10179ca4b-Paper.pdf](https://proceedings.neurips.cc/paper/2020/file/4c5bcfec8584af0d967f1ab10179ca4b-Paper.pdf)
> > >
> > > [7] Song, J., Meng, C. and Ermon, S., 2020. Denoising diffusion implicit models. In ICLR. 2021. URL [https://openreview.net/forum?id=St1giarCHLP](https://openreview.net/forum?id=St1giarCHLP)
> > > [8] Lai, C.H., Song, Y., Kim, D., Mitsufuji, Y. and Ermon, S., 2025. The Principles of Diffusion Models. arXiv preprint arXiv:2510.21890. [https://www.arxiv.org/pdf/2510.21890](https://www.arxiv.org/pdf/2510.21890)

---

### Author Response · Authors · 2025-11-21
**Main revision update**

We sincerely thank all reviewers for their constructive feedback, valuable suggestions, and the time invested in evaluating our work.
We have carefully addressed all raised points and provide below a consolidated summary of the revisions made in the updated manuscript. In addition to this overview and the individual point-by-point responses, all modifications in the revised version of the paper are highlighted in blue for clarity.

**Model selection**

Based on our results, we provided some guidance to practitioners on how to select one of our proposed methods (univariate, multivariate, mixture) for a specific experiment. This is described in the conclusion lines 477 - 480.

**Additional runtime analysis**

Following the reviewers’ suggestions, we include a detailed computational runtime analysis in Appendix D.1 (Table 11). Because extensive efficiency ablations for large-scale models such as Marigold (~900M parameters) are computationally prohibitive, we focus on the two PDE tasks; given the minimal architectural and training differences, these results directly transfer to the remaining experiments. The analysis shows no significant runtime differences between DDPM and the univariate or multivariate normal variants. For the mixture normal model, runtime increases with the number of components $K$, whereas the sample-based method of Bortoli et al. (2025) [1] exhibits substantially higher computational cost. Overall, our method achieves improved performance with essentially no additional computational overhead, except in settings involving large mixture sizes.


**Additional correlation analysis for uncertainties**

To further validate the uncertainty estimates beyond individual qualitative examples, we added a quantitative analysis of predicted epistemic and aleatoric uncertainty for the KS equation in Appendix G. Specifically, we study the behavior of AU and EU together with the predictability of the solution—measured via the Spearman correlation between the mean prediction and the ground truth—for increasing autoregressive steps $s$. An additional figure summarizes this behavior, averaged over the entire test set, and confirms that the uncertainty estimates behave as expected for the KS equation (see lines 2167–2190).


**Additional univariate visualizations**

As a sanity check for our models, we provided additional visualizations of single-step predictions of the PDE tasks. This allows to more clearly assess the predictions of the models and the behavior of generated samples and the predictive mean.


**Comparison with improved DDPM***

As suggested by Reviewer wwWQ, we added a diffusion model as a baseline, which models the variance. While the approach from Bortoli et al. [1] also learn the variance, we agree that it is interesting to use more standard methods for this as well. Thus, we decided to use improved DDPM (iDDPM) [2] for this, an established diffusion model that immediately builds upon DDPM, using the same architecture but a very different loss functions. For a more indepth description and results on the KS equation, see Appendix F. The results show, that iDDPM improves the DDPM baseline, but performs worse than our univariate and mixture models.

**New Section 2.3 on Scoring Rules**

Following the suggestion of Reviewer QhqQ, to extend on our introduction of scoring rules, we moved the introduction to from 3.1 to 2.3 where we hope to provide a more gentle introduction now.

[1] Bortoli, Valentin De, Alexandre Galashov, J Swaroop Guntupalli, et al. "Distributional Diffusion Models with Scoring Rules." The Forty-second International Conference on Machine Learning, 2025.

[2] Nichol, A.Q. and Dhariwal, P., 2021, July. Improved denoising diffusion probabilistic models. In International conference on machine learning (pp. 8162-8171). PMLR.

---

### Meta-Review · Area_Chair_hqsR · 2025-12-31

**Summary:**

The authors propose an application/extension of diffusion models to a probabilistic regression setting, modelling the full conditional distribution of the target values given the covariates. Unlike classical approaches to probabilistic regression, which typically enforce Gaussianity to the predictive distribution, their work follows the recent trend of more flexible parameterisations (e.g., as previously done via flow models), making it attractive to settings in which Gaussianity does not hold. To achieve this, the authors propose a scoring-rule-based objective and evaluate the approach on UCI datasets as well as on higher-dimensional and more realistic regression tasks.

**Reviewer Concerns:**

The reviewers found this submission to be interesting and tackling a relevant problem. However, there has also been quite some debate about (i) the incremental nature and limited technical contribution of the proposed work [wwWQ, vvhz]; (ii) insufficient empirical assessment and ablations, for example, against more contemporary and related approaches [QhqQ, wwWQ, vvhz]; (iii) and issues regarding the presentation of the work [QhqQ]. Some of those challenges can be addressed by carefully drafting a resubmission, restructuring the paper, and conducting additional experiments and ablations. Hence, I encourage the authors to carefully consider the reviewers' comments for any future resubmission of the work.

**Reviewer Scores:**

In my assessment, I am assuming the following change in scores to reflect whether the reviewer's concerns have been addressed or resolved.

- QhqQ: I believe the concerns have been adequately addressed in this case. (potential increase to a 5/6)

- wwWQ: Explicitly stated to have been satisfied. (potential increase to a 5/6)

- vvhz: Explicitly stated to have been satisfied. (potential increase to a 5/6)

---

### Decision · Program_Chairs · 2026-01-26

Reject